# Evaluation of CMIP6 Models Performance in Simulating Historical Biogeochemistry across Southern South China Sea

Winfred Marshal[1], Jing Xiang Chung [2], Nur Hidayah Roseli [1,2], Roswati Md Amin[2], Mohd Fadzil Akhir [1]

[1]Institute of Oceanography and Environment, University Malaysia Terengganu, Kuala Nerus, Terengganu, Malaysia

[2]Faculty of Science and Marine Environment, Universiti Malaysia Terengganu, Kuala Terengganu 21030, Malaysia

*Correspondence to*: Jing Xiang Chung (jingxiang@umt.edu.my)

**Abstract.** This study evaluates the ability of Earth System Models (ESMs) from the Coupled Model Intercomparison Project Phase 6 (CMIP6) to simulate biogeochemical variables in the southern South China Sea (SCS). The analysis focuses on key biogeochemical variables: chlorophyll, phytoplankton, nitrate and oxygen, based on their availability in the selected models at annual and seasonal scales. The models' performance is assessed against Copernicus Marine Environment Monitoring Service (CMEMS) data using statistical metrics such as the Taylor diagram and Taylor skill score. The results show that the models generally capture the observed spatial patterns of surface biogeochemical variables. However, they exhibit varying degrees of overestimation or underestimation in their quantitative measure. Specifically, their mean bias error ranges from -0.02 to +2.5 mg/m³ for chlorophyll, -0.5 to +1 mmol/m³ for phytoplankton, -0.1 to +1.3 mmol/m³ for nitrate, and -2 to +2.5 mmol/m³ for oxygen. The performance of the models is also influenced by the season, with some models showing better performance during June, July, August than December, January, February. Overall, the top five best-performing models for biogeochemical variables are MIROC-ES2H, GFDL-ESM4, CanESM5-CanOE, MPI-ESM1-2-LR and NorESM2-LM. The findings of this study have implications for researchers and end-users of the datasets, providing guidance for model improvement and understanding the impacts of climate change on the SCS ecosystem.

## 1 INTRODUCTION

Climate change has profound and wide-ranging effects on marine ecosystems, impacting both the physical environment and the primary productivity that inhabit it. Marine primary productivity plays a crucial role in sustaining life in the oceans and has far-reaching implications for the entire planet. Under climate change, understanding the importance of marine primary productivity becomes even more critical due to its various ecological, economic, and climate-related implications. For example, Kwiatkowski et al. (2020) discovered that the multi-model global mean projections from the Coupled Model Intercomparison Project Phase 6 (CMIP6), under high-emission to low-emission scenarios, indicate a consistent decrease in net primary production. Notably, there is a significant increase in inter-model uncertainty compared to CMIP5. This increased uncertainty is linked to changes in the temporal patterns of phytoplankton resource availability and grazing pressure within CMIP6 (Kwiatkowski et al., 2020). This carries significant implications for evaluating ecosystem impacts on

30 a regional scale. (Tagliabue et al., 2021). Ocean Biogeochemistry (BGC) models are essential tools in understanding and simulating the interactions between the physical, chemical, and biological processes that occur in the ocean system. These models incorporate the cycling of key elements such as chlorophyll, phytoplankton, zooplankton, carbon, nitrogen, phosphorus, oxygen, etc., through the atmosphere and ocean ecosystems. The importance of ocean BGC models lies in their ability to provide a more comprehensive and integrated understanding of the marine environment. The reliability and

35 accuracy of climate projections made by climate models are closely tied to how well these models are able to replicate or simulate past climate conditions (Jia et al., 2023; Shikha and Valsala, 2018; Tang et al., 2021). While a model's successful reproduction of historical climate patterns suggests it has captured relevant physical processes and interactions within the Earth's system but this is not always the case. Ocean BGC models can sometimes appear to accurately represent historical climate patterns for incorrect reasons, as their results are highly dependent on the physical forcing applied (Friedrichs et al.,

40 2006; Sinha et al., 2010). For example, minor changes in ocean model circulation can lead to significant variations in biogeochemical conditions. Similarly, Glessmer et al. (2008) discovered that even slight alterations in mixing greatly affects the simulation of primary production and export in the global general circulation models. Therefore, caution is needed when using these models for future climate projections. As part of the World Climate Research Programme (WCRP), the Coupled Model Intercomparison Project Phase 6 (CMIP6) oversees the implementation of General Circulation Models (GCMs) by

45 multiple modeling institutions, aiming to simulate the Earth's climate system behaviour to study a wide range of climate-related phenomena such as climate variabilities and past & future climate projection ( Mohanty et al., 2024; Peng et al., 2021; Pereira et al., 2023; Petrik et al., 2022). This simulation aims to explore how Earth's climate responds to various climate forcing under distinct scenarios known as "Socioeconomic pathways (SSPs)". The intention is to provide a broader array of potential futures for simulation studies (Riahi et al., 2017). For example, Kwiatkowski et al. (2020) indicates that

50 forthcoming climate change is anticipated to exert a noteworthy and adverse influence on ocean biogeochemistry across different CMIP6 Shared Socioeconomic Pathways (SSPs). Specifically, the low-emission scenario SSP1-2.6 and the high-emission scenario SSP5-8.5 are projected to induce moderate to highly severe alterations. By the conclusion of the 21st century, global mean sea surface temperature is expected to rise, while surface pH, subsurface oxygen and nitrate concentration are anticipated to decrease. These transformations are likely to negatively affect ocean productivity, resulting

55 in a global mean decline. The performance of CMIP6 models varies more at the regional scale than at the global scale (Oh et al., 2023). This is because regional climate features are more sensitive to the details of the models' representations of physical processes, such as cloud formation, convection, submesoscale eddies, wave-interactions, etc. However, before we can leverage GCMs to study regional biogeochemical changes, rigorous performance testing is needed. These tests ensure the accuracy and reliability of model results, paving the way for reliable future studies.

60   There are few studies have evaluated the effectiveness of CMIP6 ocean models in simulating different biogeochemical variables over the globe scale but Intercomparison of CMIP6 BGC model's performance and ranking them according to their performance skill in regional scale had not done yet. For example, Petrik et al. (2022) evaluates the representation of mesozooplankton in six CMIP6 Earth System Models (ESMs) and compared the models' simulated

mesozooplankton biomass and distribution to observations and assess their ability to capture the observed relationship
between mesozooplankton and chlorophyll (a proxy for phytoplankton) and finds that the six CMIP6 ESMs generally
represent the large regional variations in mesozooplankton biomass at the global scale. Three of the ESMs simulate a
mesozooplankton-chlorophyll relationship within the observational bounds, which can be used as an emergent constraint on
future mesozooplankton projections. However, there is a wide ensemble spread in projected changes in mesozooplankton
biomass, reflecting the uncertainties in the models' representation of mesozooplankton in global scale. Tjiputra et al. (2020)
provides an in-depth assessment of the ocean biogeochemistry component of the Norwegian Earth System Model
(NorESM2) and discussed the implications of their findings for understanding and predicting future ocean biogeochemical
changes in global scale. NorESM2 represents a significant advancement in ocean BGC modeling, incorporating a
comprehensive representation of biogeochemical processes and demonstrating improved skill in simulating observed ocean
biogeochemical properties. Similarly, Christian et al. (2022) presents a comprehensive overview of the ocean BGC
components of two new versions of the Canadian Earth System Model (CanESM), CanESM5 and CanESM5-CanOE and
describe the models in detail and compare their performance against observations and other CMIP6 models in global scale.
CanESM5-CanOE shows improved skill relative to CanESM5 for most major tracers at most depths. However, both
CanESM5 models have some biases, such as an underestimation of surface nitrate concentrations in the subarctic Pacific and
equatorial Pacific and an overestimation in the Southern Ocean. Furthermore, Kwiatkowski et al. (2020) demonstrated that
the projected changes in global oceanic impact drivers from CMIP6 models, increases with radiative forcing across the SSPs.
This underscores the advantages of reducing emissions for upper-ocean ecosystems. The anticipated warming, acidification,
and deoxygenation in the benthic ocean are less pronounced compared to the surface, with increased inter-model uncertainty
relative to scenario uncertainty. This opens a way to perform more regional Intercomparison skill assessments on CMIP6
BGC models.

Sunda shelf region of southern South China Sea (SCS) is located in the centre of the Southeast Asian monsoonal
system with heavy precipitation rates (You and Ting, 2021), river input that deliver freshwater (Lee et al., 2019), dissolved
nutrients (Jiang et al., 2019) and surrounded by volcanic islands, the Himalaya in the background, this region is characterized
by one of the largest sediment discharge rates worldwide (Milliman et al., 1999). The Sunda Shelf Sea in Southeast Asia
stands out as one of the world's largest and most diverse shelf seas. Despite its ecological significance, it faces considerable
human population density along its coastline, leading to substantial stress on its marine habitats. This is particularly evident
in urbanized marine ecosystems exposed to significant human-induced pressures (Todd et al., 2019). Concurrently, our
knowledge of the biogeochemistry of tropical shelf seas lags behind that of higher-latitude environments, posing challenges
in predicting the impact of anthropogenic pressures on tropical seas (Lønborg et al., 2021). This calls for the need for
credible future BGC projections to help devise appropriate mitigation measures to curb and mitigate the impacts of climate
change in this region. Based on this backdrop, this study objectively aims at ranking 13 CMIP6 ocean models' historical
simulations based on their ability to reproduce the selective reference biogeochemical variables such as chlorophyll,
phytoplankton, nitrate and oxygen over the southern SCS. The rest of this paper will be structured as follows: Section 2 gives

a brief description of the study domain and the dataset used. Section 3 will elaborate the methodology employed. Section 4 presents the results and discussion under sub-topics; spatial variation and bias, Taylor diagram and model ranking. Conclusion of the study is summarized in Section 5.

## 2 STUDY DOMAIN AND DATA

### 2.1 Study domain

This study focuses on the Sunda shelf region, also referred to as the southern SCS, delineated by latitudes 8°S – 15°N and longitudes 98°E – 121°E, as illustrated in **Fig. 1**. Situated within the tropical rim of the Northwestern Pacific Ocean, this region is an integral part of one of the world's largest marginal seas, the SCS. The southern SCS is characterized by a shallow bathymetry, with a maximum depth of approximately 100 meters, except for the central part, where depths exceed 1,000 meters. The region's circulation and hydrodynamics are strongly influenced by monsoonal winds, along with other factors such as complex bathymetry, coastline configuration, the presence of large islands, river discharge, mixing, upwelling, internal waves, and eddies (Daryabor et al., 2014, 2015). In southern SCS, ocean circulation exhibits substantial variations driven by the monsoon cycle (Gan et al., 2016). During the northeast monsoon, spanning December to February, winds prevail from the northeast, generating a basin-wide cyclonic gyre within the southern SCS. Conversely, during the southwest monsoon, from June to August, winds blow from the southwest, establishing a double-gyre circulation pattern in the southern SCS. The transition between the northeast monsoon and southwest monsoon circulation patterns is gradual, occurring over several weeks. While the timing of this transition varies from year to year, it typically takes place in April and October. The monsoon-driven circulation in the southern SCS has significant implications for the region's biogeochemistry and marine ecosystems. For instance, the northward currents during the southwest monsoon transport nutrients from the equator to the northern SCS, fostering high levels of productivity in this region. Conversely, the southward currents during the northeast monsoon transport nutrients from the northern SCS to the equator, where they are utilized by phytoplankton and other marine organisms (Liu et al., 2002).

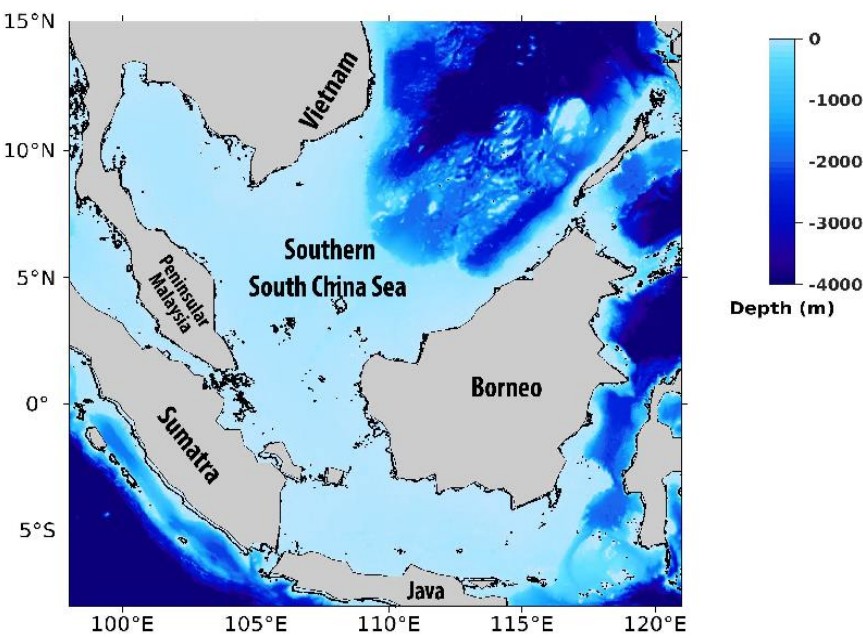

**Figure 1.** Map and bathymetry of the study domain

## 2.2 Datasets

Within the southern SCS, extensive observations have demonstrated that phytoplankton growth, serving as the primary source of organic matter, significantly influences oceanic carbon cycles. This phytoplankton growth is influenced by monsoon-driven physical and biogeochemical processes, with phytoplankton demonstrating a notable sensitivity to these environmental dynamics (Pinkerton et al., 2021; Yuwono and Rendy, 2023). These processes enhance mixing throughout the basin, influencing the overall nutrient supply and primary productivity in the euphotic zone (Palacz et al., 2011; Tseng et al.,

2005). Therefore, chlorophyll, phytoplankton, nitrate and oxygen are the primary biogeochemical variables examined in this study. We analysed the historical experiment outputs of 13 CMIP6 ESMs for the study region. The model designations and spatial resolutions are provided in **Table 1**. This selection was based on the common availability of the chosen variables and their corresponding socioeconomic scenario projections at the time of this study. The selection of the evaluation period was primarily based on the availability of reference datasets for comparison with the model outputs. The Copernicus Marine

Environment Monitoring Service (CMEMS) data (von Schuckmann et al., 2020), a standardized collocated grid with a horizontal resolution of 1/4 degree (approximately 27 km) and temporal coverage from 1993 onwards, was chosen as the reference dataset to assess the models' ability to simulate biogeochemical variables over the southern SCS. CMEMS biogeochemistry model data is the only available timeseries biogeochemical hindcast dataset. Although this product is assimilation free dataset, rigorous validations has been done to evaluate the CMEMS global product quality. Mercator-

Ocean's Quality Information Document (QuID) confirms and published the quality of this data through comparisons with

recognized datasets like Ocean Color, World Ocean Atlas and GlobColour products (Perruche et al., 2019) and prove to be of high skills in reproducing the climatology and variability of the available biogeochemical variables. In order to improve more confidence in this dataset for our study region, Wahyudi et al. (2023) validated the POC, Chlorophyll, Dissolved Oxygen, Nitrate, Phosphate and Silicate obtained from CMEMS biogeochemistry product by comparing it with in-situ data

collected during the Widya Nusantara Expedition 2015 (Triana et al., 2021) in the upwelling area of southwestern Sumatra waters. They found that the mean absolute percentage error values were lower than 15%, indicating the reliability of the CMEMS biogeochemistry model data in our study area. Additionally, Chen et al. (2023) also used the daily chlorophyll concentration data from the same CMEMS biogeochemical product in south china sea region. By utilizing this CMEMS biogeochemistry model dataset, Wahyudi et al. (2023) and Chen et al. (2023) highlights the proficiency of the CMEMS

biogeochemistry model data in reproducing both the climatic patterns and fluctuations observed within its biogeochemical variables in our study domain. This gave us confidence in using this CMEMS biogeochemical dataset as the reference model to assess other models in southern SCS region. Furthermore, we have assessed the CMEMS product using observation data from the World Ocean Atlas 2018 (WOA18) for nitrate and oxygen, and satellite data from GlobColour (Product ID: OCEANCOLOUR_GLO_BGC_L4_MY_009_104) for chlorophyll and phytoplankton. The validation results are presented

in supplementary **Table S1**, with the spatial percentage bias detailed in supplementary **Figs. S1 – S2**. The validation results demonstrated good agreement, with region-wide differences less than ±5% for chlorophyll and phytoplankton, and less than ±10% for nitrate and oxygen. The spatial pattern comparison indicates that the largest differences between the CMEMS and WOA observation data occur in coastal areas. These differences may be attributed to the insufficient number of WOA observation data in our study domain and the coarse resolution of WOA (~111 km). However, the small differences between

their climatology (less than ±10%) give us confidence that CMEMS is reliable. Therefore, given that CMEMS has all the required parameters, and our analysis established the reliability of the CMEMS in our study region, we believe that using CMEMS as a reference data allows for a fair performance assessment of the CMIP6 ESMs across all the parameters evaluated. The evaluation method relies on the analysis of an average year to represent the regional climatology. A longer period is generally considered more representative, and within the constraints of data availability, the 22-year period from

1993 to 2014, encompassing the end of the CMIP6 historical 120 experiments, was selected. As the CMIP6 models have different horizontal scales, all model outputs were regridded to a common horizontal resolution using the bilinear interpolation method. The CMIP6 climate models are publicly available and archived at https://esgf-node.llnl.gov/search/cmip6/, while the CMEMS data can be accessed at https://data.marine.copernicus.eu/products

**Table 1.** List of 13 CMIP6 models used including model name, modeling institution, coupled BGC Model and the horizontal resolution.

| Model Abbreviation | Institution | BGC Model | Horizontal Resolution | References |
|---|---|---|---|---|
| ACCESS-ESM1-5 | CSIRO, Australia | WOMBAT | 250 km | (Ziehn et al., 2020) |

| | | | | |
|---|---|---|---|---|
| CanESM5 | CCCma, Canada | CMOC | 100 km | (Swart et al., 2019) |
| CanESM5-CanOE | | CanOE | | |
| GFDL-ESM4 | NOAA-GFDL, USA | COBALTv2 | 1º x 1º degree | (Dunne et al., 2020) |
| MIROC-ES2H | MIROC, Japan | OECO2 | 100 km | (Kawamiya et al., 2020) |
| MIROC-ES2L | | | | (Hajima et al., 2020) |
| MPI-ESM-1-2-HAM | Consortium of Switzerland, Germany, UK, Finland | HAMOCC6 | 220 km | (Neubauer et al., 2019) |
| MPI-ESM1-2-HR | Max Planck Institute for Meteorology, Germany | | 50 km | (Müller et al., 2018) |
| MPI-ESM1-2-LR | | | 220 km | (Mauritsen et al., 2019) |
| MRI-ESM2-0 | Meteorological Research Institute, Japan | MRI.COM4.4 | 100 km | (Yukimoto et al., 2019) |
| NorESM2-LM | NCC, Norway | iHAMMOC | 100 km | (Tjiputra et al., 2020) |
| NorESM2-MM | | | | |
| UKESM1-0-LL | Met Office Hadley Centre, UK | MEDUSA2 | 100 km | (Sellar et al., 2019) |

## 3 METHODOLOGY

### 3.1 Evaluation metrics and Ranking

To evaluate the ability of CMIP6 models to simulate biogeochemical variables in comparison to CMEMS data, spatial variation, Mean Bias Error (MBE), Correlation Coefficient (CC), Root Mean Square Difference (RMSD), and Normalized Standard Deviation (NSD) were employed. CC, RMSD, and NSD were visualized using Taylor Diagram (TD) which offers a succinct statistical summary of the degree of similarity in patterns between simulated and reference data based on their CC, RMSD, and variance ratio. Smaller values of RMSE and BIAS indicate better model performance, while a larger positive

value of CC, ranging from -1 to 1, suggests improved correlation between the simulated and reference climate variables. The specific equations used to calculate MBE and CC, RMSD and SD using TD are presented in equations (1) – (4), respectively.

$$\text{MBE} = \frac{1}{n} \sum_{i=1}^{n}(M_i - R_i) \tag{1}$$


$$CC = \frac{\sum_{i=1}^{n}(M_i - \overline{M})\,(R_i - \overline{R})}{\sqrt{\sum_{i=1}^{n}(M_i - \overline{M})}\sqrt{\sum_{i=1}^{n}(R_i - \overline{R})}} \qquad (2)$$

$$RMSD = \sqrt{\frac{1}{n}\sum_{i=1}^{n}(M_i - R_i)^2} \qquad (3)$$

$$SD = \sqrt{\frac{1}{n}\sum_{i=1}^{n}(M_i - \overline{M})^2} \qquad (4)$$

$$TSS = \frac{(1+R)^4}{4\left(SDR + \frac{1}{SDR}\right)^2} \qquad (5)$$

Where 'n' represents the total number of grids within the ocean areas of the analysis domain, and $M_i$ and $R_i$ denote the
model and reference at $i^{th}$ grid, respectively. $\overline{M}$ and $\overline{R}$ represent the mean values of the model and the reference data.

The assessment of model performance hinges on various factors, including the specific variables, the regions under analysis, and the chosen evaluation metrics. Achieving a fair and standardized comparison necessitates consideration of these elements. In this context, models were ranked based on their annual performance utilizing the Taylor Skill Score (TSS), as outlined in equations (5). Here, 'R' signifies the pattern correlation between the models and the reference data,
while 'SDR' stands for the ratio of the spatial standard deviations of the models to that of the reference. The Taylor Skill Score quantifies the resemblance between the model and reference data concerning both the distribution and amplitude of the spatial pattern (Taylor, 2001). All evaluation metrics were applied to each variable, leading to the generation of individual and overall rankings for the models. All analyses in this study were conducted using MATLAB software and its numerical functionalities; Scientific colour maps 7.0 was used to make figures that are readable by readers with colour vision
impairments (Crameri et al., 2020, 2021).

## 4 RESULTS & DISCUSSION

### 4.1 Spatial variation & Bias

Although temporal cycles, such as the yearly cycle of seasons, are indeed important components of climate variability
(Behrenfeld et al., 2006; Kwiatkowski et al., 2017), they offer only a partial perspective on the long-term changes associated with climate change. These long-term changes encompass shifts in average temperatures, alterations in precipitation patterns, changes in the frequency and intensity of extreme weather events, and other systemic shifts that extend beyond the

periodicity of seasonal cycles. In contrast, spatial patterns provide a more comprehensive understanding of how climate is changing across different regions and ecosystems (Chi et al., 2023). Consequently, the spatial distributions of each

biogeochemical variable were analysed seasonally, during the southwest and northeast monsoons, and compared to reference data. This approach provides a general overview of the models, their differences from observations, and their relative performance. The months June, July, and August (JJA) as southwest (summer) and December, January, and February (DJF) as northeast (winter) were selected to represent the respective seasons. Seasonal bias and their mean bias error are presented in **Fig. 2 - 25**, demonstrating the ability of each model to reproduce the seasonal distribution for the southern SCS region.

While most CMIP6 climate models effectively capture the reference seasonal pattern of each biogeochemical variable, some models exhibit overestimations or underestimations of the observed magnitude.

Three ESMs, namely ACCESS-ESM1-5, CanESM5-CanOE, and MPI-ESM1-2-HR, consistently showed an overestimation of chlorophyll concentrations during both the DJF and JJA seasons **(Figs. 2, 3)**. Their mean bias error surpassed +0.1 mg/m$^3$, indicating a notable discrepancy between the simulated and reference chlorophyll levels **(Fig. 4)**. This

overestimation raises concerns about potential shortcomings in these models' representation of biogeochemical processes governing phytoplankton growth and chlorophyll production. In contrast to the overestimating models, CanESM5 consistently underestimated chlorophyll concentrations in both seasons, with a mean bias error of -0.02 mg/m$^3$. This suggests that the model consistently generated chlorophyll values lower than those reference data. Possible explanations for this underestimation could be an underrepresentation of nutrient availability or an overestimation of grazing pressure on

phytoplankton. The remaining ESMs demonstrated a moderate ability to replicate reference chlorophyll concentrations. Here, "moderate" represent the model performs at an average level in simulating the variables compared to the reference data. Their mean biases generally fell within an acceptable range of ≤ ±0.1 mg/m$^3$, indicating that these models capture the overall patterns of chlorophyll distribution in the southern SCS. Three models, namely CanESM5-CanOE, MPI-ESM1-2-HR, and UKESM1-0-LL, consistently overestimated phytoplankton carbon levels in both seasons **(Fig. 5, 6)**, exhibiting

mean bias error exceeding +0.5 mmol/m$^3$ **(Fig. 7)**. This overestimation suggests potential shortcomings in these models' representation of phytoplankton growth and carbon fixation processes. For example, UKESM1-0-LL model does not overestimate chlorophyll but overestimates phytoplankton. This could stem from that UKESM1-0-LL model explicitly simulates chlorophyll concentrations, allowing for a more accurate representation of chlorophyll levels (Sellar et al., 2019). However, UKESM1-0-LL uses nitrogen as its primary model currency, which results in a more pronounced quantitative

representation of nutrient levels. This might lead to enhanced nutrient uptake by phytoplankton due to differences in model parameterizations and consequently result in the overestimation of phytoplankton biomass. In contrast, CanESM5 exhibited a persistent underestimation of phytoplankton carbon throughout the year. Its mean bias error of -0.5 mmol/m$^3$ highlights a discrepancy between simulated and reference phytoplankton carbon levels. This underestimation could stem from factors such as an overestimation of zooplankton grazing or an underestimation of phytoplankton productivity. The remaining

ESMs, with the exception of CanESM5, moderately replicated the reference phytoplankton carbon patterns. Their mean bias error was generally within an acceptable range of ≤ ±0.4 mmol/m$^3$.

Most models showed spatial uniformity in their underestimation or overestimation of chlorophyll and phytoplankton, with a few models exhibiting spatial diversities in their estimates. For example, the CanESM5-CanOE model consistently overestimates chlorophyll concentration and phytoplankton biomass in both seasons, with mean bias error of +0.28 mg/m³ in DJF and +1.7 mg/m³ in JJA for chlorophyll, and +1.2 mmol/m³ in DJF and +0.8 mmol/m³ in JJA for phytoplankton. This overestimation is particularly pronounced in the region between Sumatra, Peninsular Malaysia, and Borneo, where chlorophyll exceeds 1 mg/m³ and phytoplankton exceeds 5 mmol/m³. Similarly, the UKESM1-0-LL model overestimates chlorophyll and phytoplankton in both seasons, especially in the Gulf of Thailand. These models may have insufficient spatial resolution to capture the fine-scale physical and biological processes in these regions. Important features like small-scale currents, eddies and upwelling events, which significantly affect chlorophyll and phytoplankton distributions, may not be adequately resolved, leading to spatial bias. Generally, ESMs in CMIP6 are developed for open ocean conditions rather than shelf seas. Most CMIP6 ESMs have a coarse resolution ($\geq$ 100 km horizontal) for the ocean component, although some have resolutions of $\leq$ 25 km, which is considered eddy-permitting. This suggests these ESMs can represent barotropic processes at smaller scales but not baroclinic ones (Chelton et al., 1998). The ability of coarse-resolution CMIP6 ESMs to represent shallow continental shelf waters dynamics with high skill, such as the southern SCS' Sunda shelf region, is limited. Variability in this region is influenced by inflows like the Indonesian Throughflow and SCS Throughflow, which are not resolved by coarse-resolution models (Wang et al., 2024).

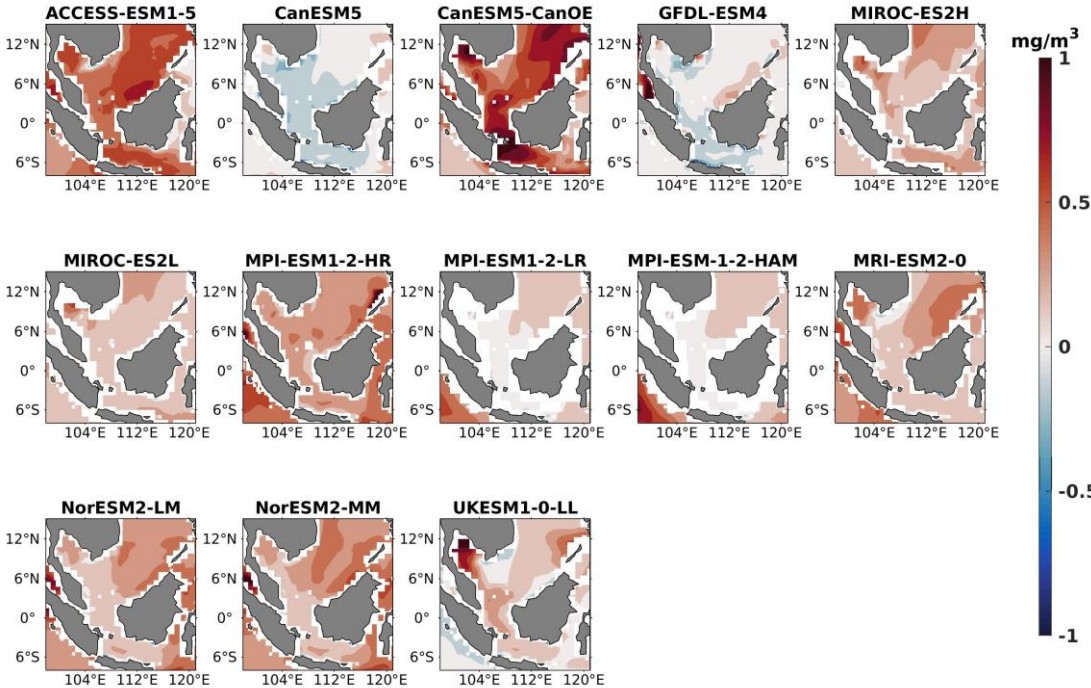


**Figure 2.** DJF spatial biases of surface chlorophyll for 13 individual CMIP6 ESMs relative to reference.

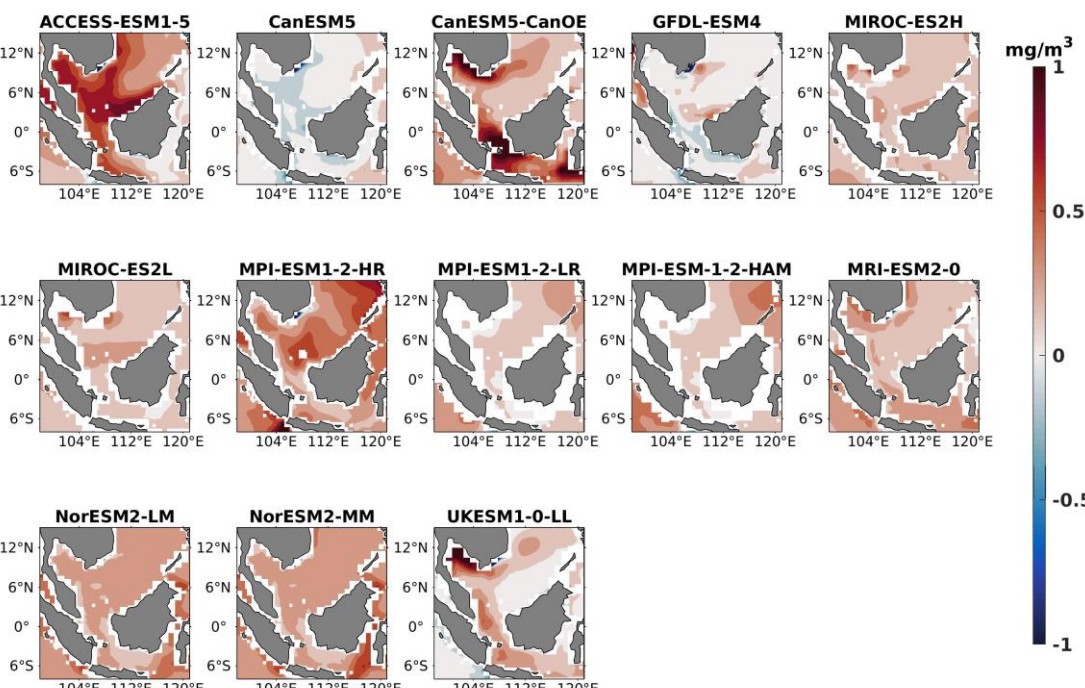

**Figure 3.** Same as Fig.2 but for JJA.


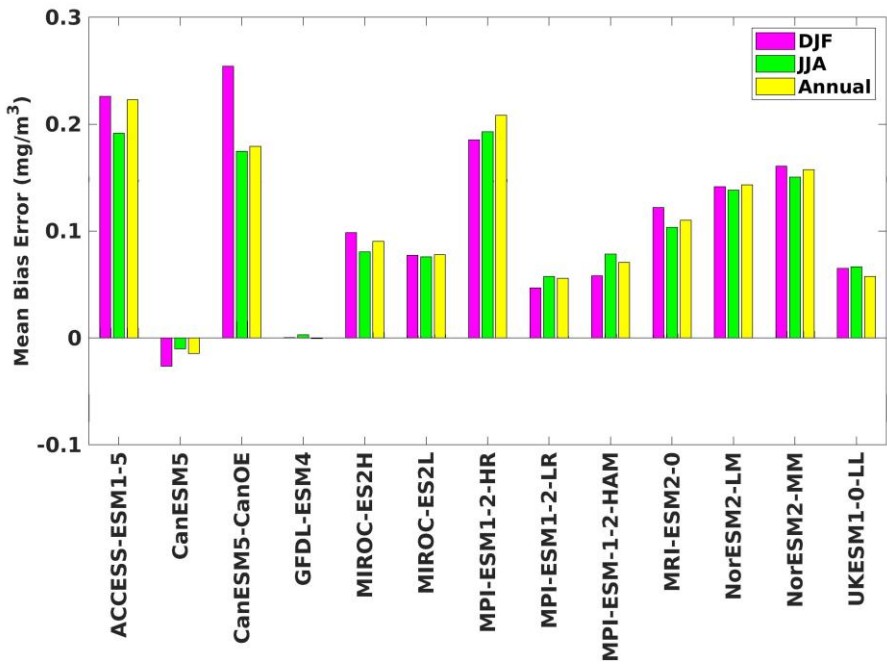

**Figure 4.** The mean bias error of surface chlorophyll for both seasons (DJF, JJA) and annual.

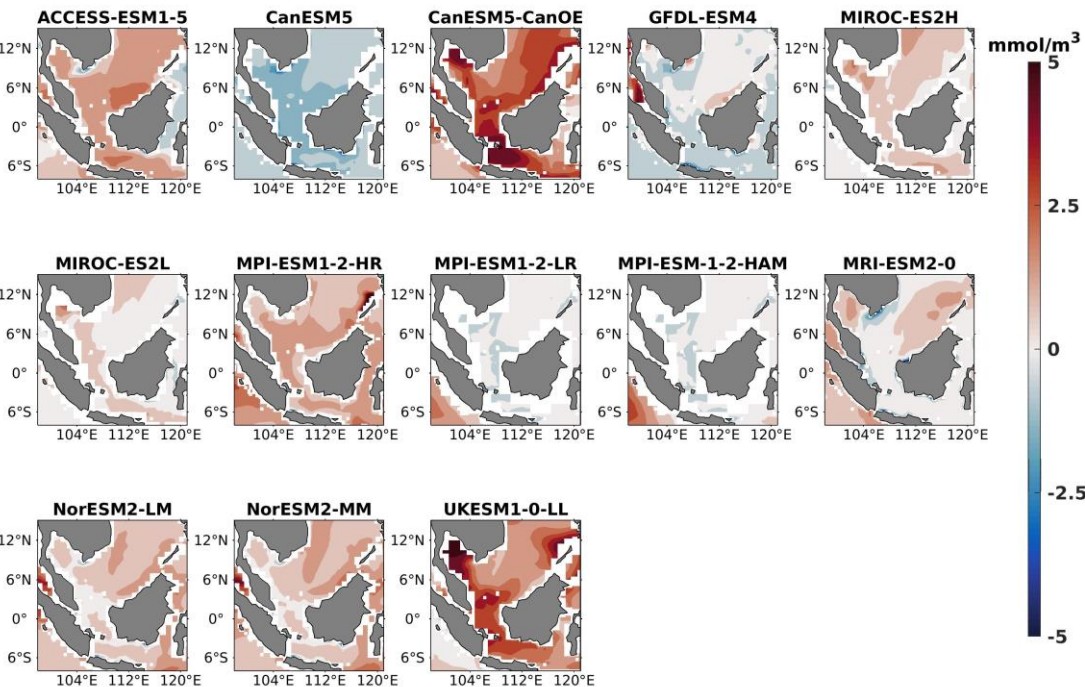

**Figure 5.** DJF spatial biases of surface phytoplankton for 13 individual CMIP6 ESMs relative to reference.

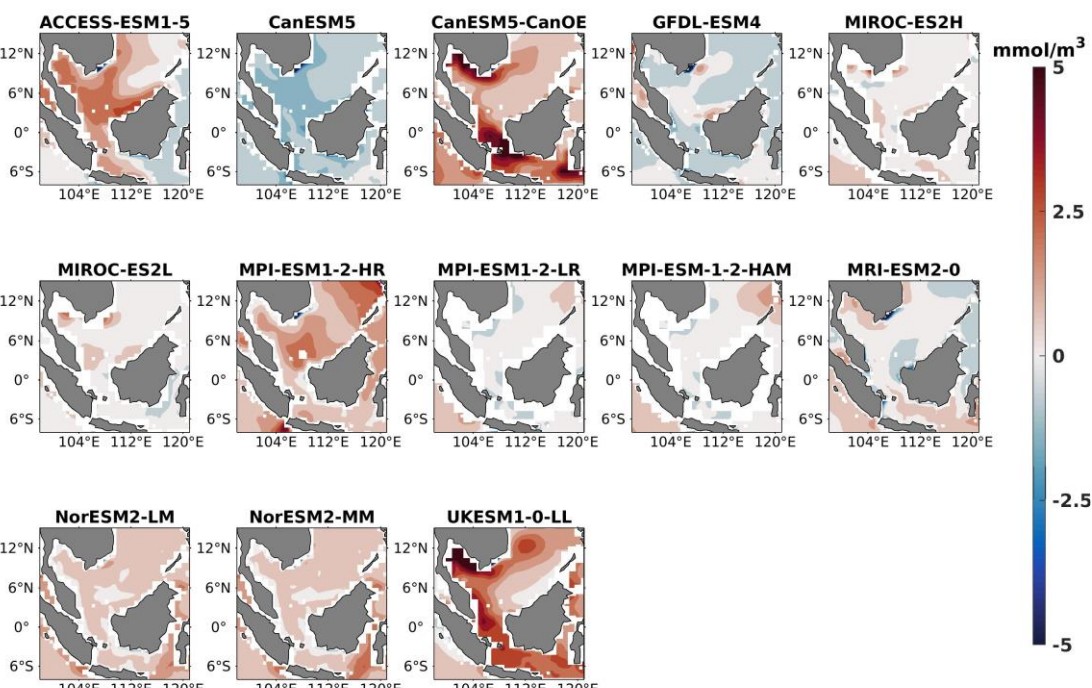

**Figure 6.** Same as Fig. 5 but for JJA.

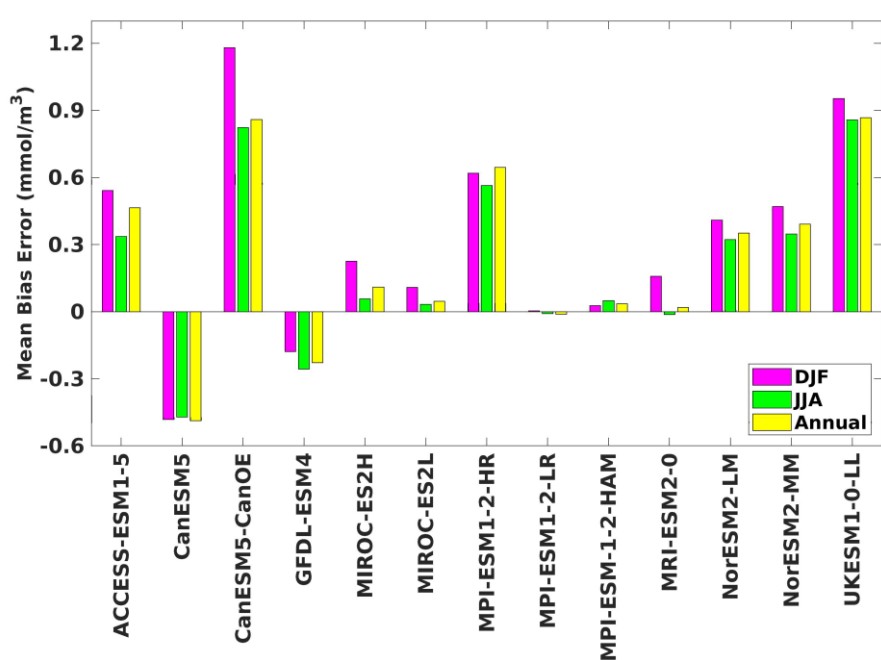


**Figure 7.** The mean bias error of surface phytoplankton for both seasons (DJF, JJA) and annual.

The analysis revealed a divergent pattern among ESMs in replicating reference surface nitrate concentrations in both seasons **(Fig. 8, 9)**. ACCESS-ESM1-5 exhibited an extreme overestimation of nitrate levels, with a mean bias error > +1 mmol/m$^3$, suggesting potential shortcomings in the model's representation of nitrate uptake by phytoplankton or denitrification processes. GFDL-ESM4 and NorESM2-based models also displayed substantial overestimations, with mean bias error exceeding +0.4 mmol/m$^3$ **(Fig. 10)**. These overestimations could stem from factors such as an underestimation of nitrate removal processes or an overestimation of nutrient inputs from rivers. The remaining ESMs moderately replicated the reference nitrate patterns, indicating a reasonable representation of nitrate dynamics in these models. Furthermore, delving into model biases at deeper levels, especially concerning nutrient dynamics, will provide more insights into the model's accuracy in simulating the nutricline. Therefore, we have presented the nitrate profile for each selected model compared with reference data (CMEMS) and observation data (WOA18) in supplementary **Figs. S3 and S4**. For simplicity, here we have discussed the nitrate concentration biases at depths of 70m **(Figs. 11 – 13)** and 1000m **(Figs. 14 - 16)**. Similar to surface nitrate, most models exhibited a positive bias at 70m with an average mean bias error ranging at +0.5 to +3 mmol/m$^3$ across the study area but at 1000m, the models exhibit mean bias error range of -1 to +2 mmol/m$^3$. Among these models, MPI-based models showed the least positive bias at 70m depth; however, as depth increased to 1000m, their bias shifted towards the negative **(Fig. 13 and 16,** respectively**)**. MIROC-based and MPI-based models exhibited the least bias in nitrate concentrations at both surface and deep layers compared to reference data. This may be attributed to the near balance achieved between nitrogen cycle sources (such as nitrogen fixation, atmospheric nitrogen deposition, and riverine nitrogen input) and sinks (including denitrification, nitrous oxide emission, and sedimentary loss) over the long spin-up period (Mauritsen et al., 2019; Hajima et al., 2020). In contrast, CanESM5-based models demonstrated minimal nitrate bias at the surface but showed varying positive and negative biases in deep layers. These discrepancies arise from the simplified parameterization of denitrification in their BGC models. In these models, denitrification in the deep layers is set to balance the rate of nitrogen fixation and is vertically distributed in proportion to the detrital remineralization rate. However, in reality, nitrogen fixation and denitrification are not constrained to balance within the water column at any single location; rather, denitrification primarily occurs in anoxic areas (Swart et al., 2019). Notably, no seasonal variations in bias in all selected models were observed at the deep layer (1000m; **Fig. 16**).

Three models, namely MIROC-ES2H, MPI-ESM1-2-LR, and MPI-ESM-1-2-HAM, consistently overestimated surface oxygen levels in both seasons **(Fig. 17, 18)**, exhibiting mean bias error exceeding +0.5 mmol/m$^3$. This overestimation suggests potential shortcomings in these models' representation of oxygen production through photosynthesis or oxygen consumption through respiration and microbial processes. Conversely, ACCESS-ESM1-5, CanESM5 and NorESM2-based models displayed persistent underestimations of oxygen throughout the year. Their mean bias error exceeding -1 mmol/m$^3$ highlight a discrepancy between simulated and reference oxygen levels **(Fig. 19)**. This underestimation could stem from factors such as an overestimation of oxygen consumption processes or an underestimation of oxygen production through photosynthesis. The remaining ESMs moderately replicated the reference oxygen patterns, indicating a reasonable representation of oxygen dynamics in these models. Similar to the nitrate profile, we have also presented the oxygen profile

for each selected model compared with reference data (CMEMS) and observation data (WOA18) in supplementary **Figs. S5 and S6**. For simplicity, here we have discussed the oxygen concentration biases at depths of 70m and 1000m. During the observation of oxycline dynamics in most of the models, it was noted that the oxygen exhibited a negative bias at a depth of 70m and transitioning to a positive bias with increasing depth (1000m) **(Fig. 22 and 25)**. Moreover, UKESM1-0-LL consistently exhibited a substantial negative mean bias error from the surface to the depth of 70m ($> +15$ mmol/m³) and shifts to positive bias of $> +7$ mmol/m³ at 1000m relative to its surface bias. Similarly, CanESM5 and MIROC-based models also displayed markedly high positive biases at a depth of 1000m, but with comparatively lesser biases at 70m. Multiple factors could contribute to biases in the simulation of nutricline/oxycline dynamics by models. Inaccuracies in simulating nutricline dynamics may arise from errors in parameterizing physical, chemical and biological processes relevant to these dynamics. In DJF, most models overestimate oxygen levels at the surface and underestimate at a depth of 70 meters. This bias in oxygen concentration may result from excessively intense winter mixing of cold, oxygen-rich waters from the northern boundary of the southern SCS into the Sunda Shelf region (Thompson et al., 2016). This intense mixing leads to a surplus of oxygen being brought to the surface, causing models to predict higher than actual oxygen levels there. Simultaneously, this mixing reduces the oxygen concentration at intermediate depths, as the oxygen-rich water is redistributed upwards, resulting in underestimated oxygen levels at 70 meters. Additionally, nutrient trapping issues may also contribute to the remaining model bias (Six and Maier-Reimer, 1996). Moreover, the exclusion of relevant processes or feedback mechanisms influencing nutricline dynamics within the model, such as nutrient upwelling, microbial remineralization and ocean stratification, may lead to biased simulations. Additionally, structural uncertainties embedded in the model formulation, including simplifications or assumptions regarding complex processes, may also play a role in generating biases in simulation results. For example, advancements in model parameterization and representation of biogeochemical fluxes have led to consistent improvements in the mean states of nutrient dynamics in CMIP6 models (Séférian et al., 2020), such as GFDL-ESM4, MIROC-based, MPI-ESM1-based, and NorESM2-based models. Specifically, improvements in GFDL-ESM4 performance are attributed to a series of updates and changes in model physics (such as mixing and climate dynamics) and biogeochemical parameterizations such as the implementation of a revised remineralization scheme for organic matter that depends on oxygen and temperature (Laufkötter et al., 2017).

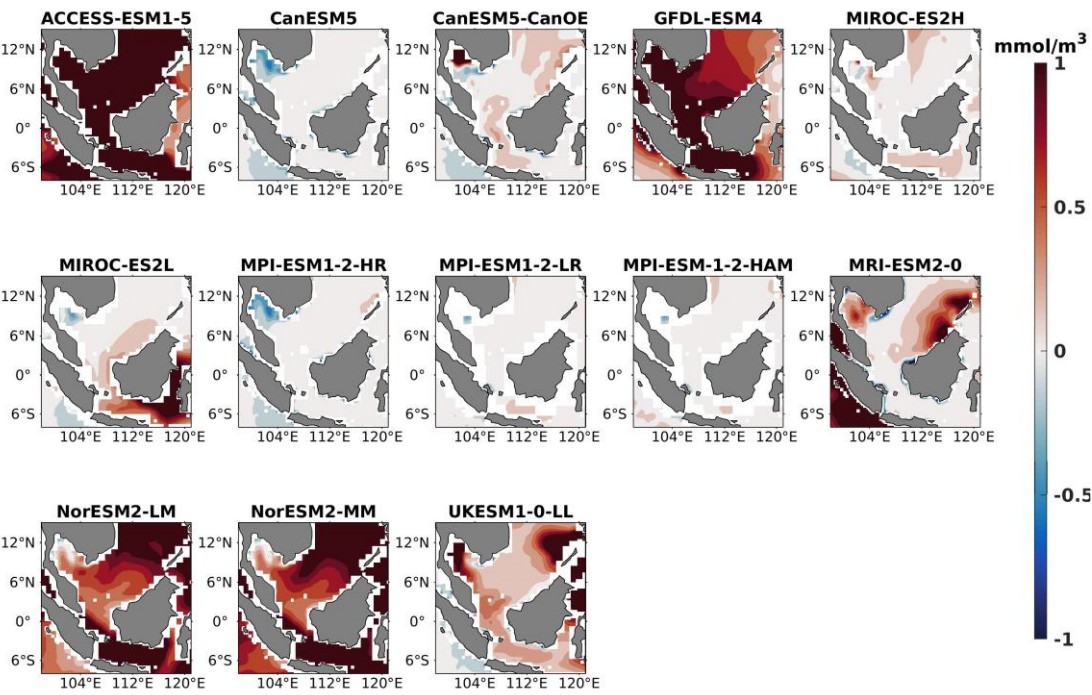

**Figure 8.** DJF spatial biases of surface nitrate for 13 individual CMIP6 ESMs relative to reference.

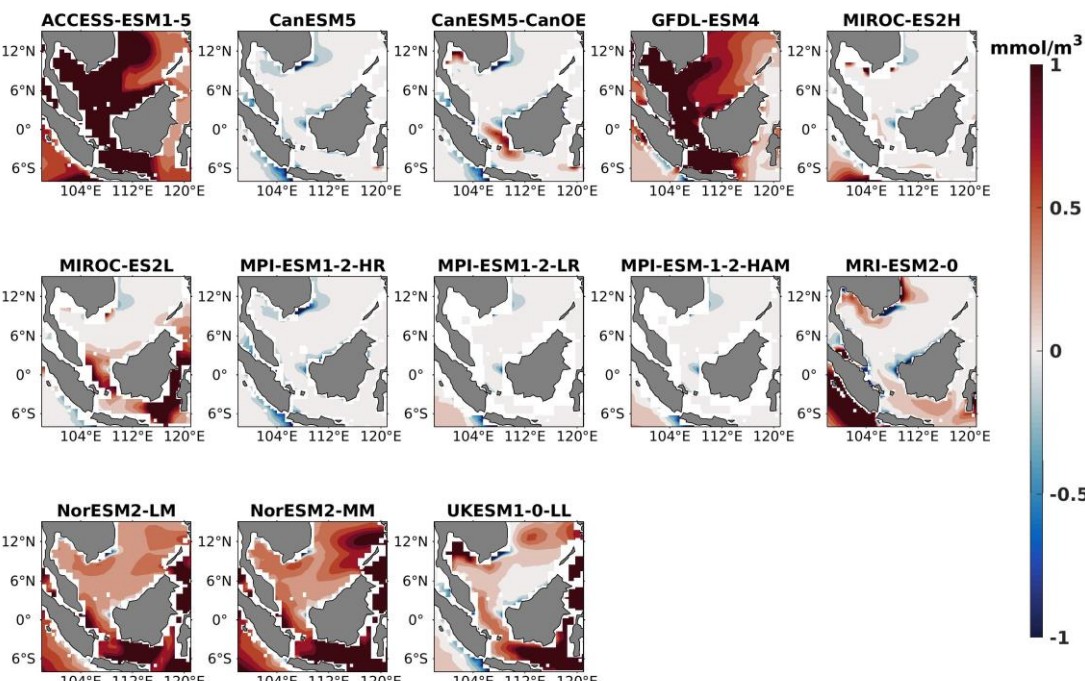


**Figure 9.** Same as Fig. 8 but for JJA.

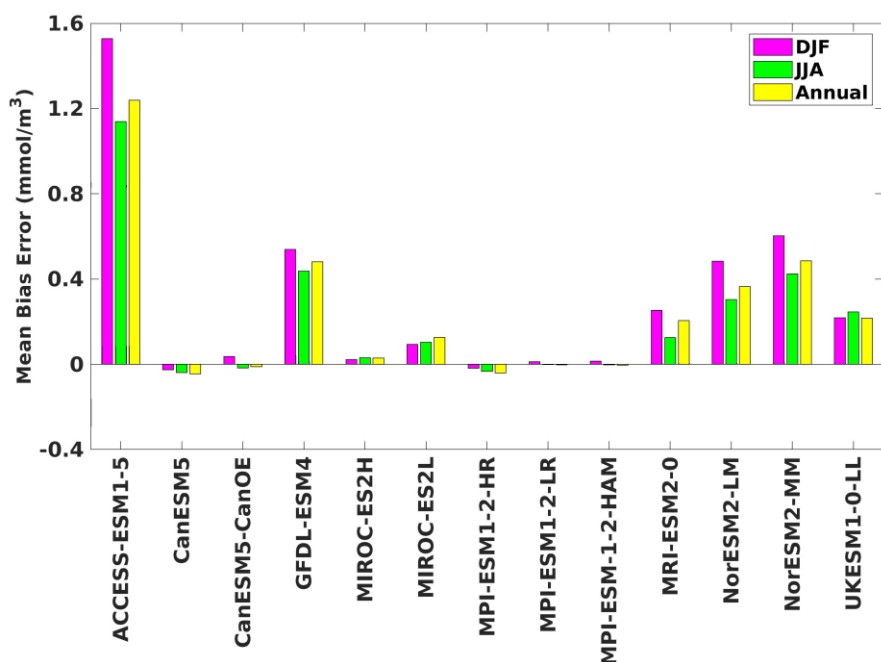

**Figure 10.** The mean bias error of surface nitrate for both seasons (DJF, JJA) and annual.

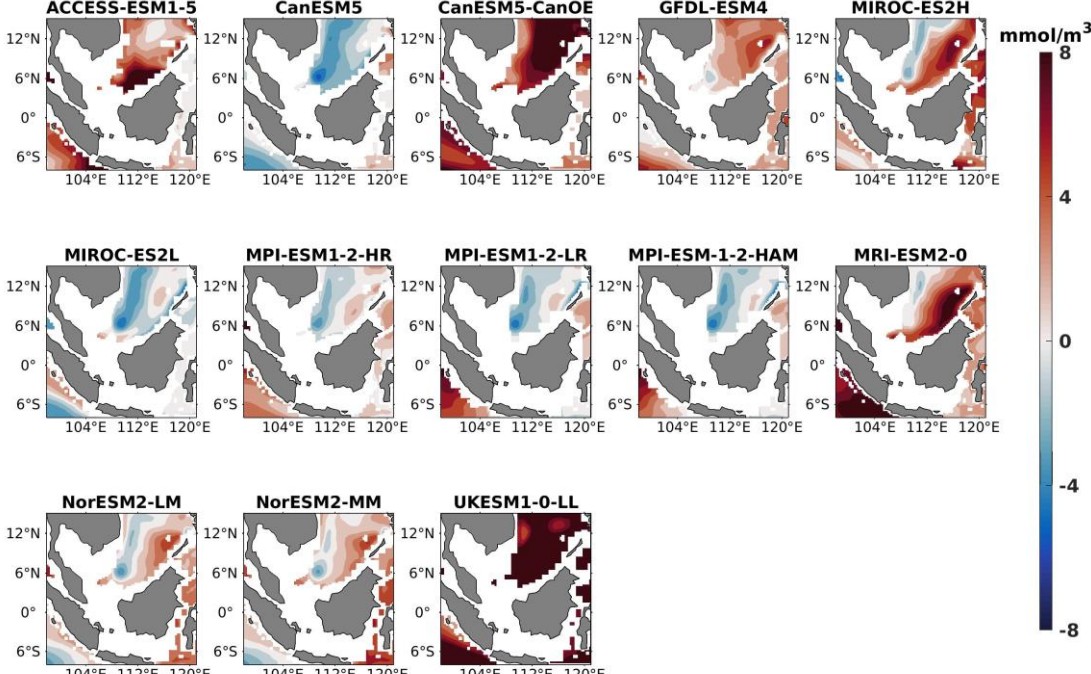

**Figure 11.** DJF spatial biases of nitrate at 70-meter depth for 13 individual CMIP6 ESMs relative to reference.

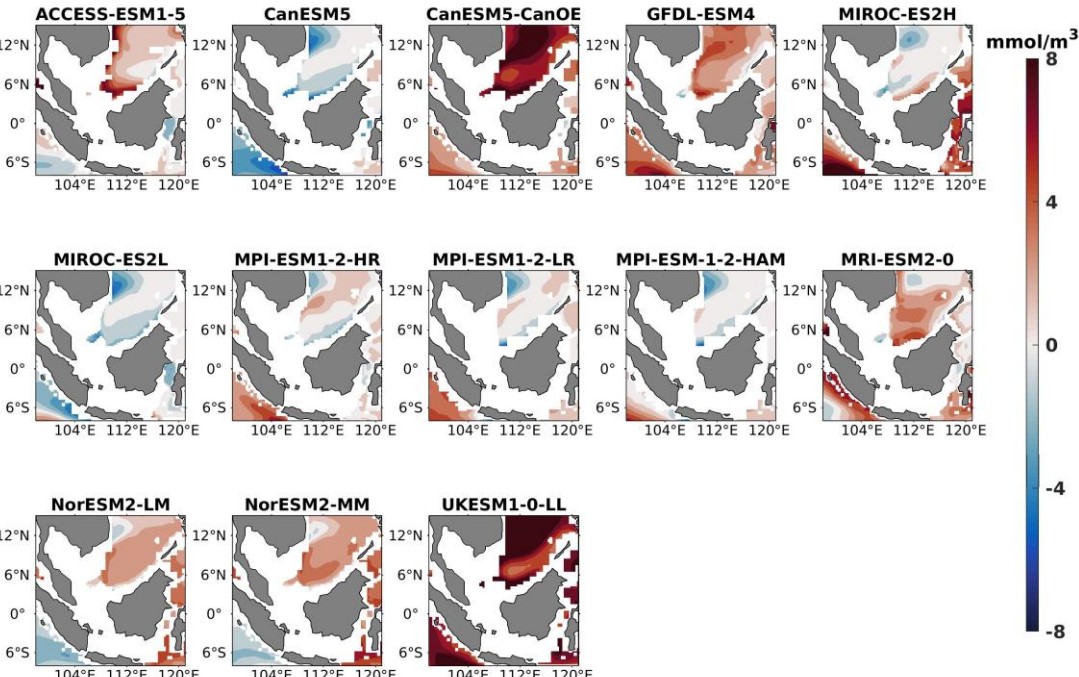

**Figure 12.** Same as Fig. 11 but for JJA.

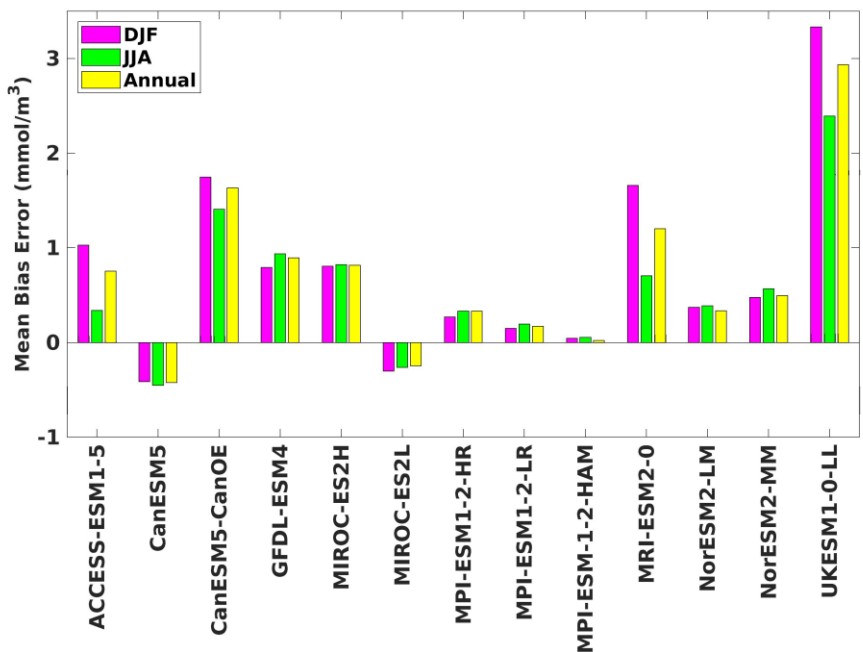

**Figure 13.** The mean bias error of nitrate at 70-meter depth for both seasons (DJF, JJA) and annual.

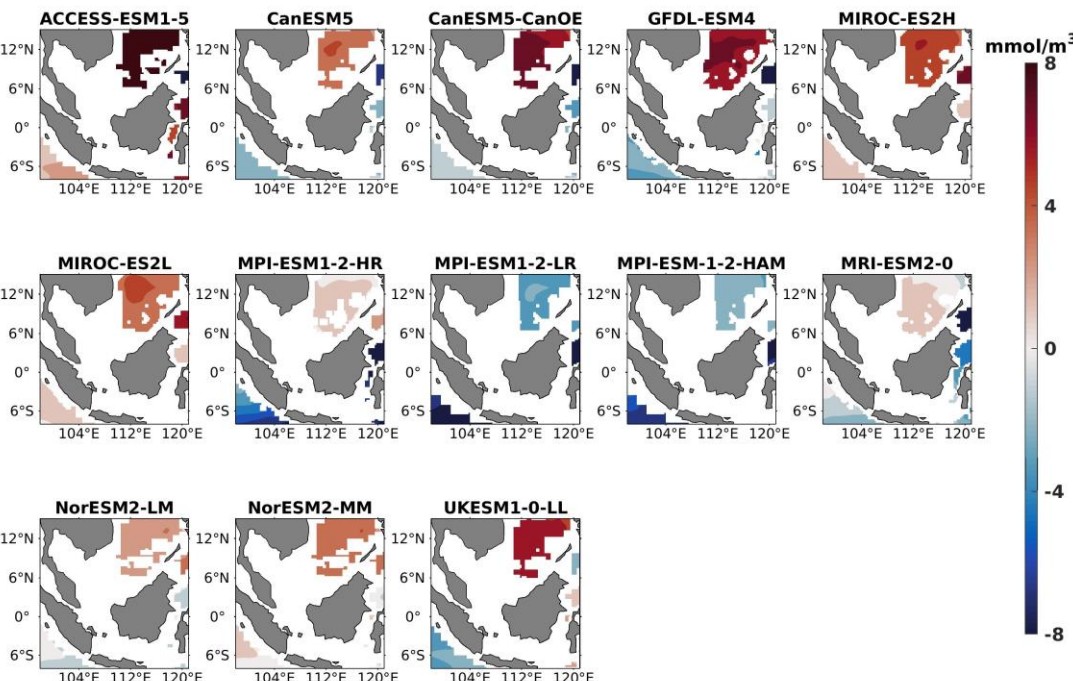

**Figure 14.** DJF spatial biases of nitrate at 1000-meter depth for 13 individual CMIP6 ESMs relative to reference.

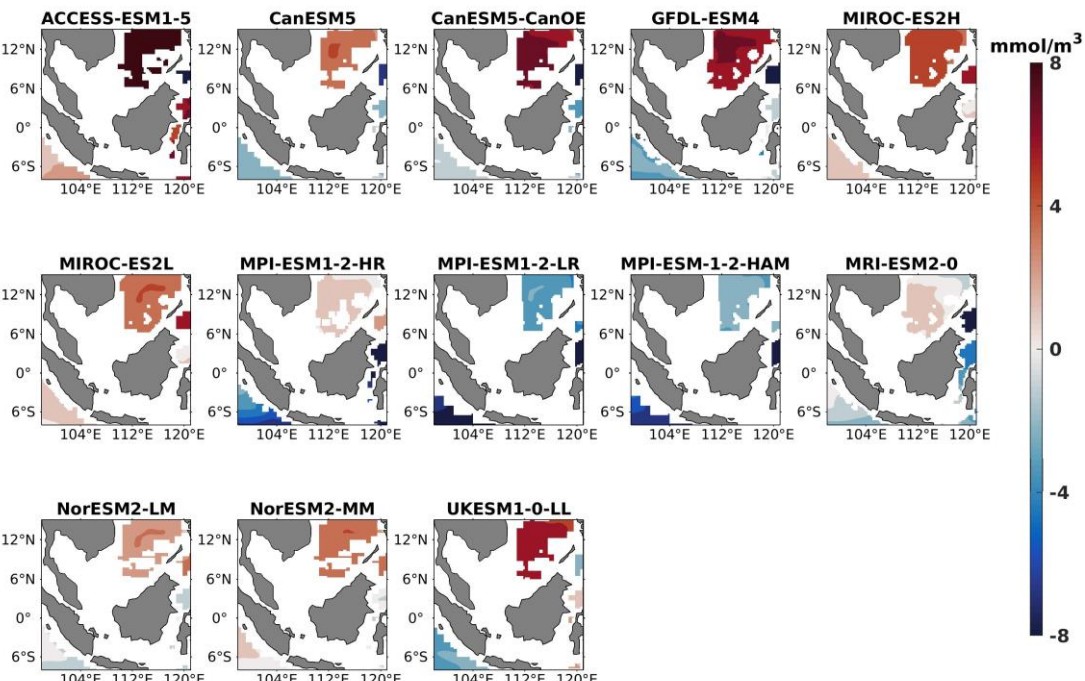

**Figure 15.** Same as Fig. 14 but for JJA.

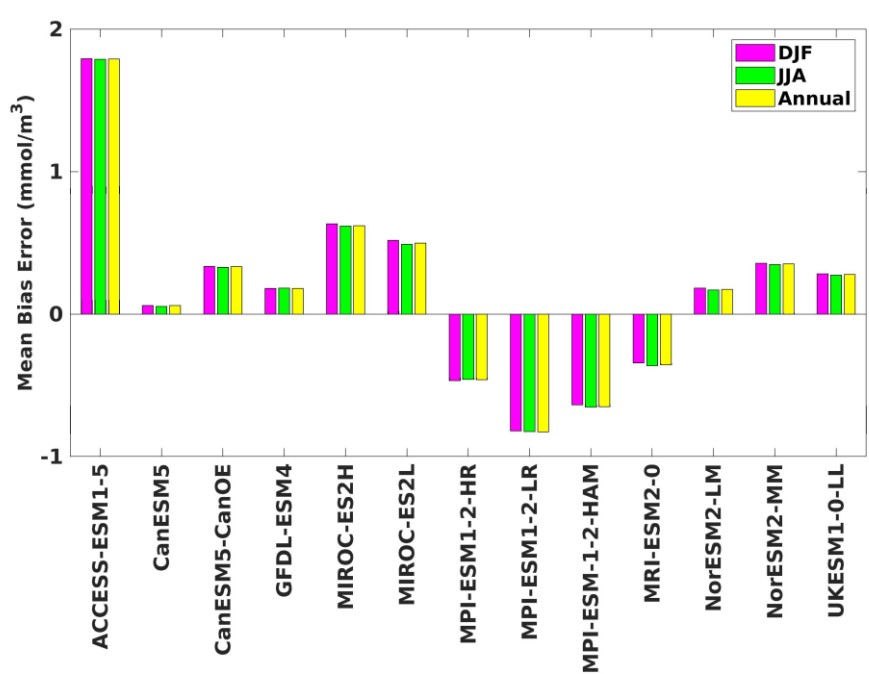


**Figure 16.** The mean bias error of nitrate at 1000-meter depth for both seasons (DJF, JJA) and annual.

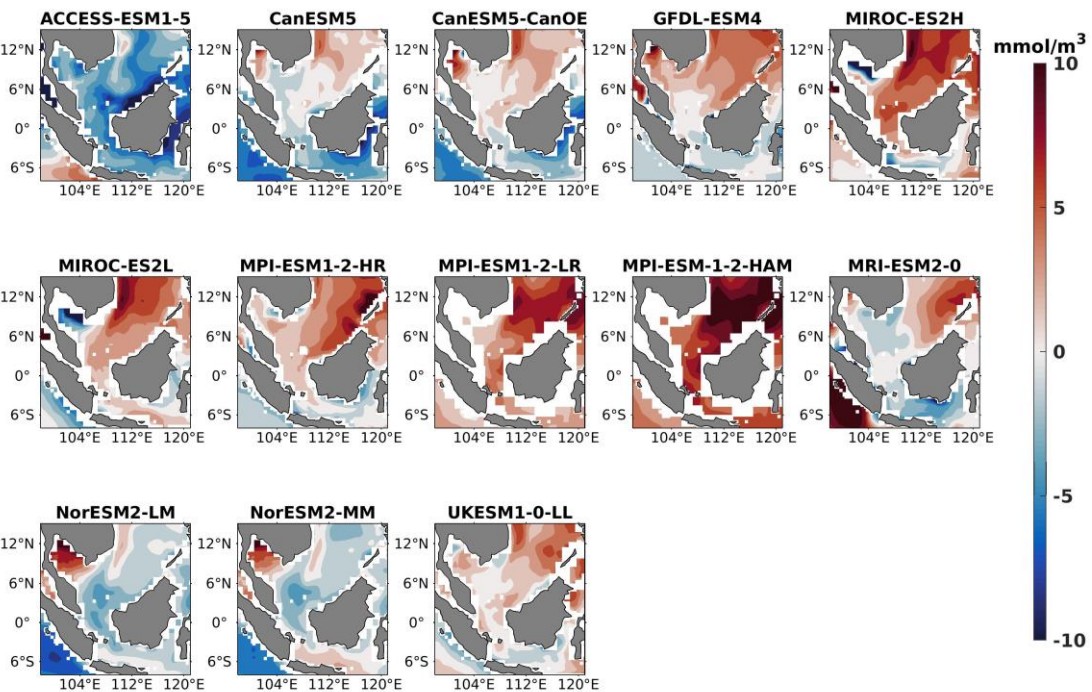

**Figure 17.** DJF spatial biases of surface oxygen for 13 individual CMIP6 ESMs relative to reference.


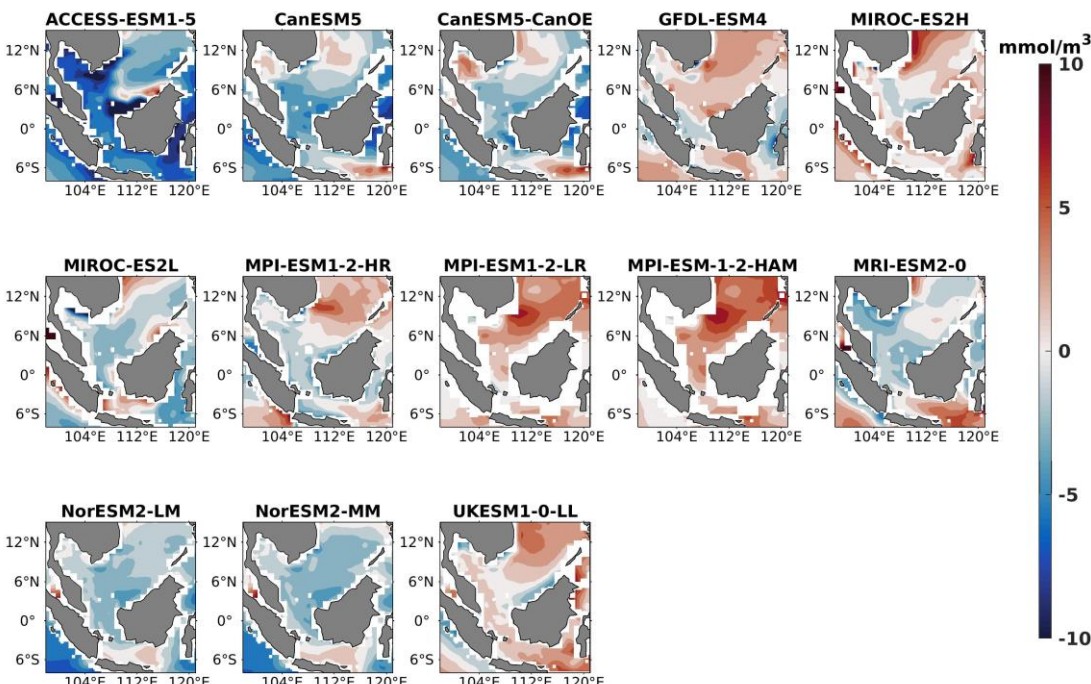

**Figure 18.** Same as Fig. 17 but for JJA.

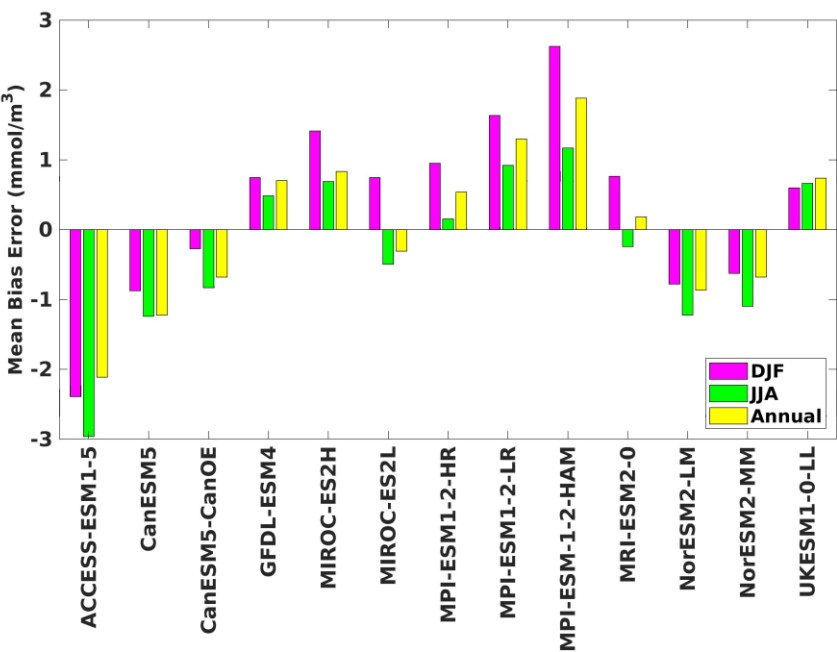

**Figure 19.** The mean bias error of surface oxygen for both seasons (DJF, JJA) and annual.

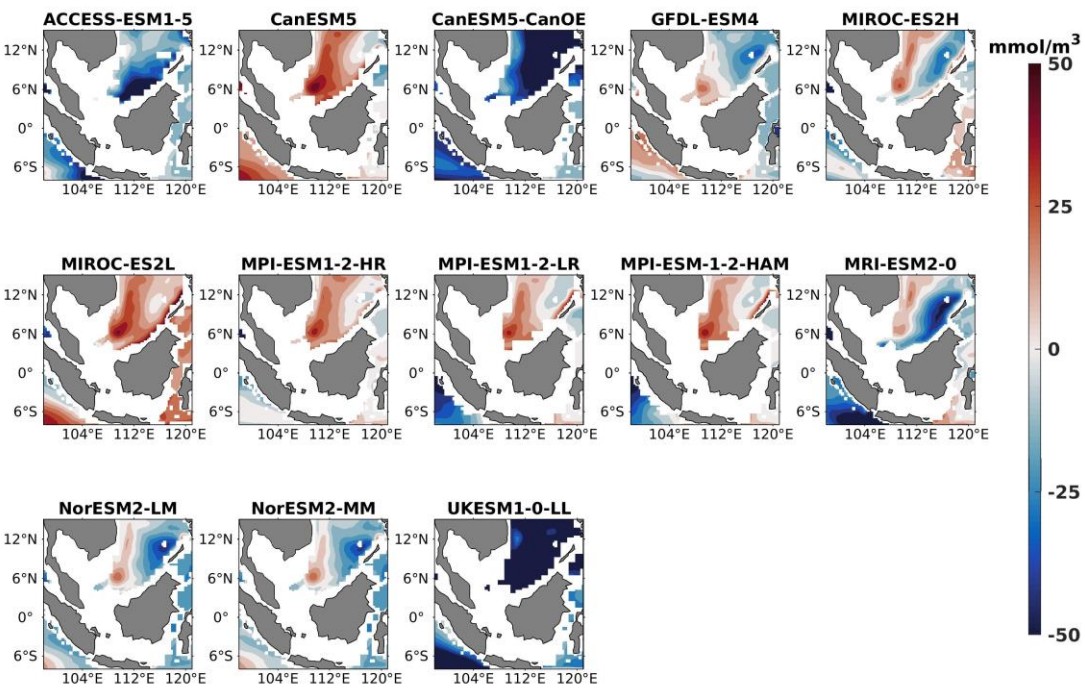

**Figure 20.** DJF spatial biases of oxygen at 70-meter depth for 13 individual CMIP6 ESMs relative to reference.

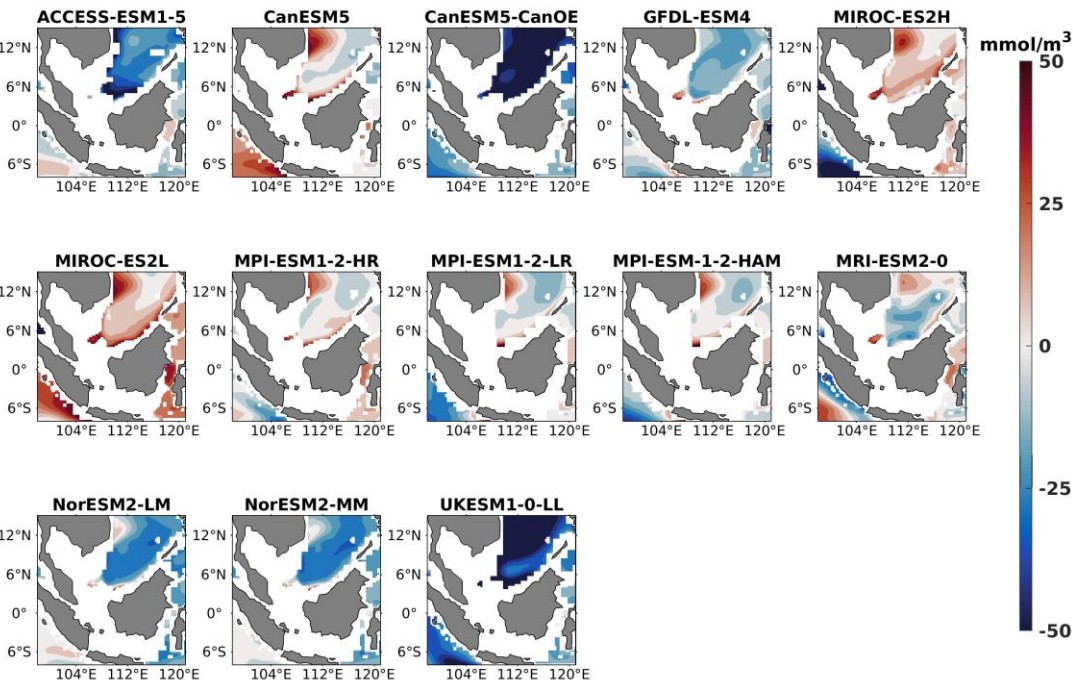


**Figure 21.** Same as Fig. 20 but for JJA.

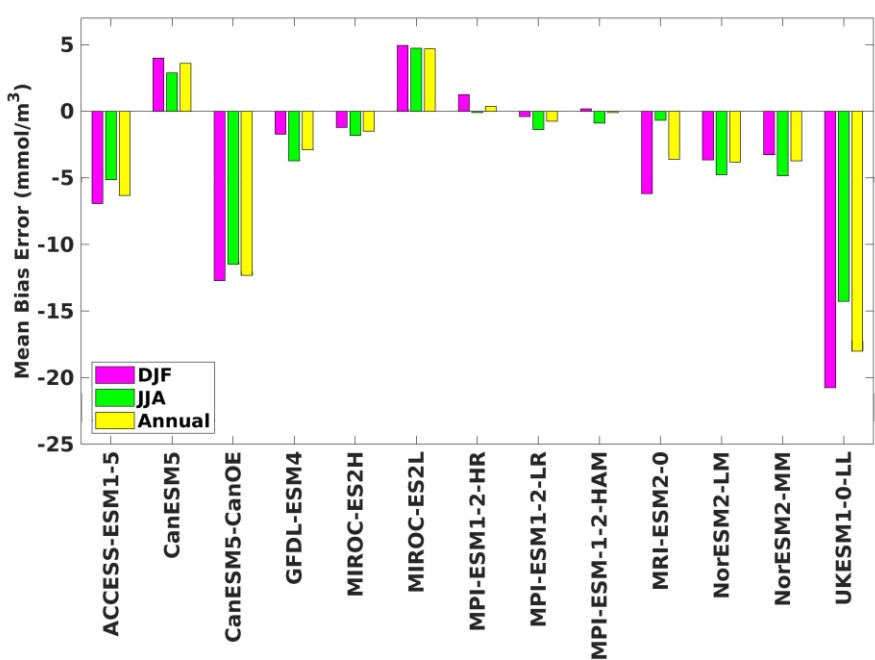

**Figure 22.** The mean bias error of oxygen at 70-meter depth for both seasons (DJF, JJA) and annual.

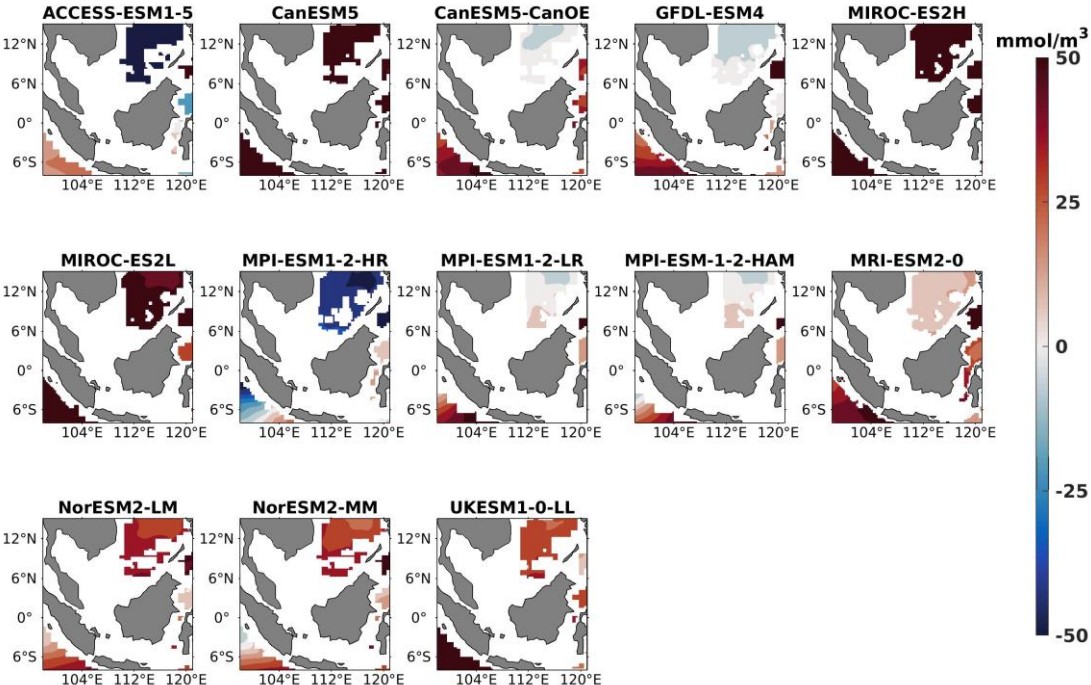

**Figure 23.** DJF spatial biases of oxygen at 1000-meter depth for 13 individual CMIP6 ESMs relative to reference.

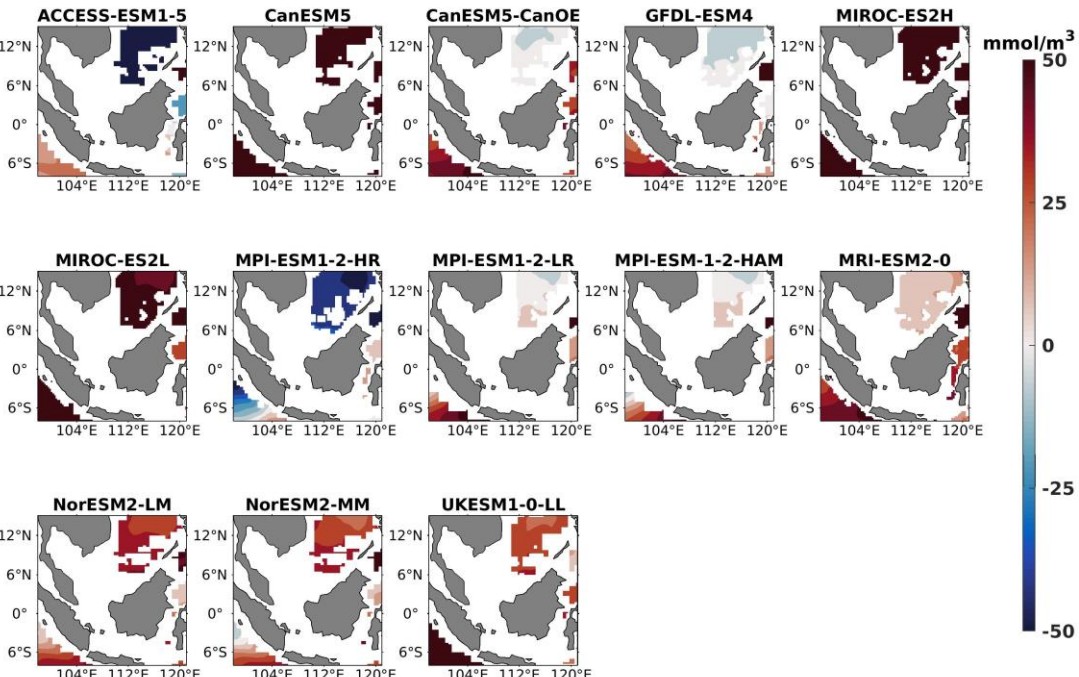

**Figure 24.** Same as Fig. 23 but for JJA.

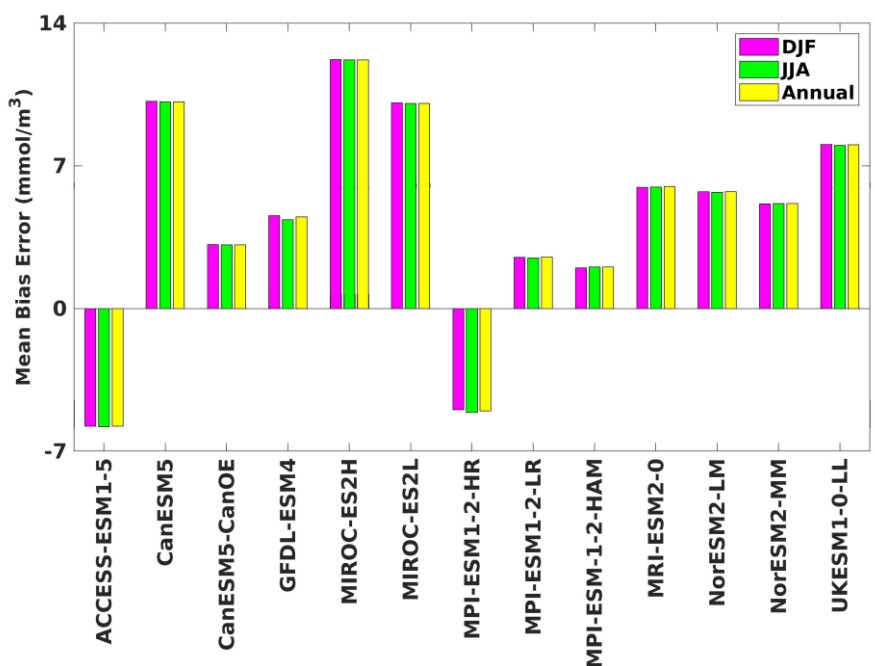

**Figure 25.** The mean bias error of oxygen at 1000-meter depth for both seasons (DJF, JJA) and annual.

The observed bias underscores the necessity of error correction, as these errors are likely to persist in the projections, potentially introducing significant uncertainty. The flow of nutrients and other biogeochemical tracers around the globe is significantly influenced by ocean circulation patterns. Alterations in these patterns can lead to shifts in nutrient distribution, consequently impacting biogeochemical processes (Lu et al., 2020). It is important to note that not all CMIP6 ESMs fail to reproduce the reference seasonal pattern of biogeochemistry, and it is unrealistic to expect a single ESM to

accurately represent all biogeochemical variables. While some ESMs can effectively reproduce the reference pattern for individual variables, there remains significant uncertainty regarding the reasons why some ESMs outperform others in this respect. The ESMs employed in CMIP6 are influenced by a variety of external factors, including solar radiation, greenhouse gas concentrations, and land-use changes. These external forcing are typically incorporated into ESMs as time-series data. Inaccuracies in these time-series data can lead to discrepancies in the simulated biogeochemistry, including errors in the

seasonal cycle (Sun and Mu, 2021). Even the most advanced ESMs do not fully capture all of the biogeochemical processes that occur in the real world. This implies that some ESMs may overlook important processes such as nutrient cycling, light availability, temperature variations and phytoplankton phenology, that contribute to the observed seasonal pattern of biogeochemistry. Additionally, the ability of an ESM to replicate the reference seasonal pattern of biogeochemistry can be influenced by the model's resolution and the specific numerical methods used to solve its equations.


## 4.2 Inter-variable Relationship

### 4.2.1 Chlorophyll – Phytoplankton

The correlation between chlorophyll and phytoplankton serves as a critical indicator of marine ecosystem health and productivity. As chlorophyll is a pigment essential for photosynthesis in phytoplankton, its concentration is often used as a

proxy for phytoplankton in aquatic environments (e.g. Petrik et al., 2022). A strong positive correlation between chlorophyll and phytoplankton signifies robust primary production and nutrient availability, highlighting favorable conditions for marine life. Conversely, a weak or negative correlation may indicate nutrient limitation, environmental stressors, or other factors affecting phytoplankton growth. Understanding this correlation provides valuable insights into ecosystem dynamics, nutrient cycling and the impacts of environmental changes on marine ecosystems. However, it's important to note that not all models

simulate chlorophyll concentrations prognostically. Instead, some models derive chlorophyll concentrations from the carbon-to-chlorophyll ratio and phytoplankton biomass. This approach acknowledges the intricate relationship between chlorophyll production and phytoplankton biomass, ensuring a comprehensive representation of primary productivity in marine environments. The correlation between chlorophyll and phytoplankton biomass can also provide insights into whether a model produces chlorophyll prognostically or not. If a model simulates chlorophyll concentration prognostically, there

should be a strong positive correlation between chlorophyll concentration and phytoplankton biomass. This correlation arises from the direct influence of phytoplankton biomass on chlorophyll production through photosynthesis. On the other hand, if chlorophyll concentrations are derived from the carbon-to-chlorophyll ratio and phytoplankton biomass, the correlation may

still exist but could be influenced by additional factors such as nutrient availability and environmental conditions. Therefore, analysing the correlation between chlorophyll and phytoplankton biomass can help discern the modeling approach used to simulate chlorophyll dynamics within the ocean model. During our analysis of the linear regression between chlorophyll and phytoplankton, most of the selected CMIP6 models showed a strong positive correlation between chlorophyll and phytoplankton concentrations **(Fig. 26)**. Notably, ACCESS-ESM1-5, MIROC-ES2L, and MRI-ESM2-0 demonstrated particularly robust positive correlations, with coefficient (R) reaching 1. However, examination of the 95% confidence interval suggests that the relationship between chlorophyll and phytoplankton in MIROC-ES2H, MPI-based models and NorESM-based models deviates from proximity. This disparity in confidence intervals among those models could arise from the differences in model parameterizations and structural complexities, resulting in differing levels of uncertainty in the simulated relationships between chlorophyll and phytoplankton. Models with simpler representations of biological processes or less accurate parameterizations may exhibit wider confidence intervals due to increased uncertainty in their outputs. Specifically, MPI-based and NorESM-based models employed the HAMOCC version of the biogeochemistry model, which does not explicitly simulate chlorophyll concentrations (Paulsen et al., 2017; Tjiputra et al., 2020).

### 4.2.2 Nitrate – Phytoplankton

Through the application of linear regression analysis, we investigated the correlation between surface nitrate levels and phytoplankton biomass. While analysing the various CMIP6 models, it was observed that none of the models displayed a correlation (R) > 0.8. However, among the selected models, CanESM5-ConOE, MIROC-ES2H, MPI-ESM1-2-LR, MRI-ESM2-0, and UKESM1-0-LL exhibited a statistically significant positive correlation (R > 0.5, p < 0.001) with surface nitrate and phytoplankton **(Fig. 27)**. Conversely, the remaining models demonstrated considerably weaker positive correlations, with only CanESM5, GFDL-ESM4, MIROC-ES2L and MPI-ESM1-2-HR displaying a slight negative correlation. This slight negative correlation could stem from various factors that it may reflect discrepancies in those model dynamics, such as the representation of nutrient uptake or phytoplankton growth rates. Biological processes within the models might not accurately capture the complexities of phytoplankton-nutrient interactions. For example, variations in biogeochemical tracers within model frameworks could influence model efficacy. Specifically, except UKESM1-0-LL and MIROC-based models, all other selected models utilize carbon as their primary model currency for representing phytoplankton biomass, incorporating explicit calculations for phytoplankton biomass and they also utilize nitrate and phosphate to constrain bulk phytoplankton growth rates alongside temperature and light. Consequently, their representation of phytoplankton biomass exhibited a weaker correlation with nitrate. Despite the use of carbon tracer, MPI-ESM1-2-LR incorporates a newly resolved nitrogen-fixing formulation within its biogeochemistry model. This updated formulation introduces an additional prognostic phytoplankton class, replacing the diagnostic formulation of nitrogen-fixation utilized in MPI-ESM-LR (Paulsen et al., 2017; Mauritsen et al., 2019). As a result, this adjustment enables the model to capture the nitrogen response to phytoplankton biomass positively. UKESM1-0-LL employed nitrogen as its primary currency, resulting in a more pronounced quantitative representation of phytoplankton biomass in response to increased nitrate levels compared to the other models **(Fig. 27)**.

While MIROC-ES2L primarily utilizes nitrogen as its tracer, it also integrates the phosphorus cycle within the model framework to accurately depict the strong phosphorus limitation on the growth of diazotrophic phytoplankton (Hajima et al., 2020). Consequently, this incorporation of the phosphorus cycle may account for phosphorus limitation, resulting in the

observed negative correlation between nitrate and phytoplankton biomass within our study area. In the case of GFDL-ESM4, the negative correlation between nitrate and phytoplankton could potentially originate from their model parametrization. In their framework, phytoplankton were categorized based on size and functional type, with small phytoplankton being nitrogen-rich and large phytoplankton phosphate-rich, thereby attributing characteristic N:P ratios (Stock et al., 2020). Thus, differences in parameterizations, data initialization and model resolution could contribute to divergent simulated responses

(Séférian et al., 2020). This discrepancy underscores the variability among model outputs and highlights the importance of further scrutinizing model performances based on their parametrization and phenological structural. It is important to analyse how these models are formulated and their roles in nutrient uptake, zooplankton grazing, phytoplankton growth, and plankton mortality within the trophic transfer processes. This approach will aim to refine our comprehension of the intricate dynamics governing marine ecosystems within each model. Furthermore, this analysis highlights significant variability in

phytoplankton-nutrient correlations across CMIP6 models, the observed discrepancies underscore the potential benefits of employing more advanced phytoplankton parameterizations, such as nutrient quota or flexible N:C ratios. These approaches could provide a more nuanced representation of phytoplankton response to nutrient availability. Models like UKESM1-0-LL, which utilize nitrogen as their primary currency for phytoplankton biomass, demonstrate good correlations with nitrate, suggesting that explicit consideration of nutrient stoichiometry may enhance model accuracy. Similarly, integrating

phosphorus cycles, as seen in MIROC-ES2L, could better capture phosphorus limitations affecting phytoplankton growth. Future model developments should prioritize these parameterizations to improve the fidelity of biogeochemical simulations and better understand ecosystem responses to environmental changes.

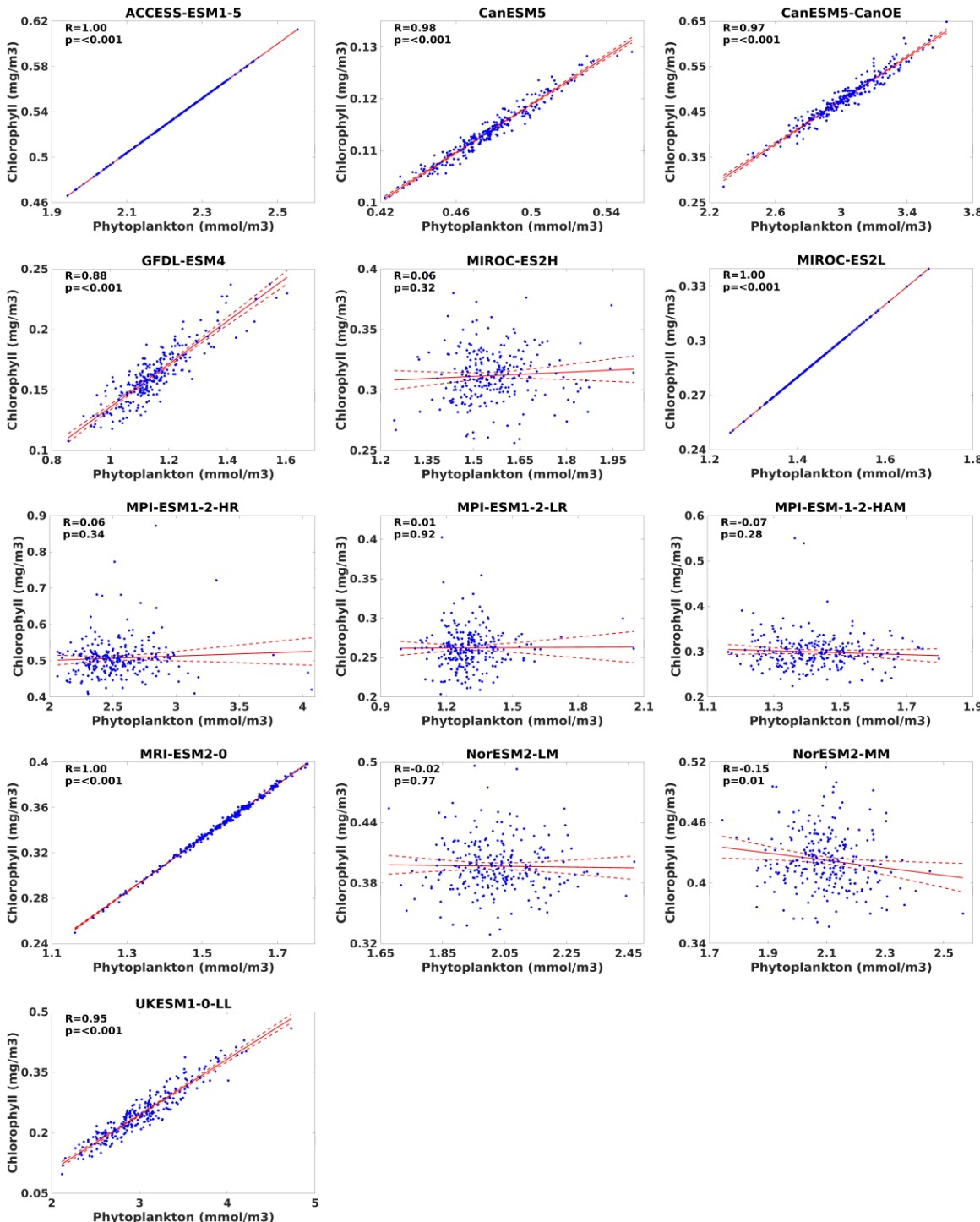

**Figure 26.** Relationships between chlorophyll and phytoplankton for 13 individual CMIP6 ESMs in southern South China Sea during the study period (1993 – 2014). Dashed red lines represent 95% confidence interval with 0.05 as the level of significance (α).

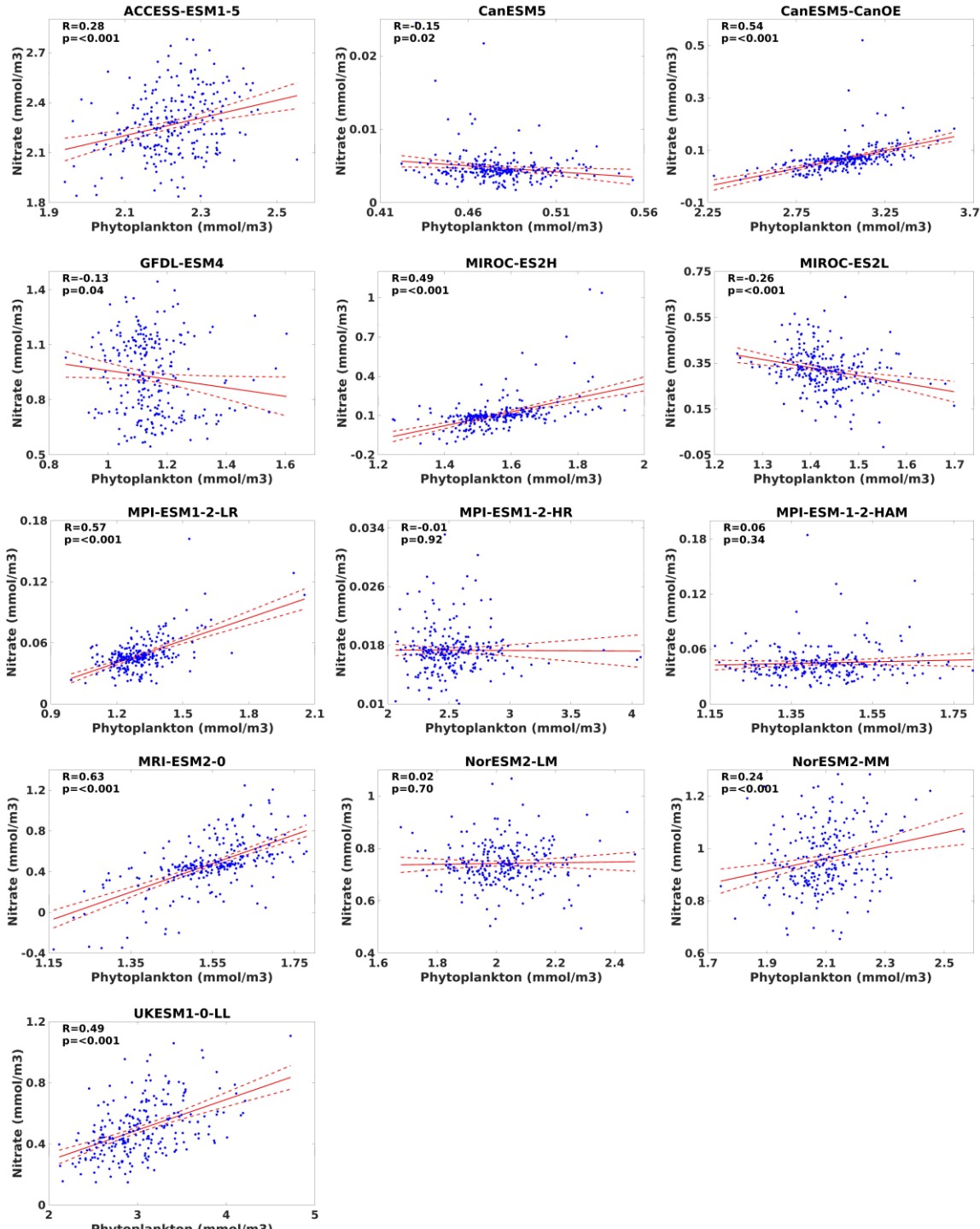

**Figure 27.** Relationships between nitrate and phytoplankton for 13 individual CMIP6 ESMs in southern South China Sea during the study period (1993 – 2014). Dashed red lines represent 95% confidence interval with 0.05 as the level of significance (α).

## 4.3 Taylor diagram

The Taylor diagram was used to evaluate and summarize the performance of CMIP6 models across various parameters they simulated. Taylor diagram provide a concise and visually appealing way to compare the performance of multiple models against reference data. Taylor diagrams facilitate the inter-comparison of multiple CMIP6 ESMs, allowing researchers to identify models that consistently perform well or poorly across a range of biogeochemical parameters by incorporate three key metrics: correlation coefficient, normalized standard deviation and normalized root mean squared difference. These metrics collectively evaluate the agreement between model simulations and reference in terms of their overall pattern, magnitude, and phase relationship by baseline observed point where the correlation is 1 and the RMSD is 0. If the simulation point is close to the reference point, it means that both the model and reference data are highly similar. If the model is far away from the reference point, they are considered as poor models and the models placed intermediate are moderate models. Taylor diagrams can be applied to assess model performance at regional and seasonal scales, providing insights into the models' ability to capture spatial and temporal variations in biogeochemical processes. This information can be valuable for understanding regional climate-biogeochemistry interactions. Taylor diagram for assessing the selected models were performed based on their annual climatology.

Regarding chlorophyll, overall performance of the 13 CMIP6 ESMs in simulating chlorophyll spatial patterns was moderate, with spatial correlations ranging from -0.2 to 0.7 and RMSDs below 2 (**Fig. 28a**). MIROC-ES2H and MIROC-ES2L were the best performing models, with correlation coefficients of 0.6 and 0.4, RMSDs of 0.7 and 0.9, and standard deviations of 0.6 and 0.5, respectively when compared to other models. ACCESS-ESM1-5 and UKESM1-0-LL had the poorest performance, with correlation coefficients of 0.1 and 0.2, RMSDs of 2.1 and 1.8, and standard deviations of 2 and 1.7, respectively. Similar to chlorophyll, the performance of the ESMs in simulating phytoplankton spatial patterns was moderate (**Fig. 28b**). MIROC-ES2H was the best performing model, with a correlation coefficient of 0.4, an RMSD of 0.9, and a standard deviation of 0.8. CanESM5 and MPI-ESM-1-2-HAM had negative correlations of -0.08 and -0.04, respectively. ACCESS-ESM1-5 and UKESM1-0-LL had correlation coefficients of 0.04 and 0.27, RMSDs of 2.3 and 2.6, and standard deviations of 2.1 and 2.7, respectively.

The performance of the ESMs in simulating nitrate concentrations was moderate to poor (**Fig. 28c**). MPI-ESM1-2-HR, MPI-ESM1-2-LR, and MPI-ESM-1-2-HAM were the best performing models, with correlation coefficients of 0.24, 0.35, and 0.39, RMSDs of 0.99, 0.95, and 0.95, and standard deviations of 0.5, 0.19, and 0.16, respectively. MIROC-ES2L, NorESM2-MM, and UKESM1-0-LL had negative correlations of -0.07, -0.02, and -0.15, respectively. ACCESS-ESM1-5 had a correlation coefficient of 0.42, an RMSD of 9.24, and a standard deviation of 9.49. It's worth noting that, due to the NSD range limit, ACCESS-ESM1-5 is not visible in the **Fig. 28c**. All ESMs simulated oxygen concentrations with positive correlations (**Fig. 28d**). MPI-ESM1-2-LR and UKESM1-0-LL were the best performing models, with correlation coefficients of 0.57 and 0.52, RMSDs of 1.29 and 1.38, and standard deviations of 1.57 and 1.6, respectively. ACCESS-ESM1-5 had the poorest performance, with a correlation coefficient of 0.58, an RMSD of 2.33, and a standard deviation of

2.75. The spatial statistics of nitrate and oxygen at deep layer (1000m) for the 13 CMIP6 ESMs is presented in
supplementary **Table S2**.

Overall, the performance of the ESMs in simulating surface biogeochemical variables ranged from moderate to
poor (**Table 2**), likely because the ESMs in CMIP6 were developed based on the broader conditions of the open ocean,
which differ significantly from the more complex and variable conditions of shelf seas. Additionally, the coarse resolution of
these models is insufficient to accurately capture the fine-scale dynamics and interactions occurring in shelf sea
environments. Together, these factors likely contribute to the moderate to poor performance of the ESMs observed in this
study. MIROC-ES2H and MIROC-ES2L were the best performing models for chlorophyll and phytoplankton, while MPI-
ESM1-2-HR, MPI-ESM1-2-LR, and MPI-ESM-1-2-HAM were the best performing models for nitrate. ACCESS-ESM1-5
generally performed poorly across all variables.

**Table 2**. Spatial statistics for surface variables of 13 CMIP6 ESMs.

| CMIP6 ESMs | chlorophyll | | | phytoplankton | | | nitrate | | | oxygen | | |
|---|---|---|---|---|---|---|---|---|---|---|---|---|
| | CC | NSD | NRMSD | CC | NSD | NRMSD | CC | NSD | NRMSD | CC | NSD | NRMSD |
| ACCESS-ESM1-5 | 0.10 | 2.01 | 2.15 | 0.04 | 2.14 | 2.33 | 0.30 | 9.49 | 9.24 | 0.58 | 2.75 | 2.33 |
| CanESM5 | -0.11 | 0.54 | 1.19 | -0.08 | 0.58 | 1.20 | 0.07 | 0.07 | 1.00 | 0.72 | 2.34 | 1.77 |
| CanESM5-CanOE | 0.40 | 2.01 | 1.85 | 0.37 | 2.39 | 2.23 | 0.20 | 1.08 | 1.32 | 0.71 | 2.35 | 1.78 |
| GFDL-ESM4 | 0.43 | 1.56 | 1.45 | 0.43 | 1.74 | 1.59 | 0.64 | 3.89 | 3.34 | 0.61 | 1.90 | 1.52 |
| MIROC-ES2H | 0.62 | 0.65 | 0.78 | 0.46 | 0.82 | 0.96 | 0.45 | 1.38 | 1.29 | 0.50 | 2.23 | 1.94 |
| MIROC-ES2L | 0.40 | 0.60 | 0.93 | 0.22 | 0.76 | 1.12 | -0.07 | 2.23 | 2.51 | 0.42 | 2.40 | 2.18 |
| MPI-ESM1-2-HR | 0.07 | 1.20 | 1.51 | 0.01 | 1.27 | 1.61 | 0.24 | 0.05 | 0.99 | 0.59 | 2.01 | 1.63 |
| MPI-ESM1-2-LR | 0.13 | 0.95 | 1.29 | 0.02 | 1.19 | 1.54 | 0.35 | 0.19 | 0.95 | 0.57 | 1.57 | 1.29 |
| MPI-ESM-1-2-HAM | 0.04 | 1.22 | 1.54 | -0.04 | 1.48 | 1.82 | 0.39 | 0.16 | 0.95 | 0.59 | 1.80 | 1.45 |
| MRI-ESM2-0 | 0.29 | 0.89 | 1.13 | 0.20 | 1.10 | 1.33 | 0.18 | 4.18 | 4.12 | 0.30 | 1.80 | 1.78 |
| NorESM2-LM | 0.15 | 0.73 | 1.15 | 0.22 | 0.93 | 1.21 | 0.07 | 2.91 | 3.01 | 0.58 | 2.34 | 1.93 |
| NorESM2-MM | 0.13 | 0.84 | 1.22 | 0.18 | 1.05 | 1.32 | -0.02 | 3.37 | 3.53 | 0.54 | 1.95 | 1.65 |
| UKESM1-0-LL | 0.20 | 1.73 | 1.81 | 0.28 | 2.77 | 2.67 | -0.15 | 2.81 | 3.12 | 0.52 | 1.60 | 1.38 |

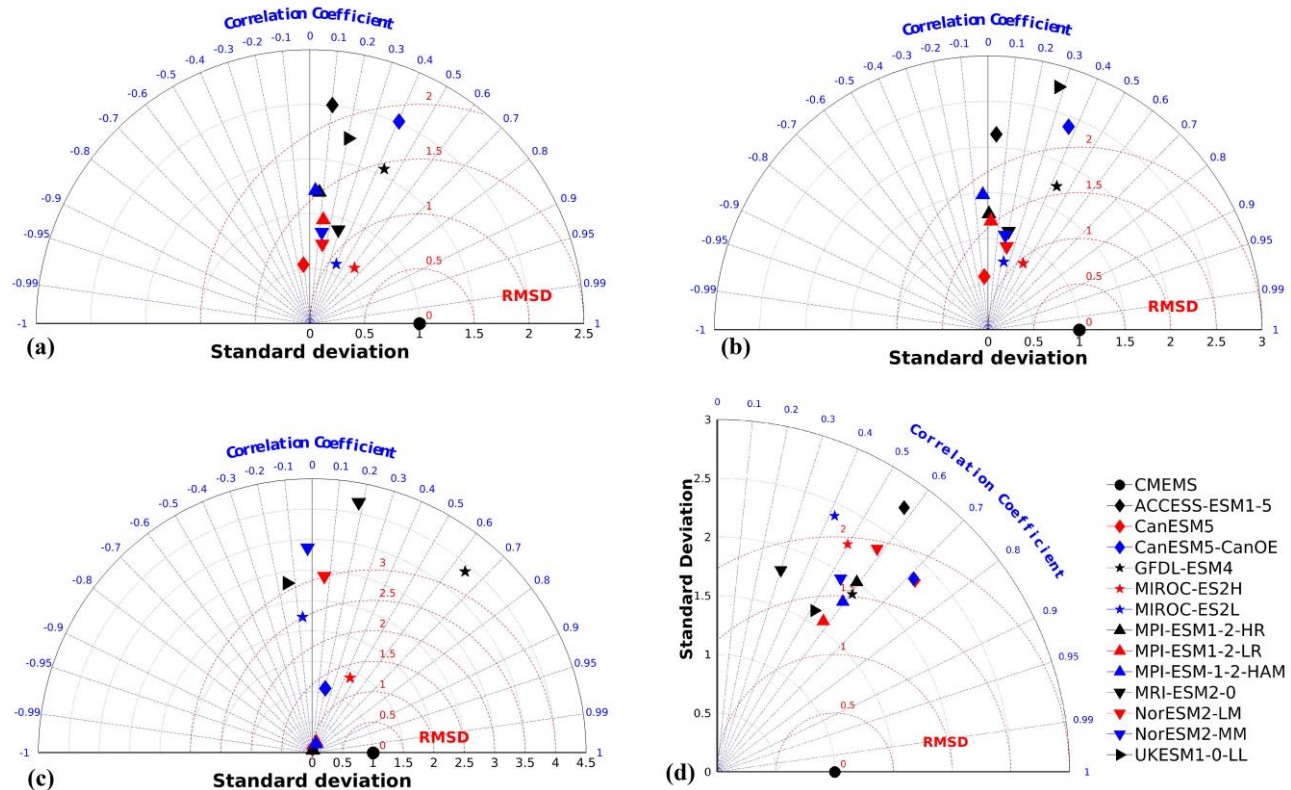

**Figure 28.** Annual Taylor Diagram for (a) chlorophyll, (b) phytoplankton, (c) nitrate and (d) oxygen.

**4.4 Model ranking**

In addition to the qualitative analysis presented in the Taylor diagram, a skill score is calculated using Equation (5) to further validate the models' proficiency in reproducing biogeochemical variables, serving as a quantitative summary of the information conveyed by the Taylor diagram. The final ranking of models is determined based on this skill score. The ranking process involves assessing the skill score for each model with respect to individual variables. Subsequently, the average of the individual variable scores for each model is computed, serving as the overall score for that particular model. The final and variable scores for the models are depicted in the **Fig. 29**. A Taylor skill score closer to 1 indicates a higher agreement between the simulation and reference. This approach, akin to previous successful studies (Kim et al., 2023; Yool et al., 2021), enhances the robustness of the assessment.

The overall performance of the selected CMIP6 ESMs is summarized in the final score graph **(Fig. 29a)**. MIROC-ES2H emerged as the top-ranked ESM, followed by GFDL-ESM4 and CanESM5-CanOE in second and third place, respectively. Beyond the top three, the final score curve exhibits a nearly linear pattern, suggesting that the remaining ESMs exhibited relatively consistent performance across various evaluation criteria. Notably, ACCESS-ESM1-5 ranked lowest

among the ESMs, consistent with its observed performance in spatial bias and Taylor diagram analysis. MIROC-ES2H demonstrated superior performance across all variables except oxygen, consistently achieving scores above 0.2. For oxygen,

MPI-ESM1-2-LR ranked highest due to its exceptional spatial representation and accurate seasonal pattern captured in the Taylor diagram. This outperformance of MPI-based models for oxygen can be attributed to the effectiveness of the physical drivers within that model. Jin et al. (2023) showed that MPI-based models excel in simulating climatological sea surface temperature (SST) during boreal winter and summer, and are among the top performers in reproducing SST climatology in Asian marginal seas due to their minimal SST biases. Thus, MPI-based models outperform others in replicating surface

oxygen variables in our study. The distribution of scores for chlorophyll, phytoplankton, oxygen and nitrate (**Fig. 29b-e**) indicates that only a few ESMs consistently achieved top performance for these variables. Since an ensemble always masks the significant differences between the individual models, the Multi-Model Ensemble comprising of all ESMs does not take into account the relative strengths and weaknesses of each model (Bannister et al., 2017; Knutti, 2010). Therefore, evaluating GCMs in order to choose the best or most appropriate ones is crucial and helps planners and policymakers feel confident in

their use of GCMs for impact assessment studies and other uses (Aloysius et al., 2016; Perez et al., 2014; Raju and Kumar, 2020).

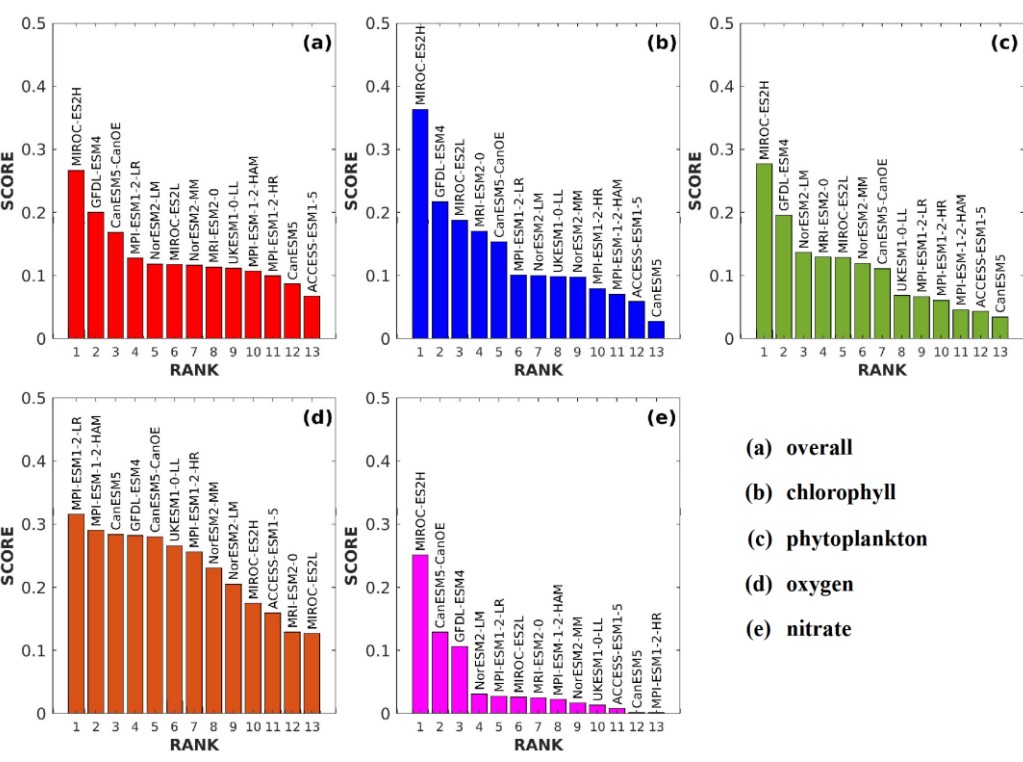

**Figure 29.** Selected CMIP6 ESMs' scores and ranks.


**5 CONCLUSIONS**

This study assessed the ability of 13 CMIP6 ESMs to replicate biogeochemical variables such as chlorophyll, phytoplankton, nitrate and oxygen over the southern SCS at annual and seasonal scales. The ESMs were compared against CMEMS data, considered a proxy for reference for the period 1993-2014. Their performance was evaluated based on their ability to
reproduce the seasonal climatology and distribution bias using statistical metrics such as the Taylor diagram and Taylor skill score. The models were ranked based on their skill score. The results revealed that some models slightly overestimated or underestimated biogeochemical variables during both seasons. The performance of the models varied between seasonal and annual scales. Most of the models exhibited a positive spatial correlation with CMEMS at both seasonal and annual scales for surface variables. However, a few models showed negative correlations, such as CanESM5 (-0.11 and -0.08 for
chlorophyll and phytoplankton, respectively), MPI-ESM-1-2-HAM (-0.04 for phytoplankton), MIROC-ES2L (-0.07 for nitrate), NorESM2-MM (-0.02 for nitrate), and UKESM1-0-LL (-0.15 for nitrate). Similarly, at the depth of 1000m, GFDL-ESM4 and MRI-ESM2-0 models alone shows positive correlation of 0.02 and 0.46, respectively and the remaining models showed negative correlation ranging -0.77 to -0.08. At the depth of 1000m for oxygen, ACCESS-ESM1-5, GFDL-ESM4 and UKESM1-0-LL alone showed negative correlation of -0.2, -0.26 and -0.06, respectively and the remaining models showed
positive correlation ranging 0.05 to 0.6. Despite their overall good performance in the ranking, some models were unable to accurately simulate the seasonal climatology of the study region. While both Taylor diagram and Taylor skill score methods aim to assess model performance, discrepancies in model ranking can arise due to their different approaches. The Taylor diagram emphasizes a balanced assessment of correlation, RMSE, and SD, providing a visual representation of model performance and allowing for qualitative assessment and identification of strengths and weaknesses. In contrast, the Taylor
skill score places more emphasis on RMSE and SD, as their normalized differences contribute more significantly to the numerical score, which may not capture the nuances of model performance observed in the Taylor diagram. Considering all these factors, the top five overall best-performing models for biogeochemical variables in southern SCS were MIROC-ES2H, GFDL-ESM4, CanESM5-CanOE, MPI-ESM1-2-LR, and NorESM2-LM. The conclusions drawn from this research hold significant implications for both researchers and policy-makers relying on the datasets. The outcomes offer valuable
insights that can guide improvements in parameterization schemes within models, particularly in instances where the reference patterns were not effectively reproduced. Addressing the existing challenges related to topography and local-scale convective effects remains a priority for ongoing model enhancements. Furthermore, analysing the physical drivers or control variables of models, such as temperature, precipitation and wind, holds immense importance in advancing our understanding of model performance and inter-model process parameterization differences. By conducting such studies, we
can gain important insights into the mechanisms governing model behaviour and the factors influencing their outcomes. This deeper understanding enables us to refine model accuracy, enhance predictive capabilities and ultimately improve our ability to simulate and predict biogeochemical processes in various environmental systems. Consequently, by comprehensively

investigating these aspects, we can identify areas for model improvement and develop more robust frameworks for studying complex ecological dynamics.

## DATA AVAILABILITY

The CMIP6 data are available from the Earth System Grid Federation (ESGF) https://esgf-node.llnl.gov/search/cmip6/. The CMEMS data can be accessed at https://data.marine.copernicus.eu/product/GLOBAL_MULTIYEAR_BGC_001_029/description. The WOA18 data can be accessed at https://www.ncei.noaa.gov/access/world-ocean-atlas-2018. The GlobColour data can be accessed at https://data.marine.copernicus.eu/product/OCEANCOLOUR_GLO_BGC_L4_NRT_009_102/services.

## AUTHOR CONTRIBUTIONS

The work, methodologies and formal analysis were conceptualized by W.M and J.X.C. The manuscript was authored by W.M, with input from all co-authors. Part of the discussion was contributed by N.H.R and R.M.A. Work supervision was conducted by M.F.A. All authors participated in contributing to the article and endorsed the submitted version.

## CONFLICT OF INTEREST

The authors declare that the research was conducted in the absence of any commercial or financial relationships that could be construed as a potential conflict of interest.

## FUNDING

This research was funded by Ministry of Higher Education (MoHE), Malaysia research grant under the Long Term Research Grant (LRGS) Scheme. Grant number: LRGS/1/2020/UMT/01/1/2.

## ACKNOWLEDGMENTS

The authors would like to acknowledge Long-Term Research Grant (LRGS), provided from Ministry of Higher Education (MoHE), Malaysia (LRGS/1/2020/UMT/01/1/2). We also acknowledge CMEMS (Copernicus Marine Environment Monitoring Service) for freely providing biogeochemical products and the World Climate Research Programme, which, through its Working Group on Coupled Modelling, coordinated and promoted CMIP6. We thank the climate modeling groups for producing and making available their model output, the Earth System Grid Federation (ESGF) for archiving the data and providing access, and the multiple funding agencies who support CMIP6 and ESGF.

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
