# Peer review of "Evaluation of CMIP6 Models Performance in Simulating Historical Biogeochemistry across Southern South China Sea"

_EGUsphere, 2024_

## Referee Comment (RC2)

The paper by Marshal et al. evaluates how well 13 CMIP6 ESMs simulate biogeochemical variables in the southern South China Sea (SCS) by comparing their simulation of chlorophyll, phytoplankton biomass, nitrate, and oxygen with CMEMS data. The authors use statistical metrics, such as correlation coefficient, RMSD, and mean bias error, to evaluate and rank which model can represent CMEMS data at southern SCS the best. They found that although most ESMs capture the observed seasonal pattern of each biogeochemical property, some models exhibit overestimation or underestimation of their magnitude. The paper provides an important evaluation of the CMIP6 ESMs in the southern SCS, with a thorough assessment of phytoplankton, nutrient, and oxygen -related biogeochemical properties. However, I have some issues with the choice of the reference dataset and the structure of the manuscript.

General comments:

The aim of the paper is to rank 13 CMIP6 ESM simulations based on their ability to reproduce selected observed biogeochemical variables. However, the dataset that the author chose is not strictly observations. Based on the link they provided in line 123, the CMEMS ocean biogeochemistry product is based on the PISCES model output (although it is forced with reanalysis product). I also noticed that among the 13 CMIP6 ESMs, the authors have not chosen IPSL-CM6A ESM, which includes PISCES as its ocean biogeochemical model. I understand that in-situ observations may be rare in this region, but to truly assess the CMIP6 ensemble and individual models, I suggest the authors could compare the CMIP6 models with satellite-derived chlorophyll-a and primary production, as well as the World Ocean Atlas product for nitrate and oxygen. Since the paper also looks at the seasonal trend of biogeochemical properties, it could benefit from exploring whether different CMIP6 models can capture phytoplankton phenology (e.g., Racault et al., 2015; Gittings et al., 2018), which is an important indicator.

Indeed, most of the biological activity occurs near the surface layers of the ocean, but it's important to consider the biogeochemical dynamics near the seabed, particularly in shelf seas, as they can have complex structures through interactions of ocean physics with biological processes, such as export and remineralization. I would appreciate the inclusion of depth profiles and benthic concentrations of oxygen and nitrate – this would provide a more thorough assessment of the biogeochemical properties. Furthermore, most of the biogeochemical models used in CMIP6 are not specifically built for shelf seas. It would be interesting to see whether these models can represent nutrient and oxygen distribution at shallower depths.

Although the authors put a great effort in evaluating CMIP6 model outputs, the model structures could also be evaluated; how biogeochemical tracers are represented, and whether these representations affect the performance of the model in the southern SCS. Perhaps the authors can add another table which biogeochemical tracers these

models represent (e.g., in MEDUSA-2 (UKESM), it does not represent diazotrophic phytoplankton, explicitly calculates phytoplankton chlorophyll, and uses N as model currency, while in OECO-2 (MIROC), it has diazotrophic phytoplankton with C as model currency and includes Phosphate as nutrients), and perhaps also how they are formulated, especially when it involves trophic transfer (e.g. nutrient uptake, zooplankton grazing, and phytoplankton growth, and plankton mortality). These additions can add some discussion on how model representation (and structure) may affect model performance in the shelf seas, instead of repeatedly saying that underestimation/overestimation is due to zooplankton grazing/phytoplankton productivity/nutrient uptake.

The presentation of the results can also be improved. I think it will be easier to follow the results if the authors describe the observed distribution of nitrate, chlorophyll, phytoplankton biomass, and oxygen, then compare them with the model. For the figures, it would be more interesting to see the difference between the CMEMS data and CMIP6 outputs with better figure resolution (especially figure 6). Additional discussion on regions where bias usually occurs in different models will also be interesting (e.g., the shelf seas between Sumatra, the Malaysian peninsula, and Borneo are always high in phytoplankton biomass for UKESM, CanESM5, ACCESS, MPI-ESM1-2, NorESM2).

Specific comments:

L12 – perhaps the authors can add a % or number on the degrees of overestimation and underestimation.

L22-23 - Based on CMIP6 models, NPP trend is uncertain, apart maybe at the Southern Ocean (Tagliabue et al., 2021)

L33-34 - This is not always the case - OBGC models can give the seemingly good representation of historical climate pattern but for the wrong reason. Furthermore, OBGC model results is dependent on its physical forcings (see Sinha et al., 2010)

L60 – typo: Tjiputra et **al**, (2020)

L61-72 – I'm not so sure if these are appropriate examples. Maybe add studies like Kwiatkowski et al., 2020, Hinrichs et al., 2023

L83-L85 – Why only phytoplankton, chlorophyll, nitrogen, and oxygen? Why not net primary production and or carbon?

L103-L105 - This sounds like phytoplankton is controlling the physical biogeochemical process?

L122-123 – is this the hindcast global ocean biogeochemistry? Do you also use the GlobColour for chlorophyll? Please be more specific.

L125 – Perhaps, instead of having 2/3 ESMs with the same OBGC model, maybe choose one of them instead, so you can also look at other models such as PISCESv2 (Aumont et al., 2016), MARBL (Long et al., 2021), BFM5.2 (Lovato et al., 2022)?

L132 – Do you mean visualised using taylor diagram? How do you calculate model/data comparison using a diagram?

L164-165 Can you provide a reference on this statement?

L172 – but CMEMS data is not really observation, isn't it?

L177-L179 - perhaps spell out how these models represent their phytoplankton growth and chlorophyll concentration? And compare it to models that have better RMSD?

L184, 219 – what is acceptable range?

L186 - why is UKESM not overestimating chlorophyll, but overestimates phytoplankton carbon?

L232-L242 – maybe move this to the study domain part instead of on the results section?

L251 – Can you give example of the important processes?

L286 - Why do you think this is? could it be that the ESMs in CMIP6 is developed based on the condition of the open ocean, but not the shelf seas? Or is it because the resolution is too coarse for shelf seas?

Figure 6 could do with higher resolution.

References:

Racault, M.-F., Raitsos, D.E., Berumen, M.L., Brewin, R.J.W., Platt, T., Sathyendranath, S., Hoteit, I., 2015. Phytoplankton phenology indices in coral reef ecosystems: Application to ocean-color observations in the Red Sea. Remote Sensing of Environment 160, 222–234. https://doi.org/10.1016/j.rse.2015.01.019

Gittings, J.A., Raitsos, D.E., Kheireddine, M. *et al*. Evaluating tropical phytoplankton phenology metrics using contemporary tools. *Sci Rep* **9**, 674 (2019). https://doi.org/10.1038/s41598-018-37370-4

Tagliabue, A., Kwiatkowski, L., Bopp, L., Butenschön, M., Cheung, W., Lengaigne, M., Vialard, J., 2021. Persistent Uncertainties in Ocean Net Primary Production Climate Change Projections at Regional Scales Raise Challenges for Assessing Impacts on Ecosystem Services. Frontiers in Climate 3. https://doi.org/10.3389/fclim.2021.738224

Sinha, B., Buitenhuis, E.T., Quéré, C.L., Anderson, T.R., 2010. Comparison of the emergent behavior of a complex ecosystem model in two ocean general circulation models. Progress in Oceanography 84, 204–224. https://doi.org/10.1016/j.pocean.2009.10.003

Kwiatkowski, L., Torres, O., Bopp, L., Aumont, O., Chamberlain, M., Christian, J., Dunne, J., Gehlen, M., Ilyina, T., John, J., Lenton, A., Li, H., Lovenduski, N., Orr, J., Palmieri, J., Schwinger, J., Séférian, R., Stock, C., Tagliabue, A., Takano, Y., Tjiputra, J., Toyama, K., Tsujino, H., Watanabe, M., Yamamoto, A., Yool, A., Ziehn, T., 2020. Twenty-first century ocean warming, acidification, deoxygenation, and upper ocean nutrient decline from CMIP6 model projections. Biogeosciences Discussions 1–43. https://doi.org/10.5194/bg-2020-16

Hinrichs, C., Köhler, P., Völker, C., and Hauck, J.: Alkalinity biases in CMIP6 Earth system models and implications for simulated $CO_2$ drawdown via artificial alkalinity enhancement, Biogeosciences, 20, 3717–3735, https://doi.org/10.5194/bg-20-3717-2023, 2023.

Aumont, O., Ethé, C., Tagliabue, A., Bopp, L. & Gehlen, M. PISCES-v2: an ocean biogeochemical model for carbon and ecosystem studies. Geosci. Model Dev. 8, 2465–2513 (2015).

Lovato, T., Peano, D., Butenschön, M., Materia, S., Iovino, D., Scoccimarro, E., et al. (2022). CMIP6 simulations with the CMCC Earth System Model (CMCC-ESM2). Journal of Advances in Modeling Earth Systems, 14, e2021MS002814. https://doi.org/10.1029/2021MS002814

Long, M. C. et al. Simulations with the marine biogeochemistry library (MARBL). J. Adv. Model. Earth Syst. 13, e2021MS002647 (2021).

---

## Author Comment (AC1)

Response to review of **"Evaluation of CMIP6 Models Performance in Simulating Historical Biogeochemistry across Southern South China Sea"**

**Manuscript egusphere-2024-72**

**Response to Anonymous Referee #1:**

We sincerely appreciate the time and effort invested by both the reviewer and the editor in evaluating our paper titled **"Evaluation of CMIP6 Models Performance in Simulating Historical Biogeochemistry across Southern South China Sea"** submitted for publication in Biogeosciences. We are grateful for the positive feedback and the insightful comments provided, which is detailed in this report and also in the upcoming revised manuscript. The majority of the suggestions put forth by the reviewers have been incorporated, and in the limited cases where we have not, we have provided a detailed description of the justification for each decision. For ease of reference, we have provided a detailed point-by-point response to the reviewer's comments, with line numbers in each response refers to the revised manuscript. Additionally, we have included the recently added discussions, revised figures and newly added references in this "response to the review" report for the convenience of the reviewer/editor to refer.

**REVIEWER-1 MAJOR COMMENTS**

**Comment 1:**

An in-depth process evaluation is needed

First, irrespective from the valuable effort (and outcome) of this work, that is evaluated marine biogeochemistry models embedded in CMIP6 ESMs, I think further attention to other physical drivers or control variables would be needed to better understand the performance of the models.

Another need would be to scrutinize inter-parameters relationship (chl-biomass, biomass-nitrate). I think the reader could be interested in further understanding of biases propagation across the marine biogeochemical cycles.

A similar attention would be needed also to dive a bit further into the model process parameterization. For instance, not all model simulates prognostically chlorophyll. This latter is derived from the carbon-to-chlorophyll ratio and phytoplankton biomass (see Séférian et al. 2020, who did an in-depth evaluation of model parameterization).

**Response:**

- Thank you for highlighting the importance of analysing the physical drivers or control variables of models to enhance our understanding of their performance. In this paper, our primary focus is to assess the performance of CMIP6 models in reproducing historical biogeochemical variables. While we appreciate the suggestion to include analysis of physical drivers or control variables, we believe that expanding the scope in this direction may detract from the main focus of our research. Moreover, it's worth noting that previous studies such as Jin et al. (2023) and Fan & Zhou (2023) have already evaluated the performance of CMIP6 models on some physical drivers/control variables (i.e., SST, Asian-Pacific Oscillation and precipitation), including those relevant to our study domain (southern South China Sea) and common models utilized in our research (ACCESS-ESM1-5, CanESM-5, GFDL-ESM4, MIROC, MPI-ESM-1-2-HAM, MPI-ESM-1-2-HR, MPI-ESM-1-2-LR, NorESM2-LM, NorESM2-MM & MRI-ESM2-0). However, we acknowledge the value of conducting a separate analysis on the performance of individual model's physical drivers or control variables alongside biogeochemical responses in future research. This suggestion has been duly noted and included as a recommendation in our paper **(L564 – L577)**.

- Thank you for highlighting the importance of inter-parameters relationship. We also agree with your suggestion, and as it aligns with the scope of our study, we accordingly produced the correlation between Chlorophyll-Phytoplankton biomass and Nitrate-Phytoplankton biomass **(L395 – L466)**. To explain this, we added the new section to the revised manuscript as follows:

> *"4.2 Inter-variable Relationship*
> *4.2.1 Chlorophyll – Phytoplankton*
>     *The correlation between chlorophyll and phytoplankton serves as a critical indicator of marine ecosystem health and productivity. As chlorophyll is a pigment essential for photosynthesis in phytoplankton, its concentration is often used as a proxy for phytoplankton in aquatic environments (e.g. Petrik et al., 2022). A strong positive correlation between chlorophyll and phytoplankton signifies robust primary production and nutrient availability, highlighting favorable conditions for marine life. Conversely, a weak or negative correlation may indicate nutrient limitation, environmental stressors, or other factors affecting phytoplankton growth. Understanding this correlation provides valuable insights into ecosystem dynamics, nutrient cycling and the impacts of environmental changes on marine ecosystems. However, it's important to note that not all models simulate chlorophyll concentrations prognostically. Instead, some models derive chlorophyll concentrations from the carbon-to-chlorophyll ratio and phytoplankton biomass. This approach acknowledges the intricate relationship between chlorophyll production and phytoplankton biomass, ensuring a comprehensive representation of primary productivity in marine environments. The correlation between chlorophyll and phytoplankton biomass can also provide insights into whether a model produces chlorophyll prognostically or not. If a model*

*simulates chlorophyll concentration prognostically, there should be a strong positive correlation between chlorophyll concentration and phytoplankton biomass. This correlation arises from the direct influence of phytoplankton biomass on chlorophyll production through photosynthesis. On the other hand, if chlorophyll concentrations are derived from the carbon-to-chlorophyll ratio and phytoplankton biomass, the correlation may still exist but could be influenced by additional factors such as nutrient availability and environmental conditions. Therefore, analysing the correlation between chlorophyll and phytoplankton biomass can help discern the modeling approach used to simulate chlorophyll dynamics within the ocean model. During our analysis of the linear regression between chlorophyll and phytoplankton, most of the selected CMIP6 models showed a strong positive correlation between chlorophyll and phytoplankton concentrations (**Fig. 26**). Notably, ACCESS-ESM1-5, MIROC-ES2L, and MRI-ESM2-0 demonstrated particularly robust positive correlations, with coefficient (R) reaching 1. However, examination of the 95% confidence interval suggests that the relationship between chlorophyll and phytoplankton in MIROC-ES2H, MPI-based models and NorESM-based models deviates from proximity. This disparity in confidence intervals among those models could arise from the differences in model parameterizations and structural complexities, resulting in differing levels of uncertainty in the simulated relationships between chlorophyll and phytoplankton. Models with simpler representations of biological processes or less accurate parameterizations may exhibit wider confidence intervals due to increased uncertainty in their outputs. Specifically, MPI-based and NorESM-based models employed the HAMOCC version of the biogeochemistry model, which does not explicitly simulate chlorophyll concentrations (Paulsen et al., 2017; Tjiputra et al., 2020).*

**4.2.1 Nitrate – Phytoplankton**

*Through the application of linear regression analysis, we investigated the correlation between surface nitrate levels and phytoplankton biomass. While analysing the various CMIP6 models, it was observed that none of the models displayed a correlation (R) > 0.8. However, among the selected models, CanESM5-ConOE, MIROC-ES2H, MPI-ESM1-2-LR, MRI-ESM2-0, and UKESM1-0-LL exhibited a statistically significant positive correlation (R > 0.5, p < 0.001) with surface nitrate and phytoplankton. Conversely, the remaining models demonstrated considerably weaker positive correlations, with only CanESM5, GFDL-ESM4, MIROC-ES2L and MPI-ESM1-2-HR displaying a slight negative correlation. This slight negative correlation could stem from various factors that it may reflect discrepancies in those model dynamics, such as the representation of nutrient uptake or phytoplankton growth rates. Biological processes within the models might not accurately capture the complexities of phytoplankton-nutrient interactions. For example, variations in biogeochemical tracers within model frameworks could influence model efficacy. Specifically, except UKESM1-0-LL and MIROC-based models, all other selected models utilize carbon as their primary model currency for representing phytoplankton biomass, incorporating explicit calculations for phytoplankton biomass and they also utilize nitrate and phosphate to constrain bulk phytoplankton growth rates alongside temperature and light. Consequently, their representation of phytoplankton biomass exhibited a weaker correlation with nitrate. Despite the use of carbon tracer, MPI-ESM1-2-LR incorporates a newly resolved nitrogen-fixing formulation within its biogeochemistry model. This updated formulation introduces an additional prognostic phytoplankton class, replacing the diagnostic formulation of nitrogen-fixation utilized in MPI-ESM-LR (Paulsen et al., 2017; Mauritsen et al., 2019). As a result, this adjustment enables the model to capture the nitrogen response to phytoplankton biomass positively. UKESM1-0-LL employed nitrogen as its primary currency, resulting in a more pronounced quantitative representation of phytoplankton biomass in response to increased nitrate levels compared to the other models (**Fig. 27**). While MIROC-ES2L primarily utilizes nitrogen as its tracer, it also integrates the phosphorus cycle within the model framework to accurately depict the strong phosphorus limitation on the growth of diazotrophic phytoplankton (Hajima et al., 2020). Consequently, this incorporation of the phosphorus cycle may account for phosphorus limitation, resulting in the observed negative correlation between nitrate and phytoplankton biomass within our study area. In the case of GFDL-ESM4, the negative correlation between nitrate and phytoplankton could potentially originate from their model parametrization. In their framework, phytoplankton were categorized based on size and functional type, with small phytoplankton being nitrogen-rich and large phytoplankton*

*phosphate-rich, thereby attributing characteristic N:P ratios (Stock et al., 2020). Thus, differences in parameterizations, data initialization and model resolution could contribute to divergent simulated responses (Séférian et al., 2020). This discrepancy underscores the variability among model outputs and highlights the importance of further scrutinizing model performances based on their parametrization and phenological structural. It is important to analyse how these models are formulated and their roles in nutrient uptake, zooplankton grazing, phytoplankton growth, and plankton mortality within the trophic transfer processes. This approach will aim to refine our comprehension of the intricate dynamics governing marine ecosystems within each model."*

- Thank you for bringing up the importance of model process parameterization. We acknowledge your point that not all models incorporate chlorophyll prognostically. However, delving deeply into model process parameterization might extend beyond the scope of our present study, which primarily aims at evaluating model reproducing skill on biogeochemical variables. Nevertheless, we recognize the potential value in comparing the parameterization processes of different models for future research. We suggest considering this as a separate study to explore the performance of these models more comprehensively in the future. Additionally, we have included this suggestion in our recommendations (**L L564 – L575**).
* * *
**Comment 2:**

The choice of biogeochemical tracers

The authors focused on key biogeochemical variables: chlorophyll, plankton biomass, nitrate and oxygen. All of these variables are analyzed at surface oceans — which makes sense for biological markers such as chlorophyll and plankton biomass but a surface analysis can hide model biases for oxygen and nitrate (oxycline and nutricline). There an expanded analysis including nitrate and oxygen profiles across the multi-model ensemble would be relevant.

I also think that phosphate and net marine productivity would be relevant to liase with global studies such as Kwiatkowski et al. 2020.

**Response:**

- Thank you for bringing up the importance of analysing the bias at depth to understand the oxycline and nutricline dynamics of the models. In response, considering the complex bathymetry of southern SCS region, we have addressed this concern by presenting the spatial distribution of seasonal variations in nitrate and oxygen at two distinct depths (70m and 1000m) for each model, rather than providing profiles (**L270 – L320**). The depth of 70 meters has been selected to depict the dynamics of the nutricline/oxycline in the shelf break region of Sunda Shelf region, while a depth of 1000 meters has been chosen to represent the deep layer. Accordingly, we have discussed the biases

- For nitrate (**L278 – L291**) as follows:

[revised manuscript text omitted]

- We sincerely thank the reviewer for recommending the inclusion of additional biogeochemical parameters such as phosphate and net marine productivity, following the work of Kwiatkowski et al. 2020. However, after careful consideration, we choose to include only variables that were consistently available across all selected models' historical and projection scenarios. Since phosphate and net productivity variables were absent in some of the historical and projection scenarios, we prioritized models that shared common variables in both their historical and projection scenarios.

**********************************

**Comment 3:**

Choice of reference datasets

Finally I'm also concerned by the use of the single CMEMS data are taken as ground truth observations although they aren't.

As far as I am aware, CMEMS is a model-based data product. For the ocean physics the data product benefit from observation assimilation (it is thus a reanalysis) but for marine biogeochemistry it is only the marine biogeochemical model PISCES without any constraints.

I understand a focus on a given regional domain could pose challenge in terms of data availability. However, there are many high-resolution datasets for surface chlorophyll, net primary productivity etc. based on remote-sensing observation that can provide support to this work.

I would recommend to use them in addition to CMEMS. Indeed, past work (Lee et al. 2016) showed that this data product do not outcomes standard CMIP5 models when compared to true insitu observations.

**Response:**

- Thank you for your concern about the usage of reference data (CMEMS). Unfortunately, CMEMS is the only available timeseries hindcast data for the biogeochemistry in southern South China Sea region. As we stated already in the manuscript **(L136)** that CMEMS biogeochemistry product quality has been validated by Mercator-Ocean and they have confirmed and published the quality of this data through comparisons with recognized datasets like Ocean Color, World Ocean Atlas and Globcolour products in their Quality Information Document (QuID; Perruche et al., 2019).

- In order to improve more confidence of this dataset in southern South China Sea region, we have discussed some literatures in our revised manuscript **(L141 – L150)**, in which, authors have validated this data product with the in-situ measurement in this study region, i.e., Wahyudi et al, (2023) validated the POC, Chlorophyll, Dissolved Oxygen, Nitrate, Phosphate and Silicate obtained from CMEMS biogeochemistry product by comparing it with in-situ data collected during the Widya Nusantara Expedition 2015 (Triana et al., 2021) in the upwelling area of southwestern Sumatra waters. They found that the mean absolute percentage error values were lower than 15%, indicating the reliability of the CMEMS biogeochemistry model data in our study area. Additionally, Chen et al, (2023) also used the daily chlorophyll concentration data from the same CMEMS biogeochemical product in south china sea region. By utilizing this CMEMS biogeochemistry model dataset, Wahyudi et al. (2023) and Chen et al. (2023) highlights the proficiency of the CMEMS biogeochemistry model data in reproducing both the climatic patterns and fluctuations observed within its biogeochemical variables in southern South China Sea. This gave us confidence in utilizing the CMEMS biogeochemical dataset as the reference model to assess other models in this region (southern South China Sea).

- Thank you for recommending the other available dataset. Although there are other datasets available for surface chlorophyll and net primary productivity, the purpose of using the single reference dataset (CMEMS) is to ensure consistency in the evaluation process. By relying on a homogeneous dataset, we aim to enhance the reliability of our evaluation results and greater confidence in the findings.
* * *
**REVIEWER-1 MINOR COMMENTS**

**Comment 1:**

L9 why? they are poorly constrained

**Response:**

Thank you for bringing our oversight on explain this matter. The emphasis on selecting the biogeochemical variables is not primarily on constraint, but rather on highlighting the significance of key biogeochemical tracers and their availability across the chosen models and their corresponding projection scenarios. Accordingly, we have also explained this matter in our revised manuscript **(L08)**.
* * *
**Comment 2:**

L12 overestimations or underestimations, in what? (quantitative measures or score would be useful)

**Response:**

Thank you for bringing up this important point. We apologize for the oversight in not clarifying. The overestimations or underestimations refer to quantitative measures. We have incorporated this clarification into the revised manuscript as per your recommendation **(L13)**.
* * *
**Comment 3:**

L23-25 This statement is inexact, recent work from Kwiatkowski et al 2020 shows that marine NPP in 2100 remains largely uncertain

**Response:**

We acknowledge the inexactness of the statement regarding marine NPP uncertainty in 2100. Upon thorough examination of Kwiatkowski et al.'s (2020) findings, we recognize that the uncertainty in primary production largely persists, even within the CMIP6 framework. To address this, we have diligently incorporated the relevant insights from Kwiatkowski et al. (2020) into our revised manuscript **(L23 – L29)** as follows:

*"For example, Kwiatkowski et al. (2020) discovered that the multi-model global mean projections from the Coupled Model Intercomparison Project Phase 6 (CMIP6), under high-emission to low-emission scenarios, indicate a consistent decrease in net primary production. Notably, there is a significant increase in inter-model uncertainty compared to CMIP5. This increased uncertainty is linked to changes in the temporal patterns of phytoplankton resource availability and grazing pressure within CMIP6 (Kwiatkowski et al., 2020). This carries significant implications for evaluating ecosystem impacts on a regional scale. (Tagliabue et al., 2021)."*

\*\*\*\*\*\*\*\*\*\*\*\*\*\*\*\*\*\*\*\*\*\*\*\*\*\*\*\*\*\*\*\*\*\*\*\*\*\*

**Comment 4:**

L47 Consider to refer to Séférian et al. 2020 for the evaluation of global ESMs

**Response:**

Thank you for your valuable suggestion to consider Séférian et al. (2020). After careful review, we agree that their work on global ESMs evaluation is relevant to our study. Thus, we have incorporated a citation to Séférian et al. (2020) in our revised manuscript **(L53)**.

\*\*\*\*\*\*\*\*\*\*\*\*\*\*\*\*\*\*\*\*\*\*\*\*\*\*\*\*\*\*\*\*\*\*\*\*\*\*

**Comment 5:**

L58 This aspect has to be analysed with caveats because for several bgc models chlorophyl is derived from phytoplankton biomass improving linkage with zooplankton response (see major comments)

**Response:**

We sincerely appreciate your input on this matter. However, in this particular context, our aim was to convey the findings of Petrik et al. (2022), who extensively examined the model's simulation of zooplankton and delved into the relationship between zooplankton and chlorophyll-a. It is worth noting, as we have highlighted in our revised manuscript **(L61 – L65)**, that Petrik et al. (2022) underscored *chlorophyll-a as a proxy for phytoplankton* for several reasons: (a) it provides the most comprehensive observations across both time and space, (b) it is more readily observable compared to phytoplankton biomass, which typically requires physical sampling, or primary production, which relies on physical experiments or algorithms based on satellite chlorophyll data that yield varied outcomes, and few other reasons.

\*\*\*\*\*\*\*\*\*\*\*\*\*\*\*\*\*\*\*\*\*\*\*\*\*\*\*\*\*\*\*\*\*\*\*\*\*\*

**Comment 6:**

L102 it would be nice to have a table of the references datasets used here
L112 CMEMS : it might be good to make it clear if CMEMS data is reanalysis and model reconstruction.
As far as I am aware of CMEMS relies on NEMO-PISCES+ assimilation for ocean hydrodynamics. PISCES, the marine bgc model adjust to improved physics but it is free (no assimilation). Therefore using CMEMS for evaluating marine bgc models is comparing models with another (more constrained) model.

**Response:**

We sincerely appreciate your thoughtful consideration of this matter and your concern. The utilization of the CMEMS biogeochemical dataset in our study is primarily due to its status as the only available timeseries dataset for our study domain, specifically the southern South China Sea region. Furthermore, we have referenced literature that has validated this data product for this region through comparisons with in-situ measurements, instilling confidence in its use for evaluating the models in this area. We kindly wish to inform you that a detailed response addressing this specific issue was provided in our response to **Reviewer-1 major comment 3**.

**\*\*\*\*\*\*\*\*\*\*\*\*\*\*\*\*\*\*\*\*\*\*\*\*\*\*\*\*\*\*\*\*\*\*\*\*\***

**Comment 7:**

L125 Table 1: I'm suprised to see PISCES (IPSL, CNRM, EC-Earth) and MARBL (CESM2) excluded from this exhaustive analysis — what are the reason?

**Response:**

We appreciate your attention to this issue. As explained in **L130** that our model selection procedure was based on the availability of selected biogeochemical variables across historical or projected scenarios. Consequently, the EC-Earth models do not include the Phytoplankton biomass (phyc) variable, while CESM2 lacks the dissolved Oxygen (o2) variable in its historical dataset. Additionally, the CNRM and IPSL-based models showed a standard deviation >50 mg/m3 for the chlorophyll variable compared to reference data, resulting in their exclusion from the analysis.

**\*\*\*\*\*\*\*\*\*\*\*\*\*\*\*\*\*\*\*\*\*\*\*\*\*\*\*\*\*\*\*\*\*\*\*\*\***

**Comment 8:**

L129 Ranking: no cross-variable evaluation? SST-Chlorophyll, etc. (see major comments)

**Response:**

We appreciate your emphasis on evaluating the model's biogeochemistry in relation to its physical drivers. However, the primary focus of our paper is the assessment of CMIP6 biogeochemical model outcomes. While we acknowledge the significance of examining biomass correlations with physical drivers, delving into this aspect might deviate from the central goal of our paper. Instead, we have thoroughly analysed the model's performance concerning Chlorophyll-Phytoplankton biomass and Nitrate-Phytoplankton biomass correlations. We kindly wish to highlight that we addressed this concern in detail in our response to **Reviewer-1 major comment 1**.

**\*\*\*\*\*\*\*\*\*\*\*\*\*\*\*\*\*\*\*\*\*\*\*\*\*\*\*\*\*\*\*\*\*\*\*\*\***

**Comment 9:**

L159-160: please consider sharing scripts and data for this work (EGU journals recommandations)

**Response:**

Thank you for your thoughtful consideration of our work and for your interest in the data. We have indeed provided the source link to the dataset we utilized **(L155)** from CMEMS biogeochemistry hindcast dataset (ID: GLOBAL_MULTIYEAR_BGC_001_029) and CMIP6 dataset which was obtained from ESGF data portal, which are freely accessible for public use. Furthermore, we employed widely accepted basic formulae for our statistical analyses, as detailed in our manuscript **(L72 – L80)**.

**\*\*\*\*\*\*\*\*\*\*\*\*\*\*\*\*\*\*\*\*\*\*\*\*\*\*\*\*\*\*\*\*\*\*\*\*\***

**Comment 10:**

L164-165 yes and no there are several paper that indicates how seasonal cycle can help to constrain projections (Behrenfeld et al. 2006, Kwiatkovski et al. 2017, etc..)

**Response:**

Thank you for bringing this matter to our attention. We sincerely apologize for any confusion caused by our previous statement. Upon careful review of the works by Behrenfeld et al. (2006) and Kwiatkovski et al. (2017), we have revised our statement accordingly. Instead of asserting that *"the yearly cycle of seasons does not fully*

*capture the long-term changes associated with climate change,"* we have amended it to reflect that *"the yearly cycle of seasons partially captures the long-term changes associated with climate change"*. These long-term changes encompass shifts in average temperatures, alterations in precipitation patterns, changes in the frequency and intensity of extreme weather events, and other systemic shifts that extend beyond the periodicity of seasonal cycles. While temporal cycles are indeed important components of climate variability, they offer only a partial perspective on the broader and more profound changes occurring in the Earth's climate system. Thus, our revised statement **(L197 – L201)** aims to convey that "*the yearly cycle of seasons partially captures the long-term changes associated with climate change"*.
* * *
**Comment 11:**

L183: typo: ensemble mean ?
'Moderate' needs to be quantified with skillscore

**Response:**
Thank you for pointing out this oversight. We have corrected the typo to *"ensemble"* in the manuscript. Furthermore, we've also provided clarification on the term "*moderate"* by delineating its specific meaning in context. In the discussion of spatial variation and bias in **L219**, *"moderate"* denotes the model's performance at an average level in simulating the variables quantitatively, relative to the reference data. Likewise, in reference to Taylor's diagram in **L477**, *"moderate"* indicates the model's placement between the last and first models in TD.
* * *
**Comment 12:**

Figures 2-5: hard to see model differences.
I would present model baises against the reference

**Response:**
We sincerely appreciate your insightful feedback. We acknowledge your concern regarding the clarity of the model differences in Figures 2-5 in old manuscript. In response to your suggestion, we have changed the seasonal climatology figures to illustrate the seasonal bias against the reference data in our revised manuscript **(Figs. 2–25)**. The revised figures can also be found in this report below.
* * *
**Comment 13:**

Figure 7: why for surface oxygen of MPI-based models outbest the others whereas it is not the case for the other variables ?  For these later MIROC-ES2L outbests the other models.
These differences needs to be explained. My guess is that SST biaises in MPI-based models are much lower than the other models in this zone which explains why surface oxygen is better represented.

**Response:**
Thank you for bringing to our knowledge about the differences in model out best behaviour and also acknowledge for providing us the solution as well. Following your suggestion, we have conducted a thorough review of some literature and found our findings align with your suggestion, and included a statement in **L531 – L537** as follows:

*"MIROC-ES2H demonstrated superior performance across all variables except oxygen, consistently achieving scores above 0.2. For oxygen, MPI-ESM1-2-LR ranked highest due to its exceptional spatial representation and accurate seasonal pattern captured in the Taylor diagram. This outperformance of MPI-based models for oxygen can be attributed to the effectiveness of the physical drivers within that model. Jin et al. (2023) showed that MPI-based models excel in simulating climatological sea surface temperature (SST) during boreal winter and summer, and are among the top performers in reproducing SST climatology in Asian marginal seas due to their minimal SST biases. Thus, MPI-based models outperform others in replicating surface oxygen variables in our study."*
* * *
**Revised Figures:**

[Figure]

**Figure 1.** DJF spatial biases of surface chlorophyll for 13 individual models and model ensemble relative to reference.

[Figure]

**Figure 2.** Same as Fig.2 but for JJA.

[Figure]

**Figure 3.** The mean bias of surface chlorophyll for both seasons (DJF, JJA) and annual.

[Figure]

**Figure 4.** DJF spatial biases of surface phytoplankton for 13 individual models and model ensemble relative to reference.

[Figure]

**Figure 5.** Same as Fig. 5 but for JJA.

[Figure]

**Figure 6.** The mean bias of surface phytoplankton for both seasons (DJF, JJA) and annual.

[Figure]

**Figure 7.** DJF spatial biases of surface nitrate for 13 individual models and model ensemble relative to reference.

[Figure]

**Figure 8.** Same as Fig. 8 but for JJA.

[Figure]

**Figure 9.** The mean bias of surface nitrate for both seasons (DJF, JJA) and annual.

[Figure]

**Figure 10.** DJF spatial biases of nitrate at 70-meter depth for 13 individual models and model ensemble relative to reference.

[Figure]

**Figure 11.** Same as Fig. 11 but for JJA.

[Figure]

**Figure 12.** The mean bias of nitrate at 70-meter depth for both seasons (DJF, JJA) and annual.

[Figure]

**Figure 13.** DJF spatial biases of nitrate at 1000-meter depth for 13 individual models and model ensemble relative to reference.

[Figure]

**Figure 14.** Same as Fig. 14 but for JJA.

[Figure]

**Figure 15.** The mean bias of nitrate at 1000-meter depth for both seasons (DJF, JJA) and annual.

[Figure]

**Figure 16.** DJF spatial biases of surface oxygen for 13 individual models and model ensemble relative to reference.

[Figure]

**Figure 17.** Same as Fig. 17 but for JJA.

[Figure]

**Figure 18.** The mean bias of surface oxygen for both seasons (DJF, JJA) and annual.

[Figure]

**Figure 19.** DJF spatial biases of oxygen at 70-meter depth for 13 individual models and model ensemble relative to reference.

[Figure]

**Figure 20.** Same as Fig. 20 but for JJA.

[Figure]

**Figure 21.** The mean bias of oxygen at 70-meter depth for both seasons (DJF, JJA) and annual.

[Figure]

**Figure 22.** DJF spatial biases of oxygen at 1000-meter depth for 13 individual models and model ensemble relative to reference.

[Figure]

**Figure 23.** Same as Fig. 23 but for JJA.

[Figure]

**Figure 24.** The mean bias of oxygen at 1000-meter depth for both seasons (DJF, JJA) and annual.

[Figure]

**Figure 25.** Relationships between chlorophyll and phytoplankton for model ensemble and 13 individual models in southern South China Sea during the study period (1993 – 2014). Dashed red lines represent 95% confidence interval with 0.05 as the level of significance (α).

[Figure]

**Figure 26.** Relationships between nitrate and phytoplankton for model ensemble and 13 individual models in southern South China Sea during the study period (1993 – 2014). Dashed red lines represent 95% confidence interval with 0.05 as the level of significance (α).

[Figure]

**Figure 27.** Annual Taylor Diagram for (a) chlorophyll, (b) phytoplankton, (c) nitrate and (d) oxygen.

---

## Author Comment (AC2)

Response to review of **"Evaluation of CMIP6 Models Performance in Simulating Historical Biogeochemistry across Southern South China Sea"**

**Manuscript egusphere-2024-72**

**Response to Anonymous Referee #2:**

We sincerely appreciate the time and effort invested by both the reviewer and the editor in evaluating our paper titled **"Evaluation of CMIP6 Models Performance in Simulating Historical Biogeochemistry across Southern South China Sea"** submitted for publication in Biogeosciences. We are grateful for the positive feedback and the insightful comments provided, which is detailed in this report and also in the upcoming revised manuscript. The majority of the suggestions put forth by the reviewers have been incorporated, and in the limited cases where we have not, we have provided a detailed description of the justification for each decision. For ease of reference, we have provided a detailed point-by-point response to the reviewer's comments, with line numbers in each response refers to the revised manuscript. Additionally, we have included the recently added discussions, revised figures and newly added references in this "response to the review" report for the convenience of the reviewer/editor to refer.

**Comment 1A:**

The aim of the paper is to rank 13 CMIP6 ESM simulations based on their ability to reproduce selected observed biogeochemical variables. However, the dataset that the author chose is not strictly observations. Based on the link they provided in line 123, the CMEMS ocean biogeochemistry product is based on the PISCES model output (although it is forced with reanalysis product). I also noticed that among the 13 CMIP6 ESMs, the authors have not chosen IPSL-CM6A ESM, which includes PISCES as its ocean biogeochemical model. I understand that in-situ observations may be rare in this region, but to truly assess the CMIP6 ensemble and individual models, I suggest the authors could compare the CMIP6 models with satellite-derived chlorophyll-a and primary production, as well as the World Ocean Atlas product for nitrate and oxygen.

**Response:**

- We sincerely appreciate your attention to the matter regarding the usage of reference data (CMEMS) in our study. We understand the importance of ensuring the robustness and reliability of the data utilized in our research, but CMEMS is the only available timeseries hindcast data for the biogeochemistry in southern South China Sea region. We have acknowledged this in our manuscript **(L136)** and emphasized that the quality of CMEMS biogeochemistry product has been validated by Mercator-Ocean. Their validation process, outlined in the Quality Information Document (QuID; Perruche et al., 2019), includes comparisons with recognized datasets such as Ocean Color, World Ocean Atlas, and Globcolour products, ensuring the credibility of the data.

- Furthermore, to boost confidence in this dataset within the southern South China Sea region, we have discussed some literatures in our revised manuscript **(L141 – L150)**, in which, authors have validated this data product with the in-situ measurement in this study region, i.e., Wahyudi et al, (2023) validated the POC, Chlorophyll, Dissolved Oxygen, Nitrate, Phosphate and Silicate obtained from CMEMS biogeochemistry product by comparing it with in-situ data collected during the Widya Nusantara Expedition 2015 (Triana et al., 2021) in the upwelling area of southwestern Sumatra waters. They found that the mean absolute percentage error values were lower than 15%, indicating the reliability of the CMEMS biogeochemistry model data in our study area. Additionally, Chen et al, (2023) also used the daily chlorophyll concentration data from the same CMEMS biogeochemical product in south china sea region. By utilizing this CMEMS biogeochemistry model dataset, Wahyudi et al. (2023) and Chen et al. (2023) highlights the proficiency of the CMEMS biogeochemistry model data in reproducing both the climatic patterns and fluctuations observed within its biogeochemical variables in southern South China Sea. This gave us confidence in utilizing the CMEMS biogeochemical dataset as the reference model to assess other models in this region (southern South China Sea).

- While we appreciate your suggestion of alternative datasets, our decision to utilize CMEMS as the sole reference dataset was made to maintain consistency in the evaluation process. By adhering to a homogeneous dataset, we aim to ensure the integrity and reliability of our evaluation results, thereby instilling greater confidence in our findings.

- Additionally, IPSL-based models showed a standard deviation $>50$ mg/m$^3$ for the chlorophyll variable compared to reference data, resulting in their exclusion from the analysis.

**Comment 1B:**

Since the paper also looks at the seasonal trend of biogeochemical properties, it could benefit from exploring whether different CMIP6 models can capture phytoplankton phenology (e.g., Racault et al., 2015; Gittings et al., 2018), which is an important indicator.

**Response:**

We appreciate your insightful comment and apologize for any confusion regarding our approach. Our examination primarily focused on the seasonal spatial climatology (now presented as seasonal spatial bias in **Figs. 2-25**), not the seasonal trend map.

While we acknowledge the significance of exploring phytoplankton phenology in CMIP6 models, as suggested by the references you provided, conducting such studies requires extensive time and resources and also, we afraid that incorporating phenology in this study could potentially diverge the main scope of our current investigation. However, we recognize the importance of this aspect and have duly noted it as a potential avenue for future research in our study (**L454 – L458**). Thank you for bringing this to our attention.

************************************

**Comment 2:**

Indeed, most of the biological activity occurs near the surface layers of the ocean, but it's important to consider the biogeochemical dynamics near the seabed, particularly in shelf seas, as they can have complex structures through interactions of ocean physics with biological processes, such as export and remineralization. I would appreciate the inclusion of depth profiles and benthic concentrations of oxygen and nitrate – this would provide a more thorough assessment of the biogeochemical properties. Furthermore, most of the biogeochemical models used in CMIP6 are not specifically built for shelf seas. It would be interesting to see whether these models can represent nutrient and oxygen distribution at shallower depths.

**Response:**

- Thank you for bringing up the importance of analysing the bias at depth to understand the oxycline and nutricline dynamics of the models. In response, considering the complex bathymetry of southern SCS region, we have addressed this concern by presenting the spatial distribution of seasonal variations in nitrate and oxygen at two distinct depths (70m and 1000m) for each model, rather than providing profiles (**L270 – L320**). The depth of 70 meters has been selected to depict the dynamics of the nutricline/oxycline in the shelf break region of Sunda Shelf region, while a depth of 1000 meters has been chosen to represent the deep layer. Accordingly, we have discussed the biases

- For nitrate (**L278 – L291**) as follows:

    *"Furthermore, delving into model biases at deeper levels, especially concerning nutrient dynamics, will provide more insights into the model's accuracy in simulating the nutricline. Consequently, we analysed the nitrate concentrations at depths of 70m (**Figs. 11 – 13**) and 1000m (**Figs. 14 - 16**). In contrast to surface nitrate, most models exhibited a negative bias at deeper layers (70m and 1000m), with an average range of -2 to -8 mmol/m³ across the study area. Among these models, MPI-based models showed the least negative bias at 70m depth; however, as depth increased to 1000m, their bias shifted towards the positive (**Fig. 13 and 16**, respectively). MIROC-based and MPI-based models exhibited the least bias in nitrate concentrations at both surface and deep layers compared to reference data. This may be attributed to the near balance achieved between nitrogen cycle sources (such as nitrogen fixation, atmospheric nitrogen deposition, and riverine nitrogen input) and sinks (including denitrification, nitrous oxide emission, and sedimentary loss) over the long spin-up period (Mauritsen et al., 2019; Hajima et al., 2020). In contrast, CanESM5-based models demonstrated minimal nitrate bias at the surface but showed varying positive and negative biases in deep layers. These discrepancies arise from the simplified parameterization of denitrification in their BGC models. In these models, denitrification in the deep layers is set to balance the rate of nitrogen fixation and is vertically distributed in proportion to the detrital remineralization rate. However, in reality, nitrogen fixation and denitrification are not constrained to balance within the water column at any single location; rather, denitrification primarily occurs in anoxic areas (Swart et al., 2019). Notably, no seasonal bias in all selected models were observed at the deep layer (1000m; **Fig. 16**)."*

- For oxygen (**L293 – L313**) as follows:

    *"During the observation of oxycline dynamics in the selected models, it was noted that the oxygen exhibited a positive bias at a depth of 70m, transitioning to a negative bias with increasing depth (1000m)*

*(Fig. 20 and 25). Moreover, UKESM1-0-LL consistently exhibited a substantial positive bias from the surface to the depth of 70m (~50 mmol/m³) and shifts to negative bias of -40 mmol/m³ at 1000m relative to its surface bias. Similarly, CanESM5 and MIROC-based models displayed markedly high negative biases at a depth of 1000m, but with comparatively lesser negative biases at 70m. Multiple factors could contribute to biases in the simulation of nutricline/oxycline dynamics by models. Inaccuracies in simulating nutricline dynamics may arise from errors in parameterizing physical, chemical and biological processes relevant to these dynamics. In winter, most models overestimate oxygen levels at the surface and at a depth of 70 meters. This positive bias in oxygen concentration may result from excessively intense winter mixing of cold, oxygen-rich waters from the northern boundary of the southern SCS into the Sunda Shelf region (Thompson et al., 2016), which transports an excessive amount of surface oxygen to deeper layers. Additionally, nutrient trapping issues may also contribute to the remaining model bias (Six & Maier-Reimer, 1996). Moreover, the exclusion of relevant processes or feedback mechanisms influencing nutricline dynamics within the model, such as nutrient upwelling, microbial remineralization and ocean stratification, may lead to biased simulation outcomes. Additionally, structural uncertainties embedded in the model formulation, including simplifications or assumptions regarding complex processes, may also play a role in generating biases in simulation results. For example, advancements in model parameterization and representation of biogeochemical fluxes have led to consistent improvements in the mean states of nutrient dynamics in CMIP6 models, such as GFDL-ESM4, MIROC-based, MPI-ESM1-based, and NorESM2-based models. Specifically, improvements in GFDL-ESM4 performance are attributed to a series of updates and changes in model physics (such as mixing and climate dynamics) and biogeochemical parameterizations such as the implementation of a revised remineralization scheme for organic matter that depends on oxygen and temperature (Laufkötter et al., 2017)."*

**\*\*\*\*\*\*\*\*\*\*\*\*\*\*\*\*\*\*\*\*\*\*\*\*\*\*\*\*\*\*\*\*\*\*\*\*\***

**Comment 3:**

Although the authors put a great effort in evaluating CMIP6 model outputs, the model structures could also be evaluated; how biogeochemical tracers are represented, and whether these representations affect the performance of the model in the southern SCS. Perhaps the authors can add another table which biogeochemical tracers these models represent (e.g., in MEDUSA-2 (UKESM), it does not represent diazotrophic phytoplankton, explicitly calculates phytoplankton chlorophyll, and uses N as model currency, while in OECO-2 (MIROC), it has diazotrophic phytoplankton with C as model currency and includes Phosphate as nutrients), and perhaps also how they are formulated, especially when it involves trophic transfer (e.g. nutrient uptake, zooplankton grazing, and phytoplankton growth, and plankton mortality). These additions can add some discussion on how model representation (and structure) may affect model performance in the shelf seas, instead of repeatedly saying that underestimation/overestimation is due to zooplankton grazing/phytoplankton productivity/nutrient uptake.

**Response:**

We sincerely appreciate your insightful comments regarding the model structures. Following your suggestion, we have incorporated an overview of how tracers and model structure affect performance into our discussion, specifically when examining inter-parameter relationships such as chlorophyll-biomass and biomass-nitrate in in **L432 – L454** as follows:

*"This slight negative correlation could stem from various factors that it may reflect discrepancies in those model dynamics, such as the representation of nutrient uptake or phytoplankton growth rates. Biological processes within the models might not accurately capture the complexities of phytoplankton-nutrient interactions. For example, variations in biogeochemical tracers within model frameworks could influence model efficacy. Specifically, except UKESM1-0-LL and MIROC-based models, all other selected models utilize carbon as their primary model currency for representing phytoplankton biomass, incorporating explicit calculations for phytoplankton biomass and they also utilize nitrate and phosphate to constrain bulk phytoplankton growth rates alongside temperature and light. Consequently, their representation of phytoplankton biomass exhibited a weaker correlation with nitrate. Despite the use of carbon tracer, MPI-ESM1-2-LR incorporates a newly resolved nitrogen-fixing formulation within its biogeochemistry model. This updated formulation introduces an additional*

*prognostic phytoplankton class, replacing the diagnostic formulation of nitrogen-fixation utilized in MPI-ESM-LR (Paulsen et al., 2017; Mauritsen et al., 2019). As a result, this adjustment enables the model to capture the nitrogen response to phytoplankton biomass positively. UKESM1-0-LL employed nitrogen as its primary currency, resulting in a more pronounced quantitative representation of phytoplankton biomass in response to increased nitrate levels compared to the other models (**Fig. 27**). While MIROC-ES2L primarily utilizes nitrogen as its tracer, it also integrates the phosphorus cycle within the model framework to accurately depict the strong phosphorus limitation on the growth of diazotrophic phytoplankton (Hajima et al., 2020). Consequently, this incorporation of the phosphorus cycle may account for phosphorus limitation, resulting in the observed negative correlation between nitrate and phytoplankton biomass within our study area. In the case of GFDL-ESM4, the negative correlation between nitrate and phytoplankton could potentially originate from their model parametrization. In their framework, phytoplankton were categorized based on size and functional type, with small phytoplankton being nitrogen-rich and large phytoplankton phosphate-rich, thereby attributing characteristic N:P ratios (Stock et al., 2020). Thus, differences in parameterizations, data initialization and model resolution could contribute to divergent simulated responses (Séférian et al., 2020).”*

\*\*\*\*\*\*\*\*\*\*\*\*\*\*\*\*\*\*\*\*\*\*\*\*\*\*\*\*\*\*\*\*\*\*\*\*\*

**Comment 4:**

The presentation of the results can also be improved. I think it will be easier to follow the results if the authors describe the observed distribution of nitrate, chlorophyll, phytoplankton biomass, and oxygen, then compare them with the model. For the figures, it would be more interesting to see the difference between the CMEMS data and CMIP6 outputs with better figure resolution (especially figure 6). Additional discussion on regions where bias usually occurs in different models will also be interesting (e.g., the shelf seas between Sumatra, the Malaysian peninsula, and Borneo are always high in phytoplankton biomass for UKESM, CanESM5, ACCESS, MPI-ESM1-2, NorESM2).

**Response:**

- We sincerely appreciate your insightful feedback and acknowledge your concern regarding the clarity of the results. Following your suggestion, we have modified the seasonal climatology figure in our revised manuscript to better illustrate the seasonal bias against the reference data (**Figs. 2 – 25**: can be found in this report below).

- Additionally, we have incorporated your recommendation to discuss the spatial diversity in model bias. Our revised manuscript now elaborates this in **L235 – L250** as follows:

*"Most models showed spatial uniformity in their underestimation or overestimation of chlorophyll and phytoplankton, with a few models exhibiting spatial diversities in their estimates. For example, the CanESM-CanOE model consistently overestimates chlorophyll concentration and phytoplankton biomass in both seasons, with mean biases of ~0.49 mg/m³ in DJF and ~0.31 mg/m³ in JJA for chlorophyll, and ~2.2 mmol/m³ in DJF and ~1.5 mmol/m³ in JJA for phytoplankton. This overestimation is particularly pronounced in the region between Sumatra, Peninsular Malaysia, and Borneo, where chlorophyll exceeds 1 mg/m³ and phytoplankton exceeds 5 mmol/m³. Similarly, the UKESM1-0-LL model overestimates chlorophyll and phytoplankton in both seasons, especially in the Gulf of Thailand. These models may have insufficient spatial resolution to capture the fine-scale physical and biological processes in these regions. Important features like small-scale currents, eddies and upwelling events, which significantly affect chlorophyll and phytoplankton distributions, may not be adequately resolved, leading to spatial bias. Generally, ESMs in CMIP6 are developed for open ocean conditions rather than shelf seas. Most CMIP6 ESMs have a coarse resolution (≥ 100 km horizontal) for the ocean component, although some have resolutions of ≤ 25 km, which is considered eddy-permitting. This suggests these ESMs can represent barotropic processes at smaller scales but not baroclinic ones (Chelton et al., 1998). The ability of coarse-resolution CMIP6 ESMs to represent shallow continental shelf waters dynamics with high skill, such as the southern SCS' Sunda shelf region, is limited. Variability in this region is influenced by inflows like the Indonesian Throughflow and SCS Throughflow, which are not resolved by coarse-resolution models (Wang et al., 2024)."*

\*\*\*\*\*\*\*\*\*\*\*\*\*\*\*\*\*\*\*\*\*\*\*\*\*\*\*\*\*\*\*\*\*\*\*\*\*

**REVIEWER-2 SPECIFIC COMMENTS**

**Comment 1:**

L12 – perhaps the authors can add a % or number on the degrees of overestimation and underestimation.

**Response:**

Thank you for highlighting this important point. We apologize for the oversight and any confusion it may have caused. The overestimations or underestimations pertain to quantitative measures that vary depending on the analysed variables and seasons. We have included this clarification in the revised manuscript, following your recommendation (**L13**).

\*\*\*\*\*\*\*\*\*\*\*\*\*\*\*\*\*\*\*\*\*\*\*\*\*\*\*\*\*\*\*\*\*\*\*\*\*

**Comment 2:**

L22-23 - Based on CMIP6 models, NPP trend is uncertain, apart maybe at the Southern Ocean (Tagliabue et al., 2021)

**Response:**

Thank you for your valuable feedback. We acknowledge the inaccuracy in our original statement regarding marine NPP uncertainty in 2100. Accordingly, the corrections have been made in revised manuscript (**L23 – L29**) as follows:

*"For example, Kwiatkowski et al. (2020) discovered that the multi-model global mean projections from the Coupled Model Intercomparison Project Phase 6 (CMIP6), under high-emission to low-emission scenarios, indicate a consistent decrease in net primary production. Notably, there is a significant increase in inter-model uncertainty compared to CMIP5. This increased uncertainty is linked to changes in the temporal patterns of phytoplankton resource availability and grazing pressure within CMIP6 (Kwiatkowski et al., 2020). This carries significant implications for evaluating ecosystem impacts on a regional scale. (Tagliabue et al., 2021)."*

\*\*\*\*\*\*\*\*\*\*\*\*\*\*\*\*\*\*\*\*\*\*\*\*\*\*\*\*\*\*\*\*\*\*\*\*\*

**Comment 3:**

L33-34 - This is not always the case - OBGC models can give the seemingly good representation of historical climate pattern but for the wrong reason. Furthermore, OBGC model results is dependent on its physical forcings (see Sinha et al., 2010)

**Response:**

Thank you for bringing this matter to our attention. In response to your suggestion, we have gone through some literatures and made changes accordingly in **L35 – L42** stating that *"While a model's successful reproduction of historical climate patterns suggests it has captured relevant physical processes and interactions within the Earth's system but this is not always the case. Ocean BGC models can sometimes appear to accurately represent historical climate patterns for incorrect reasons, as their results are highly dependent on the physical forcing applied (Friedrichs et al., 2006; Sinha et al., 2010). For example, minor changes in ocean model circulation can lead to significant variations in biogeochemical conditions. Similarly, Glessmer et al. (2008) discovered that even slight alterations in mixing greatly affects the simulation of primary production and export in the global general circulation models. Therefore, caution is needed when using these models for future climate projections."*

\*\*\*\*\*\*\*\*\*\*\*\*\*\*\*\*\*\*\*\*\*\*\*\*\*\*\*\*\*\*\*\*\*\*\*\*\*

**Comment 4:**

L60 – typo: Tjiputra et **al**, (2020)

**Response:**

Thank you for pointing out this oversight. We have corrected accordingly in **L68**

\*\*\*\*\*\*\*\*\*\*\*\*\*\*\*\*\*\*\*\*\*\*\*\*\*\*\*\*\*\*\*\*\*\*\*\*\*

**Comment 5:**

L61-72 – I'm not so sure if these are appropriate examples. Maybe add studies like Kwiatkowski et al., 2020, Hinrichs et al., 2023

**Response:**

Thank you for your valuable suggestion and for providing literature on this topic. We have thoroughly reviewed the references you recommended, and we have incorporated relevant findings from Kwiatkowski et al., 2020 into our manuscript **L88**.

\*\*\*\*\*\*\*\*\*\*\*\*\*\*\*\*\*\*\*\*\*\*\*\*\*\*\*\*\*\*\*\*\*\*\*\*\*\*

**Comment 6:**

L83-L85 – Why only phytoplankton, chlorophyll, nitrogen, and oxygen? Why not net primary production and or carbon?

**Response:**

Thank you for your concern about the selected variables. We chose to focus on phytoplankton, chlorophyll, nitrogen, and oxygen for following reasons:

1. These variables are fundamental tracers for biological and nutrient dynamics in ocean systems.

2. Phytoplankton and chlorophyll variables serve as effective proxies for primary production, thus encompassing the essential aspect of net primary production.

3. Given that the southern SCS region is recognized as a typical oligotrophic area where primary productivity is primarily constrained by nutrient availability, we specifically included nitrate and oxygen.

4. We considered variables that were consistently available across all selected models' historical and projection scenarios to ensure comparability and consistency in our analysis. Even phosphate variable is unavailable in some of the selected model's scenarios. Thus, phosphate is also excluded.

\*\*\*\*\*\*\*\*\*\*\*\*\*\*\*\*\*\*\*\*\*\*\*\*\*\*\*\*\*\*\*\*\*\*\*\*\*\*

**Comment 7:**

L103-L105 - This sounds like phytoplankton is controlling the physical biogeochemical process?

**Response:**

We sincerely apologize for the confusion caused by our statement. Upon a careful review, we rephrased the statement as *"Within the southern SCS, extensive observations have demonstrated that phytoplankton growth, serving as the primary source of organic matter, significantly influences oceanic carbon cycles. This growth is influenced by monsoon-driven physical and biogeochemical processes, with phytoplankton demonstrating a notable sensitivity to these environmental dynamics."* in **L124 – L127**.

\*\*\*\*\*\*\*\*\*\*\*\*\*\*\*\*\*\*\*\*\*\*\*\*\*\*\*\*\*\*\*\*\*\*\*\*\*\*

**Comment 8:**

L122-123 – is this the hindcast global ocean biogeochemistry? Do you also use the GlobColour for chlorophyll? Please be more specific.

**Response:**

- We sincerely apologize for the confusion made. The CMEMS product used in this study is the hindcast global ocean biogeochemistry dataset, which can be found in CMEMS biogeochemistry hindcast dataset (ID: GLOBAL_MULTIYEAR_BGC_001_029) **(L156).**

- We did not use GlobColour data in our study. Instead, Mercator Ocean used GlobColour data to validate the chlorophyll data within the CMEMS hindcast biogeochemistry dataset, as mentioned in **L138 – L141**.

\*\*\*\*\*\*\*\*\*\*\*\*\*\*\*\*\*\*\*\*\*\*\*\*\*\*\*\*\*\*\*\*\*\*\*\*\*\*

**Comment 9:**

L125 – Perhaps, instead of having 2/3 ESMs with the same OBGC model, maybe choose one of them instead, so you can also look at other models such as PISCESv2 (Aumont et al., 2016), MARBL (Long et al., 2021), BFM5.2 (Lovato et al., 2022)?

**Response:**

Thank you for suggesting a method to choose models. We will incorporate this method in our future studies. Additionally, as explained in **L129 – L132** that our current model selection procedure was based on the availability of selected biogeochemical variables across historical or projected scenarios. Based on this, CESM2 model utilizing the MARBL bgc model was excluded from our study due to the absence of the dissolved oxygen (o2) variable in its historical dataset.

\*\*\*\*\*\*\*\*\*\*\*\*\*\*\*\*\*\*\*\*\*\*\*\*\*\*\*\*\*\*\*\*\*\*\*\*\*\*\*

**Comment 10:**

L132 – Do you mean visualised using taylor diagram? How do you calculate model/data comparison using a diagram?

**Response:**

We sincerely apologize for the confusion caused by our statement. Upon a careful review, we replaced the word *"calculated"* to *"visualized"* in **L165**.

\*\*\*\*\*\*\*\*\*\*\*\*\*\*\*\*\*\*\*\*\*\*\*\*\*\*\*\*\*\*\*\*\*\*\*\*\*\*\*

**Comment 11:**

L164-165 Can you provide a reference on this statement?

**Response:**

Thank you for your concern regarding this matter. The statement is nuanced, reflecting both aspects. While some studies, such as those by Behrenfeld et al. (2006) and Kwiatkowski et al. (2017), indicate that seasonal cycles can help constrain projections, our intended message **(L197 – L201)** is that *"although temporal cycles like the yearly seasons are important components of climate variability, they provide only a partial perspective on long-term climate change. Long-term changes involve shifts in average temperatures, changes in precipitation patterns, variations in the frequency and intensity of extreme weather events, and other systemic transformations that go beyond the periodic nature of seasonal cycles."* Thus, our revised statement **(L197 – L201)** aims to convey that "*the yearly cycle of seasons partially captures the long-term changes associated with climate change"*.

\*\*\*\*\*\*\*\*\*\*\*\*\*\*\*\*\*\*\*\*\*\*\*\*\*\*\*\*\*\*\*\*\*\*\*\*\*\*\*

**Comment 12:**

L172 – but CMEMS data is not really observation, isn't it?

**Response:**

We sincerely apologize for the confusion caused by our statement. CMEMS is not observation data. We replaced the word *"observed"* to *"reference"* in **L208**.

\*\*\*\*\*\*\*\*\*\*\*\*\*\*\*\*\*\*\*\*\*\*\*\*\*\*\*\*\*\*\*\*\*\*\*\*\*\*\*

**Comment 13:**

L177-L179 - perhaps spell out how these models represent their phytoplankton growth and chlorophyll concentration? And compare it to models that have better RMSD?

**Response:**

Thank you for highlighting this concern. The model's representation of biological tracers was discussed in Inter-variable relations section in **L395** as detailed in **Reviewer-1 major comment 1**.

\*\*\*\*\*\*\*\*\*\*\*\*\*\*\*\*\*\*\*\*\*\*\*\*\*\*\*\*\*\*\*\*\*\*\*\*\*\*\*

**Comment 14:**

L184, 219 – what is acceptable range?

**Response:**

Thank you for highlighting this concern. We have addressed it by representing the acceptable bias range based on models with small mean bias. Accordingly, in the revised manuscript, we have specified the acceptable range as ≤ ±0.3 mg/m³ for chlorophyll in **L220** and ≤ ±1 mmol/m³ for phytoplankton in **L234**.
* * *
**Comment 15:**

L186 - why is UKESM not overestimating chlorophyll, but overestimates phytoplankton carbon?

**Response:**

Thank you for your concern regarding this matter. In **L225 – L229** we explained that *"UKESM1-0-LL model explicitly simulates chlorophyll concentrations, allowing for a more accurate representation of chlorophyll levels (Sellar et al., 2019). However, UKESM1-0-LL uses nitrogen as its primary model currency, which results in a more pronounced quantitative representation of nutrient levels. This might lead to enhanced nutrient uptake by phytoplankton due to differences in model parameterizations and consequently result in the overestimation of phytoplankton biomass."* This could explain why the model does not overestimate chlorophyll but does overestimate phytoplankton.
* * *
**Comment 16:**

L232-L242 – maybe move this to the study domain part instead of on the results section?

**Response:**

Thank you for your suggestion. Accordingly, we shifted this part to study domain section in **L109 – L119**.
* * *
**Comment 17:**

L251 – Can you give example of the important processes?

**Response:**

Thank you for your insightful comment. The important processes that may be overlooked by some ESMs include nutrient cycling, light availability, temperature variations, and phytoplankton phenology. These processes play important roles in shaping the seasonal patterns of biogeochemistry in marine ecosystems. We have clarified this point in the revised manuscript in **L390**.
* * *
**Comment 18:**

L286 - Why do you think this is? could it be that the ESMs in CMIP6 is developed based on the condition of the open ocean, but not the shelf seas? Or is it because the resolution is too coarse for shelf seas?

**Response:**

Thank you for your insightful question. Indeed, both factors you mentioned could contribute to the observed performance of the ESMs in simulating biogeochemical variables. The ESMs in CMIP6 are primarily developed based on the conditions of the open ocean, which may not fully capture the complexities of shelf seas. Additionally, the coarse resolution of these models may not adequately resolve the fine-scale processes occurring in shelf sea environments. Together, these factors likely contribute to the moderate to poor performance of the ESMs in simulating biogeochemical variables, as mentioned in **L502 – L506**.
* * *
**Comment 19:**

Figure 6 could do with higher resolution.

**Response:**
Thank you for your suggestion regarding Figure 6. Accordingly, we have resolved the clarity of the Figure, which is now **Fig. 12** (can be found in this report below).

\*\*\*\*\*\*\*\*\*\*\*\*\*\*\*\*\*\*\*\*\*\*\*\*\*\*\*\*\*\*\*\*\*\*\*\*

**Revised Figures:**

[Figure]

**Figure 1.** DJF spatial biases of surface chlorophyll for 13 individual models and model ensemble relative to reference.

[Figure]

**Figure 2.** Same as Fig.2 but for JJA.

[Figure]

**Figure 3.** The mean bias of surface chlorophyll for both seasons (DJF, JJA) and annual.

[Figure]

**Figure 4.** DJF spatial biases of surface phytoplankton for 13 individual models and model ensemble relative to reference.

[Figure]

**Figure 5.** Same as Fig. 5 but for JJA.

[Figure]

**Figure 6.** The mean bias of surface phytoplankton for both seasons (DJF, JJA) and annual.

[Figure]

**Figure 7.** DJF spatial biases of surface nitrate for 13 individual models and model ensemble relative to reference.

[Figure]

**Figure 8.** Same as Fig. 8 but for JJA.

[Figure]

**Figure 9.** The mean bias of surface nitrate for both seasons (DJF, JJA) and annual.

[Figure]

**Figure 10.** DJF spatial biases of nitrate at 70-meter depth for 13 individual models and model ensemble relative to reference.

[Figure]

**Figure 11.** Same as Fig. 11 but for JJA.

[Figure]

**Figure 12.** The mean bias of nitrate at 70-meter depth for both seasons (DJF, JJA) and annual.

[Figure]

**Figure 13.** DJF spatial biases of nitrate at 1000-meter depth for 13 individual models and model ensemble relative to reference.

[Figure]

**Figure 14.** Same as Fig. 14 but for JJA.

[Figure]

**Figure 15.** The mean bias of nitrate at 1000-meter depth for both seasons (DJF, JJA) and annual.

[Figure]

**Figure 16.** DJF spatial biases of surface oxygen for 13 individual models and model ensemble relative to reference.

[Figure]

**Figure 17.** Same as Fig. 17 but for JJA.

[Figure]

**Figure 18.** The mean bias of surface oxygen for both seasons (DJF, JJA) and annual.

[Figure]

**Figure 19.** DJF spatial biases of oxygen at 70-meter depth for 13 individual models and model ensemble relative to reference.

[Figure]

**Figure 20.** Same as Fig. 20 but for JJA.

[Figure]

**Figure 21.** The mean bias of oxygen at 70-meter depth for both seasons (DJF, JJA) and annual.

[Figure]

**Figure 22.** DJF spatial biases of oxygen at 1000-meter depth for 13 individual models and model ensemble relative to reference.

[Figure]

**Figure 23.** Same as Fig. 23 but for JJA.

[Figure]

**Figure 24.** The mean bias of oxygen at 1000-meter depth for both seasons (DJF, JJA) and annual.

[Figure]

**Figure 25.** Relationships between chlorophyll and phytoplankton for model ensemble and 13 individual models in southern South China Sea during the study period (1993 – 2014). Dashed red lines represent 95% confidence interval with 0.05 as the level of significance (α).

[Figure]

**Figure 26.** Relationships between nitrate and phytoplankton for model ensemble and 13 individual models in southern South China Sea during the study period (1993 – 2014). Dashed red lines represent 95% confidence interval with 0.05 as the level of significance (α).

[Figure]

**Figure 27.** Annual Taylor Diagram for (a) chlorophyll, (b) phytoplankton, (c) nitrate and (d) oxygen.

**Reference:**

Behrenfeld, M. J., O'Malley, R. T., Siegel, D. A., McClain, C. R., Sarmiento, J. L., Feldman, G. C., et al. (2006). Climate-driven trends in contemporary ocean productivity. *Nature*, *444*(7120), 752–755. https://doi.org/10.1038/nature05317

Chelton, D. B., Deszoeke, R. A., Schlax, M. G., El Naggar, K., & Siwertz, N. (1998). Geographical variability of the first baroclinic Rossby radius of deformation. *Journal of Physical Oceanography*, *28*(3). https://doi.org/10.1175/1520-0485(1998)028<0433:GVOTFB>2.0.CO;2

Friedrichs, M. A. M., Hood, R. R., & Wiggert, J. D. (2006). Ecosystem model complexity versus physical forcing: Quantification of their relative impact with assimilated Arabian Sea data. *Deep-Sea Research Part II: Topical Studies in Oceanography*, *53*(5–7). https://doi.org/10.1016/j.dsr2.2006.01.026

Glessmer, M. S., Oschlies, A., & Yool, A. (2008). Simulated impact of double-diffusive mixing on physical and biogeochemical upper ocean properties. *Journal of Geophysical Research: Oceans*, *113*(8). https://doi.org/10.1029/2007JC004455

Hajima, T., Watanabe, M., Yamamoto, A., Tatebe, H., Noguchi, M. A., Abe, M., et al. (2020a). Development of the MIROC-ES2L Earth system model and the evaluation of biogeochemical processes and feedbacks. *Geoscientific Model Development*, *13*(5), 2197–2244. https://doi.org/10.5194/gmd-13-2197-2020

Hajima, T., Watanabe, M., Yamamoto, A., Tatebe, H., Noguchi, M. A., Abe, M., et al. (2020b). Development of the MIROC-ES2L Earth system model and the evaluation of biogeochemical processes and feedbacks. *Geoscientific Model Development*, *13*(5). https://doi.org/10.5194/gmd-13-2197-2020

Jin, S., Wei, Z., Wang, D., & Xu, T. (2023). Simulated and projected SST of Asian marginal seas based on CMIP6 models. *Frontiers in Marine Science*, *10*. https://doi.org/10.3389/fmars.2023.1178974

Kwiatkowski, L., Torres, O., Bopp, L., Aumont, O., Chamberlain, M., Christian, J. R., et al. (2020). Twenty-first century ocean warming, acidification, deoxygenation, and upper-ocean nutrient and primary production decline from CMIP6 model projections. *Biogeosciences*, *17*(13), 3439–3470. https://doi.org/10.5194/bg-17-3439-2020

Laufkötter, C., John, J. G., Stock, C. A., & Dunne, J. P. (2017). Temperature and oxygen dependence of the remineralization of organic matter. *Global Biogeochemical Cycles*, *31*(7). https://doi.org/10.1002/2017GB005643

Mauritsen, T., Bader, J., Becker, T., Behrens, J., Bittner, M., Brokopf, R., et al. (2019a). Developments in the MPI-M Earth System Model version 1.2 (MPI-ESM1.2) and Its Response to Increasing CO2. *Journal of Advances in Modeling Earth Systems*, *11*(4). https://doi.org/10.1029/2018MS001400

Mauritsen, T., Bader, J., Becker, T., Behrens, J., Bittner, M., Brokopf, R., et al. (2019b). Developments in the MPI-M Earth System Model version 1.2 (MPI-ESM1.2) and Its Response to Increasing CO $_2$. *Journal of Advances in Modeling Earth Systems*, *11*(4), 998–1038. https://doi.org/10.1029/2018MS001400

Paulsen, H., Ilyina, T., Six, K. D., & Stemmler, I. (2017). Incorporating a prognostic representation of marine nitrogen fixers into the global ocean biogeochemical model HAMOCC. *Journal of Advances in Modeling Earth Systems*, *9*(1). https://doi.org/10.1002/2016MS000737

Petrik, C. M., Luo, J. Y., Heneghan, R. F., Everett, J. D., Harrison, C. S., & Richardson, A. J. (2022). Assessment and Constraint of Mesozooplankton in CMIP6 Earth System Models. *Global Biogeochemical Cycles*, *36*(11). https://doi.org/10.1029/2022GB007367

Séférian, R., Berthet, S., Yool, A., Palmiéri, J., Bopp, L., Tagliabue, A., et al. (2020). Tracking Improvement in Simulated Marine Biogeochemistry Between CMIP5 and CMIP6. *Current Climate Change Reports*. https://doi.org/10.1007/s40641-020-00160-0

Sellar, A. A., Jones, C. G., Mulcahy, J. P., Tang, Y., Yool, A., Wiltshire, A., et al. (2019). UKESM1: Description and Evaluation of the U.K. Earth System Model. *Journal of Advances in Modeling Earth Systems*, *11*(12), 4513–4558. https://doi.org/10.1029/2019MS001739

Sinha, B., Buitenhuis, E. T., Quéré, C. Le, & Anderson, T. R. (2010). Comparison of the emergent behavior of a complex ecosystem model in two ocean general circulation models. *Progress in Oceanography*, *84*(3–4). https://doi.org/10.1016/j.pocean.2009.10.003

Six, K. D., & Maier-Reimer, E. (1996). Effects of plankton dynamics on seasonal carbon fluxes in an ocean general circulation model. *Global Biogeochemical Cycles*, *10*(4). https://doi.org/10.1029/96GB02561

Stock, C. A., Dunne, J. P., Fan, S., Ginoux, P., John, J., Krasting, J. P., et al. (2020). Ocean Biogeochemistry in GFDL's Earth System Model 4.1 and Its Response to Increasing Atmospheric CO2. *Journal of Advances in Modeling Earth Systems*, *12*(10). https://doi.org/10.1029/2019MS002043

Swart, N. C., Cole, J. N. S., Kharin, V. V., Lazare, M., Scinocca, J. F., Gillett, N. P., et al. (2019). The Canadian Earth System Model version 5 (CanESM5.0.3). *Geoscientific Model Development*, *12*(11), 4823–4873. https://doi.org/10.5194/gmd-12-4823-2019

Tagliabue, A., Kwiatkowski, L., Bopp, L., Butenschön, M., Cheung, W., Lengaigne, M., & Vialard, J. (2021). Persistent Uncertainties in Ocean Net Primary Production Climate Change Projections at Regional Scales Raise Challenges for Assessing Impacts on Ecosystem Services. *Frontiers in Climate*, *3*. https://doi.org/10.3389/fclim.2021.738224

Thompson, B., Tkalich, P., Malanotte-Rizzoli, P., Fricot, B., & Mas, J. (2016). Dynamical and thermodynamical analysis of the South China Sea winter cold tongue. *Climate Dynamics*, *47*(5–6). https://doi.org/10.1007/s00382-015-2924-3

Tjiputra, J. F., Schwinger, J., Bentsen, M., Morée, A. L., Gao, S., Bethke, I., et al. (2020). Ocean biogeochemistry in the Norwegian Earth System Model version 2 (NorESM2). *Geoscientific Model Development*, *13*(5), 2393–2431. https://doi.org/10.5194/gmd-13-2393-2020

Wang, Z., Brickman, D., Greenan, B., Christian, J., DeTracey, B., & Gilbert, D. (2024). Assessment of Ocean Temperature Trends for the Scotian Shelf and Gulf of Maine Using 22 CMIP6 Earth System Models. *Atmosphere - Ocean*, *62*(1). https://doi.org/10.1080/07055900.2023.2264832

---

## Author Response (AR1)

Response to review of **"Evaluation of CMIP6 Models Performance in Simulating Historical Biogeochemistry across Southern South China Sea"**

**Manuscript egusphere-2024-72**

**Response to Anonymous Referees:**

We sincerely appreciate the time and effort invested by both the reviewers and the editor in evaluating our paper titled **"Evaluation of CMIP6 Models Performance in Simulating Historical Biogeochemistry across Southern South China Sea"** submitted for publication in Biogeosciences. We are grateful for the positive feedback and the insightful comments provided, which is detailed in this report and also in the upcoming revised manuscript. The majority of the suggestions put forth by the reviewers have been incorporated, and in the limited cases where we have not, we have provided a detailed description of the justification for each decision. For ease of reference, we have provided a detailed point-by-point response to the reviewer's comments with line numbers in the response representing the changes in revised manuscript.

*This report contains point-by-point responses to each comment from both reviewers.

**REVIEWER-1 MAJOR COMMENTS**

**Comment 1:**

An in-depth process evaluation is needed

First, irrespective from the valuable effort (and outcome) of this work, that is evaluated marine biogeochemistry models embedded in CMIP6 ESMs, I think further attention to other physical drivers or control variables would be needed to better understand the performance of the models.

Another need would be to scrutinize inter-parameters relationship (chl-biomass, biomass-nitrate). I think the reader could be interested in further understanding of biases propagation across the marine biogeochemical cycles.

A similar attention would be needed also to dive a bit further into the model process parameterization. For instance, not all model simulates prognostically chlorophyll. This latter is derived from the carbon-to-chlorophyll ratio and phytoplankton biomass (see Séférian et al. 2020, who did an in-depth evaluation of model parameterization).

**Response:**

- Thank you for highlighting the importance of analysing the physical drivers or control variables of models to enhance our understanding of their performance. In this paper, our primary focus is to assess the performance of CMIP6 models in reproducing historical biogeochemical variables. While we appreciate the suggestion to include analysis of physical drivers or control variables, we believe that expanding the scope in this direction may detract from the main focus of our research. Moreover, it's worth noting that previous studies such as Jin et al. (2023) and Fan & Zhou (2023) have already evaluated the performance of CMIP6 models on some physical drivers/control variables (i.e., SST, Asian-Pacific Oscillation and precipitation), including those relevant to our study domain (southern South China Sea) and common models utilized in our research (ACCESS-ESM1-5, CanESM-5, GFDL-ESM4, MIROC, MPI-ESM-1-2-HAM, MPI-ESM-1-2-HR, MPI-ESM-1-2-LR, NorESM2-LM, NorESM2-MM & MRI-ESM2-0). However, we acknowledge the value of conducting a separate analysis on the performance of individual model's physical drivers or control variables alongside biogeochemical responses in future research. This suggestion has been duly noted and included as a recommendation in our paper **(L575)**.

- Thank you for highlighting the importance of inter-parameters relationship. We also agree with your suggestion, and as it aligns with the scope of our study, we accordingly produced the correlation between Chlorophyll-Phytoplankton biomass **(L401)** and Nitrate-Phytoplankton biomass **(L431)**. As a result, our analysis revealed strong positive correlations between chlorophyll concentration and phytoplankton biomass in most CMIP6 models, indicating robust primary production and nutrient availability. However, we also observed deviations in some models, likely stemming from differences in parameterizations and structural complexities. These findings highlight the necessity of considering model uncertainties in our interpretations. Similarly, our examination of nitrate-phytoplankton relationships showed significant positive correlations in some CMIP6 models, while others displayed weaker correlations. This discrepancy potentially due to variations in biogeochemical tracers within model frameworks could influence model efficacy. The relevance of these discrepancies to their model dynamics is discussed in detail in **L421 to L460**.

- Thank you for bringing up the importance of model process parameterization. We acknowledge your point that not all models incorporate chlorophyll prognostically. However, delving deeply into model process parameterization might extend beyond the scope of our present study, which primarily aims at evaluating model reproducing skill on biogeochemical variables. Nevertheless, we recognize the potential value in comparing the parameterization processes of different models for future research. We suggest considering this as a separate study to explore the performance of these models more comprehensively in the future. Additionally, we have included this suggestion in our recommendations **(L575 – L582)**.

\*\*\*\*\*\*\*\*\*\*\*\*\*\*\*\*\*\*\*\*\*\*\*\*\*\*\*\*\*\*\*\*\*\*\*\*\*\*

**Comment 2:**

The choice of biogeochemical tracers

The authors focused on key biogeochemical variables: chlorophyll, plankton biomass, nitrate and oxygen. All of these variables are analyzed at surface oceans — which makes sense for biological markers such as chlorophyll and plankton biomass but a surface analysis can hide model biases for oxygen and nitrate (oxycline and nutricline). There an expanded analysis including nitrate and oxygen profiles across the multi-model ensemble would be relevant.

I also think that phosphate and net marine productivity would be relevant to liase with global studies such as Kwiatkowski et al. 2020.

**Response:**

- Thank you for bringing up the importance of analysing the bias at depth to understand the oxycline and nutricline dynamics of the models. In response, considering the complex bathymetry of southern SCS region, we have addressed this concern by presenting the spatial distribution of seasonal variations in nitrate and oxygen at two distinct depths (70m and 1000m) for each model, rather than providing profiles (**L277 – L325**). We have discussed the biases observed in each model and the ensemble, noting consistent negative bias in nitrate concentrations across models, with MPI-based models showing the least positive bias at 70m but shifting towards a negative bias at 1000m. Oxygen biases followed a similar trend, with UKESM1-0-LL displaying significant negative bias at 70m and positive bias at 1000m. Additionally, CanESM5 and MIROC-based models exhibited high negative biases at 1000m. These biases could potentially arise from inaccuracies in parameterizing relevant processes, exclusion of critical processes and structural uncertainties in model formulation. These discrepancies were discussed in detail at **L283 – L293** for nitrate and **L304 – L325** for oxygen. The depth of 70 meters has been selected to depict the dynamics of the nutricline/oxycline in the shelf break region, while a depth of 1000 meters has been chosen to represent the deep layer.

- We sincerely thank the reviewer for recommending the inclusion of additional biogeochemical parameters such as phosphate and net marine productivity, following the work of Kwiatkowski et al. 2020. However, after careful consideration, we choose to include only variables that were consistently available across all selected models' historical and projection scenarios. Since phosphate and net productivity variables were absent in some of the historical and projection scenarios, we prioritized models that shared common variables in both their historical and projection scenarios.

\*\*\*\*\*\*\*\*\*\*\*\*\*\*\*\*\*\*\*\*\*\*\*\*\*\*\*\*\*\*\*\*\*\*\*

**Comment 3:**

Choice of reference datasets

Finally I'm also concerned by the use of the single CMEMS data are taken as ground truth observations although they aren't.

As far as I am aware, CMEMS is a model-based data product. For the ocean physics the data product benefit from observation assimilation (it is thus a reanalysis) but for marine biogeochemistry it is only the marine biogeochemical model PISCES without any constraints.

I understand a focus on a given regional domain could pose challenge in terms of data availability. However, there are many high-resolution datasets for surface chlorophyll, net primary productivity etc. based on remote-sensing observation that can provide support to this work.

I would recommend to use them in addition to CMEMS. Indeed, past work (Lee et al. 2016) showed that this data product do not outcomes standard CMIP5 models when compared to true insitu observations.

**Response:**

- Thank you for your concern about the usage of reference data (CMEMS). Unfortunately, CMEMS is the only available timeseries hindcast data for the biogeochemistry in southern South China Sea region. As we stated already in the manuscript **(L137)** that CMEMS biogeochemistry product quality has been validated by Mercator-Ocean and they have confirmed and published the quality of this data through comparisons with recognized datasets like Ocean Color, World Ocean Atlas and Globcolour products in their Quality Information Document (QuID; Perruche et al., 2019).

- In order to improve more confidence of this dataset in southern South China Sea region, we have discussed some literatures in our revised manuscript **(L142 – L151)**, in which, authors have validated this data product with the in-situ measurement in this study region, i.e., Wahyudi et al, (2023) validated the POC, Chlorophyll, Dissolved Oxygen, Nitrate, Phosphate and Silicate obtained from CMEMS biogeochemistry product by comparing it with in-situ data collected during the Widya Nusantara Expedition 2015 (Triana et al., 2021) in the upwelling area of southwestern Sumatra waters. They found that the mean absolute percentage error values were lower than 15%, indicating the reliability of the CMEMS biogeochemistry model data in our study area. Additionally, Chen et al, (2023) also used the daily chlorophyll concentration data from the same CMEMS biogeochemical product in south china sea region. By utilizing this CMEMS biogeochemistry model dataset, Wahyudi et al. (2023) and Chen et al. (2023) highlights the proficiency of the CMEMS biogeochemistry model data in reproducing both the climatic patterns and fluctuations observed within its biogeochemical variables in southern South China Sea. This gave us confidence in utilizing the CMEMS biogeochemical dataset as the reference model to assess other models in this region (southern South China Sea).

- Thank you for recommending the other available dataset. Although there are other datasets available for surface chlorophyll and net primary productivity, the purpose of using the single reference dataset (CMEMS) is to ensure consistency in the evaluation process. By relying on a homogeneous dataset, we aim to enhance the reliability of our evaluation results and greater confidence in the findings.
* * *
**REVIEWER-1 MINOR COMMENTS**

**Comment 1:**

L9 why? they are poorly constrained

**Response:**
Thank you for bringing our oversight on explain this matter. The emphasis on selecting the biogeochemical variables is not primarily on constraint, but rather on highlighting the significance of key biogeochemical tracers and their availability across the chosen models and their corresponding projection scenarios. Accordingly, we have also explained this matter in our revised manuscript **(L08)**.
* * *
**Comment 2:**

L12 overestimations or underestimations, in what? (quantitative measures or score would be useful)

**Response:**
Thank you for bringing up this important point. We apologize for the oversight in not clarifying. The overestimations or underestimations refer to quantitative measures. We have incorporated this clarification into the revised manuscript as per your recommendation **(L12)**.
* * *
**Comment 3:**

L23-25 This statement is inexact, recent work from Kwiatkowski et al 2020 shows that marine NPP in 2100 remains largely uncertain

**Response:**
We acknowledge the inexactness of the statement regarding marine NPP uncertainty in 2100. Upon thorough examination of Kwiatkowski et al.'s (2020) findings, we recognize that the uncertainty in primary production largely persists, even within the CMIP6 framework. To address this, we have diligently incorporated the relevant insights from Kwiatkowski et al. (2020) into our revised manuscript **(L24 – L30)**, thereby ensuring the accuracy of our discussion on this matter.

\*\*\*\*\*\*\*\*\*\*\*\*\*\*\*\*\*\*\*\*\*\*\*\*\*\*\*\*\*\*\*\*\*\*\*\*\*

**Comment 4:**

L47 Consider to refer to Séférian et al. 2020 for the evaluation of global ESMs

**Response:**
Thank you for your valuable suggestion to consider Séférian et al. (2020). After careful review, we agree that their work on global ESMs evaluation is relevant to our study. Thus, we have incorporated a citation to Séférian et al. (2020) in our revised manuscript **(L56)**.

\*\*\*\*\*\*\*\*\*\*\*\*\*\*\*\*\*\*\*\*\*\*\*\*\*\*\*\*\*\*\*\*\*\*\*\*\*

**Comment 5:**

L58 This aspect has to be analysed with caveats because for several bgc models chlorophyl is derived from phytoplankton biomass improving linkage with zooplankton response (see major comments)

**Response:**
We sincerely appreciate your input on this matter. However, in this particular context, our aim was to convey the findings of Petrik et al. (2022), who extensively examined the model's simulation of zooplankton and delved into the relationship between zooplankton and chlorophyll-a. It is worth noting, as we have highlighted in our revised manuscript **(L65)**, that Petrik et al. (2022) underscored "*chlorophyll-a as a proxy for phytoplankton*" for several reasons: (a) it provides the most comprehensive observations across both time and space, (b) it is more readily observable compared to phytoplankton biomass, which typically requires physical sampling, or primary production, which relies on physical experiments or algorithms based on satellite chlorophyll data that yield varied outcomes, and few other reasons.

\*\*\*\*\*\*\*\*\*\*\*\*\*\*\*\*\*\*\*\*\*\*\*\*\*\*\*\*\*\*\*\*\*\*\*\*\*

**Comment 6:**

L102 it would be nice to have a table of the references datasets used here
L112 CMEMS : it might be good to make it clear if CMEMS data is reanalysis and model reconstruction.
As far as I am aware of CMEMS relies on NEMO-PISCES+ assimilation for ocean hydrodynamics. PISCES, the marine bgc model adjust to improved physics but it is free (no assimilation). Therefore using CMEMS for evaluating marine bgc models is comparing models with another (more constrained) model.

**Response:**
We sincerely appreciate your thoughtful consideration of this matter and your concern. The utilization of the CMEMS biogeochemical dataset in our study is primarily due to its status as the only available timeseries hindcast biogeochemistry dataset for our study domain, specifically the southern South China Sea region. Furthermore, we have referenced literature that has validated this data product for this region through comparisons with in-situ measurements, instilling confidence in its use for evaluating the models in this area. We kindly wish to inform you that a detailed response addressing this specific issue was provided in our response to **Reviewer-1 major comment 3**.

\*\*\*\*\*\*\*\*\*\*\*\*\*\*\*\*\*\*\*\*\*\*\*\*\*\*\*\*\*\*\*\*\*\*\*\*\*

**Comment 7:**

L125 Table 1: I'm suprised to see PISCES (IPSL, CNRM, EC-Earth) and MARBL (CESM2) excluded from this exhaustive analysis — what are the reason?

**Response:**

We appreciate your attention to this issue. As explained in **L132** that our model selection procedure was based on the availability of selected biogeochemical variables across historical or projected scenarios. Consequently, the EC-Earth models do not include the Phytoplankton biomass (phyc) variable, while CESM2 lacks the dissolved Oxygen (o2) variable in its historical dataset. Additionally, the CNRM and IPSL-based models showed a standard deviation >50 mg/m3 for the chlorophyll variable compared to reference data, resulting in their exclusion from the analysis.

\*\*\*\*\*\*\*\*\*\*\*\*\*\*\*\*\*\*\*\*\*\*\*\*\*\*\*\*\*\*\*\*\*\*\*\*\*\*

**Comment 8:**

L129 Ranking: no cross-variable evaluation? SST-Chlorophyll, etc. (see major comments)

**Response:**

We appreciate your emphasis on evaluating the model's biogeochemistry in relation to its physical drivers. However, the primary focus of our paper is the assessment of CMIP6 biogeochemical model outcomes. While we acknowledge the significance of examining biomass correlations with physical drivers, delving into this aspect might deviate from the central goal of our paper. Instead, we have thoroughly analysed the model's performance concerning Chlorophyll-Phytoplankton biomass and Nitrate-Phytoplankton biomass correlations. We kindly wish to highlight that we addressed this concern in detail in our response to **Reviewer-1 major comment 1**.

\*\*\*\*\*\*\*\*\*\*\*\*\*\*\*\*\*\*\*\*\*\*\*\*\*\*\*\*\*\*\*\*\*\*\*\*\*\*

**Comment 9:**

L159-160: please consider sharing scripts and data for this work (EGU journals recommandations)

**Response:**

Thank you for your thoughtful consideration of our work and for your interest in the data. We have indeed provided the source link to the dataset we utilized **(L156)** from CMEMS biogeochemistry hindcast dataset (ID: GLOBAL_MULTIYEAR_BGC_001_029) and CMIP6 dataset which was obtained from ESGF data portal, which are freely accessible for public use. Furthermore, we employed widely accepted basic formulae for our statistical analyses, as detailed in our manuscript **(L73 – L81)**.

\*\*\*\*\*\*\*\*\*\*\*\*\*\*\*\*\*\*\*\*\*\*\*\*\*\*\*\*\*\*\*\*\*\*\*\*\*\*

**Comment 10:**

L164-165 yes and no there are several paper that indicates how seasonal cycle can help to constrain projections (Behrenfeld et al. 2006, Kwiatkovski et al. 2017, etc..)

**Response:**

Thank you for bringing this matter to our attention. We sincerely apologize for any confusion caused by our previous statement. Upon careful review of the works by Behrenfeld et al. (2006) and Kwiatkovski et al. (2017), we have revised our statement accordingly. Instead of asserting that *"the yearly cycle of seasons does not fully capture the long-term changes associated with climate change,"* we have amended it to reflect that *"the yearly cycle of seasons partially captures the long-term changes associated with climate change"*. These long-term changes encompass shifts in average temperatures, alterations in precipitation patterns, changes in the frequency and intensity of extreme weather events, and other systemic shifts that extend beyond the periodicity of seasonal cycles. While temporal cycles are indeed important components of climate variability, they offer only a partial perspective on the broader and more profound changes occurring in the Earth's climate system. Thus, our revised statement in **L198 – L202** aims to convey that "*the yearly cycle of seasons partially captures the long-term changes associated with climate change"*.

************************************

**Comment 11:**

L183: typo: ensemble mean ?
'Moderate' needs to be quantified with skillscore

**Response:**

Thank you for pointing out this oversight. We have corrected the typo to *"ensemble"* in the manuscript. Furthermore, we've also provided clarification on the term *"moderate"* by delineating its specific meaning in context. In the discussion of spatial variation and bias in **L220**, *"moderate"* denotes the model's performance at an average level in simulating the variables quantitatively, relative to the reference data. Likewise, in reference to Taylor's diagram in **L481**, *"moderate"* indicates the model's placement between the last and first models in TD.

************************************

**Comment 12:**

Figures 2-5: hard to see model differences.
I would present model baises against the reference

**Response:**

We sincerely appreciate your insightful feedback. We acknowledge your concern regarding the clarity of the model differences in Figures 2-5 in old manuscript. In response to your suggestion, we have changed the seasonal climatology figures to illustrate the seasonal bias against the reference data in our revised manuscript **(Figs. 2–25)**.

************************************

**Comment 13:**

Figure 7: why for surface oxygen of MPI-based models outbest the others whereas it is not the case for the other variables ? For these later MIROC-ES2L outbests the other models.
These differences needs to be explained. My guess is that SST biaises in MPI-based models are much lower than the other models in this zone which explains why surface oxygen is better represented.

**Response:**

Thank you for bringing to our knowledge about the differences in model out best behaviour and also acknowledge for providing us the solution as well. Following your suggestion, we have conducted a thorough review of some literature. Our findings align with your suggestion and discussed in our revised manuscript **(L535 – L542)** that MPI-based models exhibit superior performance in simulating oxygen levels in our study, largely attributed to their effective physical drivers. Research conducted by Jin et al. (2023) verifies this by demonstrating that MPI-based models excel in replicating SST patterns, particularly during boreal winter and summer. They are noted for their minimal SST biases, particularly in Asian marginal seas, which contributes to their accuracy in representing surface oxygen variables in our study.

************************************

**REVIEWER-2 GENERAL COMMENTS**

**Comment 1A:**

The aim of the paper is to rank 13 CMIP6 ESM simulations based on their ability to reproduce selected observed biogeochemical variables. However, the dataset that the author chose is not strictly observations. Based on the link they provided in line 123, the CMEMS ocean biogeochemistry product is based on the PISCES model output (although it is forced with reanalysis product). I also noticed that among the 13 CMIP6 ESMs, the authors have not chosen IPSL-CM6A ESM, which includes PISCES as its ocean biogeochemical model. I understand that in-situ observations may be rare in this region, but to truly assess the CMIP6 ensemble and individual models, I suggest the authors could compare the CMIP6 models with satellite-derived chlorophyll-a and primary production, as well as the World Ocean Atlas product for nitrate and oxygen.

**Response:**

- Thank you for your concern about the usage of reference data (CMEMS). Unfortunately, CMEMS is the only available timeseries hindcast data for the biogeochemistry in southern South China Sea region. As we stated already in the manuscript **(L137)** that CMEMS biogeochemistry product quality has been validated by Mercator-Ocean and they have confirmed and published the quality of this data through comparisons with recognized datasets like Ocean Color, World Ocean Atlas and Globcolour products in their Quality Information Document (QuID; Perruche et al., 2019).

- In order to improve more confidence of this dataset in southern South China Sea region, we have discussed some literatures in our revised manuscript **(L142 – L151)**, in which, authors have validated this data product with the in-situ measurement in this study region, i.e., Wahyudi et al, (2023) validated the POC, Chlorophyll, Dissolved Oxygen, Nitrate, Phosphate and Silicate obtained from CMEMS biogeochemistry product by comparing it with in-situ data collected during the Widya Nusantara Expedition 2015 (Triana et al., 2021) in the upwelling area of southwestern Sumatra waters. They found that the mean absolute percentage error values were lower than 15%, indicating the reliability of the CMEMS biogeochemistry model data in our study area. Additionally, Chen et al, (2023) also used the daily chlorophyll concentration data from the same CMEMS biogeochemical product in south china sea region. By utilizing this CMEMS biogeochemistry model dataset, Wahyudi et al. (2023) and Chen et al. (2023) highlights the proficiency of the CMEMS biogeochemistry model data in reproducing both the climatic patterns and fluctuations observed within its biogeochemical variables in southern South China Sea. This gave us confidence in utilizing the CMEMS biogeochemical dataset as the reference model to assess other models in this region (southern South China Sea).

- While we appreciate your suggestion of alternative datasets, our decision to utilize CMEMS as the sole reference dataset was made to maintain consistency in the evaluation process. By adhering to a homogeneous dataset, we aim to ensure the integrity and reliability of our evaluation results, thereby instilling greater confidence in our findings.

- Additionally, IPSL-based models showed a standard deviation $>50$ mg/m$^3$ for the chlorophyll variable compared to reference data, resulting in their exclusion from the analysis.

**Comment 1B:**

Since the paper also looks at the seasonal trend of biogeochemical properties, it could benefit from exploring whether different CMIP6 models can capture phytoplankton phenology (e.g., Racault et al., 2015; Gittings et al., 2018), which is an important indicator.

**Response:**

- We appreciate your insightful comment and apologize for any confusion regarding our approach. Our examination primarily focused on the seasonal spatial climatology (now presented as seasonal spatial bias in **Figs. 2-25**), not the seasonal trend map.

- While we acknowledge the significance of exploring phytoplankton phenology in CMIP6 models, as suggested by the references you provided, conducting such studies requires extensive time and resources and also, we afraid that incorporating phenology in this study could potentially diverge the main scope of our current investigation. However, we recognize the importance of this aspect and have duly noted it as a potential avenue for future research in our study **(L459 – L463)**. Thank you for bringing this to our attention.

\*\*\*\*\*\*\*\*\*\*\*\*\*\*\*\*\*\*\*\*\*\*\*\*\*\*\*\*\*\*\*\*\*\*\*\*\*

**Comment 2:**

Indeed, most of the biological activity occurs near the surface layers of the ocean, but it's important to consider the biogeochemical dynamics near the seabed, particularly in shelf seas, as they can have complex structures through interactions of ocean physics with biological processes, such as export and remineralization. I would appreciate the inclusion of depth profiles and benthic concentrations of oxygen and nitrate – this would provide a more thorough assessment of the biogeochemical properties. Furthermore, most of the biogeochemical models used in CMIP6 are not specifically built for shelf seas. It would be interesting to see whether these models can represent nutrient and oxygen distribution at shallower depths.

**Response:**

- Thank you for bringing up the importance of analysing the bias at depth to understand the oxycline and nutricline dynamics of the models. In response, considering the complex bathymetry of southern SCS region, we have addressed this concern by presenting the spatial distribution of seasonal variations in nitrate and oxygen at two distinct depths (70m and 1000m) for each model, rather than providing profiles **(L277 – L325)**. We have discussed the biases observed in each model and the ensemble, noting consistent negative bias in nitrate concentrations across models, with MPI-based models showing the least positive bias at 70m but shifting towards a negative bias at 1000m. Oxygen biases followed a similar trend, with UKESM1-0-LL displaying significant negative bias at 70m and positive bias at 1000m. Additionally, CanESM5 and MIROC-based models exhibited high negative biases at 1000m. These biases could potentially arise from inaccuracies in parameterizing relevant processes, exclusion of critical processes and structural uncertainties in model formulation. These discrepancies were discussed in detail at **L283 – L293** for nitrate and **L304 – L325** for oxygen. The depth of 70 meters has been selected to depict the dynamics of the nutricline/oxycline in the shelf break region, while a depth of 1000 meters has been chosen to represent the deep layer.

\*\*\*\*\*\*\*\*\*\*\*\*\*\*\*\*\*\*\*\*\*\*\*\*\*\*\*\*\*\*\*\*\*\*\*\*\*

**Comment 3:**

Although the authors put a great effort in evaluating CMIP6 model outputs, the model structures could also be evaluated; how biogeochemical tracers are represented, and whether these representations affect the performance of the model in the southern SCS. Perhaps the authors can add another table which biogeochemical tracers these models represent (e.g., in MEDUSA-2 (UKESM), it does not represent diazotrophic phytoplankton, explicitly calculates phytoplankton chlorophyll, and uses N as model currency, while in OECO-2 (MIROC), it has diazotrophic phytoplankton with C as model currency and includes Phosphate as nutrients), and perhaps also how they are formulated, especially when it involves trophic transfer (e.g. nutrient uptake, zooplankton grazing, and phytoplankton growth, and plankton mortality). These additions can add some discussion on how model representation (and structure) may affect model performance in the shelf seas, instead of repeatedly saying that underestimation/overestimation is due to zooplankton grazing/phytoplankton productivity/nutrient uptake.

**Response:**

We sincerely appreciate your insightful comments regarding the model structures. Following your suggestion, we have incorporated an overview of how tracers and model structure affect performance into our discussion, specifically when examining inter-parameter relationships such as chlorophyll-biomass and biomass-nitrate. We explained that most models, except for UKESM1-0-LL and MIROC-based models, use carbon as the primary

measure of phytoplankton biomass and include nitrate and phosphate to constrain growth rates, resulting in a weaker correlation with nitrate. The MPI-ESM1-2-LR model has a new nitrogen-fixing formulation that better captures the nitrogen response. The UKESM1-0-LL model uses nitrogen as its primary measure, leading to a stronger response to nitrate. The MIROC-ES2L model includes the phosphorus cycle to strongly depict the phosphorus limitation on diazotrophic phytoplankton growth, which could explain the negative correlation with nitrate. For the GFDL-ESM4 model, the negative correlation could be due to its categorization of phytoplankton by size and nutrient content, attributing specific N:P ratios. This summarized version is discussed in detail at **L437 – L459** in revised manuscript.
* * *
**Comment 4:**

The presentation of the results can also be improved. I think it will be easier to follow the results if the authors describe the observed distribution of nitrate, chlorophyll, phytoplankton biomass, and oxygen, then compare them with the model. For the figures, it would be more interesting to see the difference between the CMEMS data and CMIP6 outputs with better figure resolution (especially figure 6). Additional discussion on regions where bias usually occurs in different models will also be interesting (e.g., the shelf seas between Sumatra, the Malaysian peninsula, and Borneo are always high in phytoplankton biomass for UKESM, CanESM5, ACCESS, MPI-ESM1-2, NorESM2).

**Response:**

- We sincerely appreciate your insightful feedback and acknowledge your concern regarding the clarity of the results. Following your suggestion, we have modified the seasonal climatology figure in our revised manuscript to better illustrate the seasonal bias against the reference data in **Figs. 2 – 25**.

- Additionally, we have incorporated your recommendation to discuss the spatial diversity in model bias. Our revised manuscript now elaborates this in **L236 – L251**.
* * *
**REVIEWER-2 SPECIFIC COMMENTS**

**Comment 1:**

L12 – perhaps the authors can add a % or number on the degrees of overestimation and underestimation.

**Response:**
Thank you for bringing up this important point. We apologize for the oversight in not clarifying. The overestimations or underestimations refer to quantitative measures. We have incorporated this clarification into the revised manuscript as per your recommendation **(L12)**.
* * *
**Comment 2:**

L22-23 - Based on CMIP6 models, NPP trend is uncertain, apart maybe at the Southern Ocean (Tagliabue et al., 2021)

**Response:**
Thank you for your valuable feedback. We acknowledge the inaccuracy in our original statement regarding marine NPP uncertainty in 2100. After thoroughly reviewing the findings of Tagliabue et al. (2021) and Kwiatkowski et al. (2020), we recognize that significant uncertainty in primary production indeed persists, even within the CMIP6 framework. To address this, we have incorporated the insights from Kwiatkowski et al. (2020) and Tagliabue et al. (2021) into our revised manuscript **(L24 – L30)** to ensure a accurate discussion on this matter.
* * *
**Comment 3:**

L33-34 - This is not always the case - OBGC models can give the seemingly good representation of historical climate pattern but for the wrong reason. Furthermore, OBGC model results is dependent on its physical forcings (see Sinha et al., 2010)

**Response:**

Thank you for bringing this matter to our attention. In response to your suggestion, we have gone through some literatures and made changes accordingly in **L36 – L43**.

\*\*\*\*\*\*\*\*\*\*\*\*\*\*\*\*\*\*\*\*\*\*\*\*\*\*\*\*\*\*\*\*\*\*\*\*\*

**Comment 4:**

L60 – typo: Tjiputra et **al**, (2020)

**Response:**

Thank you for pointing out this oversight. We have corrected accordingly in **L69**

\*\*\*\*\*\*\*\*\*\*\*\*\*\*\*\*\*\*\*\*\*\*\*\*\*\*\*\*\*\*\*\*\*\*\*\*\*

**Comment 5:**

L61-72 – I'm not so sure if these are appropriate examples. Maybe add studies like Kwiatkowski et al., 2020, Hinrichs et al., 2023

**Response:**

Thank you for your valuable suggestion and for providing literature on this topic. We have thoroughly reviewed the references you recommended, and we have incorporated relevant findings from Kwiatkowski et al., 2020 into our manuscript **L79**.

\*\*\*\*\*\*\*\*\*\*\*\*\*\*\*\*\*\*\*\*\*\*\*\*\*\*\*\*\*\*\*\*\*\*\*\*\*

**Comment 6:**

L83-L85 – Why only phytoplankton, chlorophyll, nitrogen, and oxygen? Why not net primary production and or carbon?

**Response:**

Thank you for your concern about the selected variables. We chose to focus on phytoplankton, chlorophyll, nitrogen, and oxygen for following reasons:

1. These variables are fundamental tracers for biological and nutrient dynamics in ocean systems.

2. Phytoplankton and chlorophyll variables serve as effective proxies for primary production, thus encompassing the essential aspect of net primary production.

3. Given that the southern SCS region is recognized as a typical oligotrophic area where primary productivity is primarily constrained by nutrient availability, we specifically included nitrate and oxygen.

4. We considered variables that were consistently available across all selected models' historical and projection scenarios to ensure comparability and consistency in our analysis. Even phosphate variable is unavailable in some of the selected model's scenarios. Thus, phosphate is also excluded.

\*\*\*\*\*\*\*\*\*\*\*\*\*\*\*\*\*\*\*\*\*\*\*\*\*\*\*\*\*\*\*\*\*\*\*\*\*

**Comment 7:**

L103-L105 - This sounds like phytoplankton is controlling the physical biogeochemical process?

**Response:**

We sincerely apologize for the confusion caused by our statement. Upon a careful review, we rephrased the statement as *"Within the southern SCS, extensive observations have demonstrated that phytoplankton growth, serving as the primary source of organic matter, significantly influences oceanic carbon cycles. This growth is*

*influenced by monsoon-driven physical and biogeochemical processes, with phytoplankton demonstrating a notable sensitivity to these environmental dynamics."* in **L125 – L128**.

\*\*\*\*\*\*\*\*\*\*\*\*\*\*\*\*\*\*\*\*\*\*\*\*\*\*\*\*\*\*\*\*\*\*\*\*\*

**Comment 8:**

L122-123 – is this the hindcast global ocean biogeochemistry? Do you also use the GlobColour for chlorophyll? Please be more specific.

**Response:**
- We sincerely apologize for the confusion made. The CMEMS product used in this study is the hindcast global ocean biogeochemistry dataset, which can be found in CMEMS biogeochemistry hindcast dataset (ID: GLOBAL_MULTIYEAR_BGC_001_029) **(L156).**

- We did not use GlobColour data in our study. Instead, Mercator Ocean used GlobColour data to validate the chlorophyll data within the CMEMS hindcast biogeochemistry dataset, as mentioned in **L139 – L142**.

\*\*\*\*\*\*\*\*\*\*\*\*\*\*\*\*\*\*\*\*\*\*\*\*\*\*\*\*\*\*\*\*\*\*\*\*\*

**Comment 9:**

L125 – Perhaps, instead of having 2/3 ESMs with the same OBGC model, maybe choose one of them instead, so you can also look at other models such as PISCESv2 (Aumont et al., 2016), MARBL (Long et al., 2021), BFM5.2 (Lovato et al., 2022)?

**Response:**
Thank you for suggesting a method to choose models. We will incorporate this method in our future studies. Additionally, as explained in **L132** that our current model selection procedure was based on the availability of selected biogeochemical variables across historical or projected scenarios. Based on this, CESM2 model was excluded from our study due to the absence of the dissolved oxygen (o2) variable and EC-Earth models do not include the phytoplankton biomass (phyc) variable, in their historical dataset.

\*\*\*\*\*\*\*\*\*\*\*\*\*\*\*\*\*\*\*\*\*\*\*\*\*\*\*\*\*\*\*\*\*\*\*\*\*

**Comment 10:**

L132 – Do you mean visualised using taylor diagram? How do you calculate model/data comparison using a diagram?

**Response:**
We sincerely apologize for the confusion caused by our statement. Upon a careful review, we replaced the word *"calculated"* to *"visualized"* in **L166**.

\*\*\*\*\*\*\*\*\*\*\*\*\*\*\*\*\*\*\*\*\*\*\*\*\*\*\*\*\*\*\*\*\*\*\*\*\*

**Comment 11:**

L164-165 Can you provide a reference on this statement?

**Response:**
Thank you for bringing this matter to our attention. We sincerely apologize for any confusion caused by our previous statement. Upon careful review of the works by Behrenfeld et al. (2006) and Kwiatkovski et al. (2017), we have revised our statement accordingly. Instead of asserting that *"the yearly cycle of seasons does not fully capture the long-term changes associated with climate change,"* we have amended it to reflect that *"the yearly cycle of seasons partially captures the long-term changes associated with climate change"*. These long-term changes encompass shifts in average temperatures, alterations in precipitation patterns, changes in the frequency and intensity of extreme weather events, and other systemic shifts that extend beyond the periodicity of seasonal cycles. While temporal cycles are indeed important components of climate variability, they offer only a partial perspective on the broader and more profound changes occurring in the Earth's climate system. Thus, our revised

statement in **L198 – L202** aims to convey that "*the yearly cycle of seasons partially captures the long-term changes associated with climate change*".

\*\*\*\*\*\*\*\*\*\*\*\*\*\*\*\*\*\*\*\*\*\*\*\*\*\*\*\*\*\*\*\*\*\*\*\*\*\*

**Comment 12:**

L172 – but CMEMS data is not really observation, isn't it?

**Response:**

We sincerely apologize for the confusion caused by our statement. CMEMS is not observation data. We replaced the word *"observed"* to *"reference"* in our revised manuscript **L209**.

\*\*\*\*\*\*\*\*\*\*\*\*\*\*\*\*\*\*\*\*\*\*\*\*\*\*\*\*\*\*\*\*\*\*\*\*\*\*

**Comment 13:**

L177-L179 - perhaps spell out how these models represent their phytoplankton growth and chlorophyll concentration? And compare it to models that have better RMSD?

**Response:**

Thank you for highlighting this concern. The model's representation of biological tracers was discussed in Inter-variable relations section in **L400**.

\*\*\*\*\*\*\*\*\*\*\*\*\*\*\*\*\*\*\*\*\*\*\*\*\*\*\*\*\*\*\*\*\*\*\*\*\*\*

**Comment 14:**

L184, 219 – what is acceptable range?

**Response:**

Thank you for highlighting this concern. We have addressed it by representing the acceptable bias range based on models with small mean bias error. Accordingly, in the revised manuscript, we have specified the acceptable range as $\leq \pm 0.15$ mg/m³ for chlorophyll in **L221** and $\leq \pm 0.4$ mmol/m³ for phytoplankton in **L235**.

\*\*\*\*\*\*\*\*\*\*\*\*\*\*\*\*\*\*\*\*\*\*\*\*\*\*\*\*\*\*\*\*\*\*\*\*\*\*

**Comment 15:**

L186 - why is UKESM not overestimating chlorophyll, but overestimates phytoplankton carbon?

**Response:**

Thank you for your concern regarding this matter. In **L226 – L231** we explained that *"UKESM1-0-LL model explicitly simulates chlorophyll concentrations, allowing for a more accurate representation of chlorophyll levels (Sellar et al., 2019). However, UKESM1-0-LL uses nitrogen as its primary model currency, which results in a more pronounced quantitative representation of nutrient levels. This might lead to enhanced nutrient uptake by phytoplankton due to differences in model parameterizations and consequently result in the overestimation of phytoplankton biomass."* This could explain why the model does not overestimate chlorophyll but does overestimate phytoplankton.

\*\*\*\*\*\*\*\*\*\*\*\*\*\*\*\*\*\*\*\*\*\*\*\*\*\*\*\*\*\*\*\*\*\*\*\*\*\*

**Comment 16:**

L232-L242 – maybe move this to the study domain part instead of on the results section?

**Response:**

Thank you for your suggestion. Accordingly, we shifted this part to study domain section in **L110 – L120**.

\*\*\*\*\*\*\*\*\*\*\*\*\*\*\*\*\*\*\*\*\*\*\*\*\*\*\*\*\*\*\*\*\*\*\*\*\*\*

**Comment 17:**

L251 – Can you give example of the important processes?

**Response:**
Thank you for your insightful comment. The important processes that may be overlooked by some ESMs include nutrient cycling, light availability, temperature variations, and phytoplankton phenology. These processes play important roles in shaping the seasonal patterns of biogeochemistry in marine ecosystems. We have clarified this point in the revised manuscript in **L395**.
* * *
**Comment 18:**

L286 - Why do you think this is? could it be that the ESMs in CMIP6 is developed based on the condition of the open ocean, but not the shelf seas? Or is it because the resolution is too coarse for shelf seas?

**Response:**
Thank you for your insightful question. Indeed, both factors you mentioned could contribute to the observed performance of the ESMs in simulating biogeochemical variables. The ESMs in CMIP6 are primarily developed based on the conditions of the open ocean, which may not fully capture the complexities of shelf seas. Additionally, the coarse resolution of these models may not adequately resolve the fine-scale processes occurring in shelf sea environments. Together, these factors likely contribute to the moderate to poor performance of the ESMs in simulating biogeochemical variables, as mentioned in **L507 – L511**.
* * *
**Comment 19:**

Figure 6 could do with higher resolution.

**Response:**
Thank you for your suggestion regarding Figure 6. Accordingly, we have resolved the clarity of the Figure, which is now **Fig. 28**.
* * *
**Reference:**

Behrenfeld, M. J., O'Malley, R. T., Siegel, D. A., McClain, C. R., Sarmiento, J. L., Feldman, G. C., et al. (2006). Climate-driven trends in contemporary ocean productivity. *Nature*, *444*(7120), 752–755. https://doi.org/10.1038/nature05317

Chelton, D. B., Deszoeke, R. A., Schlax, M. G., El Naggar, K., & Siwertz, N. (1998). Geographical variability of the first baroclinic Rossby radius of deformation. *Journal of Physical Oceanography*, *28*(3). https://doi.org/10.1175/1520-0485(1998)028<0433:GVOTFB>2.0.CO;2

Friedrichs, M. A. M., Hood, R. R., & Wiggert, J. D. (2006). Ecosystem model complexity versus physical forcing: Quantification of their relative impact with assimilated Arabian Sea data. *Deep-Sea Research Part II: Topical Studies in Oceanography*, *53*(5–7). https://doi.org/10.1016/j.dsr2.2006.01.026

Glessmer, M. S., Oschlies, A., & Yool, A. (2008). Simulated impact of double-diffusive mixing on physical and biogeochemical upper ocean properties. *Journal of Geophysical Research: Oceans*, *113*(8). https://doi.org/10.1029/2007JC004455

Hajima, T., Watanabe, M., Yamamoto, A., Tatebe, H., Noguchi, M. A., Abe, M., et al. (2020a). Development of the MIROC-ES2L Earth system model and the evaluation of biogeochemical processes and feedbacks. *Geoscientific Model Development*, *13*(5), 2197–2244. https://doi.org/10.5194/gmd-13-2197-2020

Hajima, T., Watanabe, M., Yamamoto, A., Tatebe, H., Noguchi, M. A., Abe, M., et al. (2020b). Development of the MIROC-ES2L Earth system model and the evaluation of biogeochemical processes and feedbacks. *Geoscientific Model Development*, *13*(5). https://doi.org/10.5194/gmd-13-2197-2020

Jin, S., Wei, Z., Wang, D., & Xu, T. (2023). Simulated and projected SST of Asian marginal seas based on CMIP6 models. *Frontiers in Marine Science*, *10*. https://doi.org/10.3389/fmars.2023.1178974

Kwiatkowski, L., Torres, O., Bopp, L., Aumont, O., Chamberlain, M., Christian, J. R., et al. (2020). Twenty-first century

ocean warming, acidification, deoxygenation, and upper-ocean nutrient and primary production decline from CMIP6 model projections. *Biogeosciences*, *17*(13), 3439–3470. https://doi.org/10.5194/bg-17-3439-2020

Laufkötter, C., John, J. G., Stock, C. A., & Dunne, J. P. (2017). Temperature and oxygen dependence of the remineralization of organic matter. *Global Biogeochemical Cycles*, *31*(7). https://doi.org/10.1002/2017GB005643

Mauritsen, T., Bader, J., Becker, T., Behrens, J., Bittner, M., Brokopf, R., et al. (2019a). Developments in the MPI-M Earth System Model version 1.2 (MPI-ESM1.2) and Its Response to Increasing CO2. *Journal of Advances in Modeling Earth Systems*, *11*(4). https://doi.org/10.1029/2018MS001400

Mauritsen, T., Bader, J., Becker, T., Behrens, J., Bittner, M., Brokopf, R., et al. (2019b). Developments in the MPI-M Earth System Model version 1.2 (MPI-ESM1.2) and Its Response to Increasing CO $_2$. *Journal of Advances in Modeling Earth Systems*, *11*(4), 998–1038. https://doi.org/10.1029/2018MS001400

Paulsen, H., Ilyina, T., Six, K. D., & Stemmler, I. (2017). Incorporating a prognostic representation of marine nitrogen fixers into the global ocean biogeochemical model HAMOCC. *Journal of Advances in Modeling Earth Systems*, *9*(1). https://doi.org/10.1002/2016MS000737

Petrik, C. M., Luo, J. Y., Heneghan, R. F., Everett, J. D., Harrison, C. S., & Richardson, A. J. (2022). Assessment and Constraint of Mesozooplankton in CMIP6 Earth System Models. *Global Biogeochemical Cycles*, *36*(11). https://doi.org/10.1029/2022GB007367

Séférian, R., Berthet, S., Yool, A., Palmiéri, J., Bopp, L., Tagliabue, A., et al. (2020). Tracking Improvement in Simulated Marine Biogeochemistry Between CMIP5 and CMIP6. *Current Climate Change Reports*. https://doi.org/10.1007/s40641-020-00160-0

Sellar, A. A., Jones, C. G., Mulcahy, J. P., Tang, Y., Yool, A., Wiltshire, A., et al. (2019). UKESM1: Description and Evaluation of the U.K. Earth System Model. *Journal of Advances in Modeling Earth Systems*, *11*(12), 4513–4558. https://doi.org/10.1029/2019MS001739

Sinha, B., Buitenhuis, E. T., Quéré, C. Le, & Anderson, T. R. (2010). Comparison of the emergent behavior of a complex ecosystem model in two ocean general circulation models. *Progress in Oceanography*, *84*(3–4). https://doi.org/10.1016/j.pocean.2009.10.003

Six, K. D., & Maier-Reimer, E. (1996). Effects of plankton dynamics on seasonal carbon fluxes in an ocean general circulation model. *Global Biogeochemical Cycles*, *10*(4). https://doi.org/10.1029/96GB02561

Stock, C. A., Dunne, J. P., Fan, S., Ginoux, P., John, J., Krasting, J. P., et al. (2020). Ocean Biogeochemistry in GFDL's Earth System Model 4.1 and Its Response to Increasing Atmospheric CO2. *Journal of Advances in Modeling Earth Systems*, *12*(10). https://doi.org/10.1029/2019MS002043

Swart, N. C., Cole, J. N. S., Kharin, V. V., Lazare, M., Scinocca, J. F., Gillett, N. P., et al. (2019). The Canadian Earth System Model version 5 (CanESM5.0.3). *Geoscientific Model Development*, *12*(11), 4823–4873. https://doi.org/10.5194/gmd-12-4823-2019

Tagliabue, A., Kwiatkowski, L., Bopp, L., Butenschön, M., Cheung, W., Lengaigne, M., & Vialard, J. (2021). Persistent Uncertainties in Ocean Net Primary Production Climate Change Projections at Regional Scales Raise Challenges for Assessing Impacts on Ecosystem Services. *Frontiers in Climate*, *3*. https://doi.org/10.3389/fclim.2021.738224

Thompson, B., Tkalich, P., Malanotte-Rizzoli, P., Fricot, B., & Mas, J. (2016). Dynamical and thermodynamical analysis of the South China Sea winter cold tongue. *Climate Dynamics*, *47*(5–6). https://doi.org/10.1007/s00382-015-2924-3

Tjiputra, J. F., Schwinger, J., Bentsen, M., Morée, A. L., Gao, S., Bethke, I., et al. (2020). Ocean biogeochemistry in the Norwegian Earth System Model version 2 (NorESM2). *Geoscientific Model Development*, *13*(5), 2393–2431. https://doi.org/10.5194/gmd-13-2393-2020

Wang, Z., Brickman, D., Greenan, B., Christian, J., DeTracey, B., & Gilbert, D. (2024). Assessment of Ocean Temperature Trends for the Scotian Shelf and Gulf of Maine Using 22 CMIP6 Earth System Models. *Atmosphere - Ocean*, *62*(1). https://doi.org/10.1080/07055900.2023.2264832

---

## Referee Report (RR1)

General comments:

The paper has been improved by incorporating a more in-depth discussion on the model structure, resolution, and physical processes. However, I still have some concerns regarding the datasets used in the model. If this study aims to conduct an evaluation, it should consider actual observations such as satellite products.

I understand that CMEMS datasets have been evaluated and shown good agreement with the World Ocean Atlas and ocean colour data. Nevertheless, I am curious as to why the authors did not use the available observations that was mentioned in the author's response (i.e. in-situ measurements taken from the Southern South China Sea). If these datasets are not available for the entire time series (1993-2014), could the authors consider using the specific years where observations are available?

Additionally, the manuscript could benefit from further clarity. Some sections are challenging to follow due to long paragraphs that are lacking a specific point, especially some parts of the introduction as well as the results and discussion section (chlorophyll and phytoplankton carbon, nitrate, oxygen, and the entire section 4.2). I recommend having someone outside the research group to review the manuscript for readability.

Due to the absence of observational data, I am currently unable to recommend this manuscript for publication.

Specific comments:

Line 17: The authors mostly refer southwest monsoon as JJA and northeast monsoon as DJF, so this is not representative of the text, please change to June-August and December-February.

Line 133-134: Can you provide a reference for this sentence

Line 148: I disagree, there are satellite products which count as an observation for chlorophyll (even primary production).

Line 153: Wahyudi et al., 2023 is missing from the reference list

Line 155: Triana et al., 2021 is missing from the reference list

Line 157: Chen et al., 2023 is missing from the reference list

In figures with maps, the authors show the model ensemble bias, but there are no specific comments about it in the text. Some of the figures and subfigures are also not referenced in the text.

In line 227-229, the authors mentioned that errors larger than +0.1 mg/m3 indicates a notable discrepancy, but in line 236-238, the authors said that >0.15 mg/m3 is within acceptable range – so the three models mentioned earlier are still within acceptable range?

Line 241-242: Do you mean: does not overestimate?

Line 253-268: great discussion!

Line 308-309: Do you mean: ranging from?

Perhaps the authors can show the ESM's nitrate profiles to show whether ESM can capture where the nitracline is during different season? (is this the dissolved inorganic nitrogen?)

Line 338: is this DJF?

Line 348-350: Can you provide reference for this?

Line 434: you mean all the biogeochemistry variables? Because some models can reproduce the pattern of at least one of the variables.

Section 4.2.2 – can you also make some comments on using a better phytoplankton parameterisation such as the nutrient quota? (or flexible N:C ratio of phytoplankton).

Line 520 – Taylor's diagram or Taylor diagram? Please be consistent.

Line 580-581 – You are repeating line 579 – 580.

Line 618 - Perhaps i am missing something but the authors have not mentioned about annual scales at all in the first half of the results and discussion; and is only touched in the Taylor diagram part

Line 620 – is this only at the surface? Perhaps also consider the deeper depths as well because this is the conclusion section.

Line 634 – do not use *etc*.

---

## Author Response (AR2)

Response to minor review of **"Evaluation of CMIP6 Models Performance in Simulating Historical Biogeochemistry across Southern South China Sea"**

**Manuscript egusphere-2024-72**

**Response to Anonymous Referee #2:**

We sincerely appreciate the time and effort invested by both the reviewer and the editor in re-evaluating our paper titled **"Evaluation of CMIP6 Models Performance in Simulating Historical Biogeochemistry across Southern South China Sea"** submitted for publication in Biogeosciences. We are grateful for the positive feedback and the insightful comments provided, which is detailed in this report and also in the revised manuscript. The line numbers **(L)** mentioned in the response refer to the line numbers in the revised manuscript. The newly added figures and tables in the revised manuscript or supplementary materials are also included in this report for the convenience of the reviewer and editor to refer.

*This report contains point-by-point detailed responses to each comment from the reviewer.

**REVIEWER GENERAL COMMENTS**

**Comment 1:**

The paper has been improved by incorporating a more in-depth discussion on the model structure, resolution, and physical processes. However, I still have some concerns regarding the datasets used in the model. If this study aims to conduct an evaluation, it should consider actual observations such as satellite products.

I understand that CMEMS datasets have been evaluated and shown good agreement with the World Ocean Atlas and ocean colour data. Nevertheless, I am curious as to why the authors did not use the available observations that was mentioned in the author's response (i.e. in-situ measurements taken from the Southern South China Sea). If these datasets are not available for the entire time series (1993-2014), could the authors consider using the specific years where observations are available?

**Response:**

Thank you for your comment regarding evaluating the CMEMS dataset with available observation and satellite data. Accordingly, we have presented the evaluation results in supplementary. **Table S1** and **Figs S1 & S2** shows the statistical test results of CMEMS vs observation from WOA18 and satellite data from GlobColour. The validation results demonstrated good agreement, with region-wide differences less than ±5% for chlorophyll and phytoplankton, and less than ±10% for nitrate and oxygen. The spatial pattern comparison indicates that the largest differences between the CMEMS and WOA observation data occur in coastal areas. These differences may be attributed to the insufficient number of WOA observation data in our study region **(see Fig. A below this response)** and the coarse resolution of WOA (~111 km). However, the small differences between their climatology (less than ±10%) give us confidence that CMEMS is reliable. Therefore, given that CMEMS has all the required parameters, and our analysis established the reliability of the CMEMS in our study region, we believe that using CMEMS as a reference data allows for a fair performance assessment of the CMIP6 ESMs across all the parameters evaluated.

This is also mentioned in the revised manuscript in **L152–L163** as:

*"Furthermore, we have assessed the CMEMS product using observation data from the World Ocean Atlas 2018 (WOA18) for nitrate and oxygen, and satellite data from GlobColour (Product ID: OCEANCOLOUR_GLO_BGC_L4_MY_009_104) for chlorophyll and phytoplankton. The validation results are presented in supplementary **Table S1**, with the spatial percentage bias detailed in supplementary **Figs. S1 – S2**. The validation results demonstrated good agreement, with region-wide differences less than ±5% for chlorophyll and phytoplankton, and less than ±10% for nitrate and oxygen. The spatial pattern comparison indicates that the largest differences between the CMEMS and WOA observation data occur in coastal areas. These differences may be attributed to the insufficient number of WOA observation data in our study domain and the coarse resolution of WOA (~111 km). However, the small differences between their climatology (less than ±10%) give us confidence that CMEMS is reliable. Therefore, given that CMEMS has all the required parameters, and our analysis established the reliability of the CMEMS in our study region, we believe that using CMEMS as a reference data allows for a fair performance assessment of the CMIP6 ESMs across all the parameters evaluated."*

[Figure]

**Figure A.** Number of WOA observation data and their distribution annually from 1960 to 2017 in southern South China Sea region. (This figure is not included in the manuscript or in supplementary).

\*\*\*\*\*\*\*\*\*\*\*\*\*\*\*\*\*\*\*\*\*\*\*\*\*\*\*\*\*\*\*\*\*\*\*\*\*

**Comment 2:**

Additionally, the manuscript could benefit from further clarity. Some sections are challenging to follow due to long paragraphs that are lacking a specific point, especially some parts of the introduction as well as the results and discussion section (chlorophyll and phytoplankton carbon, nitrate, oxygen, and the entire section 4.2). I recommend having someone outside the research group to review the manuscript for readability.

**Response:**

Thank you for your feedback. We tried our best to use simple words and phrases in this manuscript to ensure clarity for our readers. We have also asked some of our colleagues and students from different field to read the manuscript for comprehensibility, and it was found to be understandable.

\*\*\*\*\*\*\*\*\*\*\*\*\*\*\*\*\*\*\*\*\*\*\*\*\*\*\*\*\*\*\*\*\*\*\*\*\*

**REVIEWER SPECIFIC COMMENTS**

**Comment 1:**

Line 17: The authors mostly refer southwest monsoon as JJA and northeast monsoon as DJF, so this is not representative of the text, please change to June-August and December-February

**Response:**

Thank you for your feedback. We have made the changes according to your suggestion **L16**.

\*\*\*\*\*\*\*\*\*\*\*\*\*\*\*\*\*\*\*\*\*\*\*\*\*\*\*\*\*\*\*\*\*\*\*\*\*

**Comment 2:**

Line 133-134: Can you provide a reference for this sentence

**Response:**

Thank you for your feedback. After a thorough review, we found that (Pinkerton et al., 2021 and Yuwono & Rendy, 2023) are suitable references for the statement **L126-L128.**

Pinkerton, M. H., Boyd, P. W., Deppeler, S., Hayward, A., Höfer, J., & Moreau, S. (2021). Evidence for the Impact of Climate Change on Primary Producers in the Southern Ocean. Frontiers in Ecology and Evolution, 9. https://doi.org/10.3389/fevo.2021.592027

Yuwono, F. S., & Rendy. (2023). Seasonal response of coccolithophores and its potential to reconstruct paleomonsoon in the eastern Indonesian seas: An overview. In IOP Conference Series: Earth and Environmental Science (Vol. 1163). https://doi.org/10.1088/1755-1315/1163/1/012003

\*\*\*\*\*\*\*\*\*\*\*\*\*\*\*\*\*\*\*\*\*\*\*\*\*\*\*\*\*\*\*\*\*\*\*

**Comment 3:**

Line 148: I disagree, there are satellite products which count as an observation for chlorophyll (even primary production).

**Response:**

Thank you for your suggestion. We agree with your assessment and have included satellite observations in our study for the evaluation of the chlorophyll and phytoplankton data from CMEMS in supplementary **Table S1** and **Figs S1 & S2**.

\*\*\*\*\*\*\*\*\*\*\*\*\*\*\*\*\*\*\*\*\*\*\*\*\*\*\*\*\*\*\*\*\*\*\*

**Comment 4:**

Line 153: Wahyudi et al., 2023 is missing from the reference list
Line 155: Triana et al., 2021 is missing from the reference list
Line 157: Chen et al., 2023 is missing from the reference list

**Response:**

Thank you for bringing up this. We apologize for our oversight in this matter. Accordingly, we have provided the following reference.

Chen, Q., Li, D., Feng, J., Zhao, L., Qi, J., & Yin, B. (2023). Understanding the compound marine heatwave and low-chlorophyll extremes in the western Pacific Ocean. Frontiers in Marine Science, 10. https://doi.org/10.3389/fmars.2023.1303663.

Triana, K., Wahyudi, A. J., Murakami-Sugihara, N., & Ogawa, H. (2021). Spatial and temporal variations in particulate organic carbon in Indonesian waters over two decades. Marine and Freshwater Research, 72(12). https://doi.org/10.1071/MF20264.

Wahyudi, A. J., Triana, K., Masumoto, Y., Rachman, A., Firdaus, M. R., Iskandar, I., & Meirinawati, H. (2023). Carbon and nutrient enrichment potential of South Java upwelling area as detected using hindcast biogeochemistry variables. Regional Studies in Marine Science, 59. https://doi.org/10.1016/j.rsma.2022.102802.

\*\*\*\*\*\*\*\*\*\*\*\*\*\*\*\*\*\*\*\*\*\*\*\*\*\*\*\*\*\*\*\*\*\*\*

**Comment 5:**

In figures with maps, the authors show the model ensemble bias, but there are no specific comments about it in the text. Some of the figures and subfigures are also not referenced in the text.

**Response:**

Thank you for your feedback. The main goal of our study is to assess the skill of individual models. We included the model ensemble as an additional data. To avoid any potential confusion for future readers, we better remove the model ensemble from all the figures.

\*\*\*\*\*\*\*\*\*\*\*\*\*\*\*\*\*\*\*\*\*\*\*\*\*\*\*\*\*\*\*\*\*\*\*

**Comment 6:**

In line 227-229, the authors mentioned that errors larger than +0.1 mg/m3 indicates a notable discrepancy, but in line 236-238, the authors said that >0.15 mg/m3 is within acceptable range – so the three models mentioned earlier are still within acceptable range?

**Response:**

Thank you for bringing this to our attention. We apologize for the oversight. The acceptable range should be less than 1 mg/m³. This has been corrected in **L232** of the revised manuscript.
* * *
**Comment 7:**

Line 241-242: Do you mean: does not overestimate?

**Response:**

Thank you for bringing this to our attention. YES, we mean does not overestimate. We rephrase the statement *"UKESM1-0-LL model not overestimating chlorophyll"* to *"UKESM1-0-LL model does not overestimate chlorophyll"* in **L236.**
* * *
**Comment 8:**

Line 253-268: great discussion!

**Response:**

Thank you very much for your appreciation.
* * *
**Comment 9:**

Line 308-309: Do you mean: ranging from?

**Response:**

Thank you for bringing this spelling mistake to our attention. We have corrected accordingly in **L294** as *"rang"* to *"range".*
* * *
**Comment 10:**

Perhaps the authors can show the ESM's nitrate profiles to show whether ESM can capture where the nitracline is during diberent season? (is this the dissolved inorganic nitrogen?)

**Response:**

Thank you for your feedback. Accordingly, we have provided both ESM's nitrate and oxygen profile for both seasons (DJF & JJA) in the supplementary **Figs. S3-S4** for nitrate and **Figs. S5-S6** for oxygen. This is also mentioned in the revised manuscript in **L290** for nitrate and in **L315** for oxygen.

YES, the nitrate utilized in our study is a type of dissolved inorganic nitrogen.
* * *
**Comment 11:**

Line 338: is this DJF?

**Response:**

Thank you for your feedback. Yes, it is DJF.

\*\*\*\*\*\*\*\*\*\*\*\*\*\*\*\*\*\*\*\*\*\*\*\*\*\*\*\*\*\*\*\*\*\*\*\*

**Comment 12:**

Line 348-350: Can you provide reference for this?

**Response:**

Thank you for your feedback. After a thorough review, we found that Séférian et al., 2020 is suitable references for this statement in **L335**.

\*\*\*\*\*\*\*\*\*\*\*\*\*\*\*\*\*\*\*\*\*\*\*\*\*\*\*\*\*\*\*\*\*\*\*\*

**Comment 13:**

Line 434: you mean all the biogeochemistry variables? Because some models can reproduce the pattern of at least one of the variables.

**Response:**

Thank you for your feedback. We agree with your statement and have articulated a similar viewpoint. In the subsequent line **L400**, we already elaborated on this matter by stating, *"While some ESMs can effectively reproduce the reference pattern for individual variables, there remains significant uncertainty regarding the reasons why some ESMs outperform others in this respect."*

\*\*\*\*\*\*\*\*\*\*\*\*\*\*\*\*\*\*\*\*\*\*\*\*\*\*\*\*\*\*\*\*\*\*\*\*

**Comment 14:**

Section 4.2.2 – can you also make some comments on using a better phytoplankton parameterisation such as the nutrient quota? (or flexible N:C ratio of phytoplankton).

**Response:**

Thank you for your suggestion. Accordingly in **L474-L482,** we provided a short comment bases on your suggestion as:

*"Furthermore, this analysis highlights significant variability in phytoplankton-nutrient correlations across CMIP6 models, the observed discrepancies underscore the potential benefits of employing more advanced phytoplankton parameterizations, such as nutrient quota or flexible N:C ratios. These approaches could provide a more nuanced representation of phytoplankton response to nutrient availability. Models like UKESM1-0-LL, which utilize nitrogen as their primary currency for phytoplankton biomass, demonstrate good correlations with nitrate, suggesting that explicit consideration of nutrient stoichiometry may enhance model accuracy. Similarly, integrating phosphorus cycles, as seen in MIROC-ES2L, could better capture phosphorus limitations affecting phytoplankton growth. Future model developments should prioritize these parameterizations to improve the fidelity of biogeochemical simulations and better understand ecosystem responses to environmental changes."*

\*\*\*\*\*\*\*\*\*\*\*\*\*\*\*\*\*\*\*\*\*\*\*\*\*\*\*\*\*\*\*\*\*\*\*\*

**Comment 15:**

Line 520 – Taylor's diagram or Taylor diagram? Please be consistent.

**Response:**

Thank you for pointing this out. We have corrected this error to align consistently with the Taylor diagram.
* * *
**Comment 16:**

Line 580-581 – You are repeating line 579 – 580.

**Response:**

Thank you for pointing this out. We have revised and corrected this error in **L541-L543 from** *"In addition to the qualitative analysis presented in the Taylor diagram above, a skill score is calculated using Equation (5) to further validate the models' proficiency in reproducing biogeochemical variables. The Taylor skill score, derived from Equation (5), serves as a quantitative summary of the information conveyed by the Taylor diagram, providing a synthetic measure of the models' performance."* **to** *"In addition to the qualitative analysis presented in the Taylor diagram, a skill score is calculated using Equation (5) to further validate the models' proficiency in reproducing biogeochemical variables, serving as a quantitative summary of the information conveyed by the Taylor diagram."*
* * *
**Comment 17:**

Line 618 - Perhaps i am missing something but the authors have not mentioned about annual scales at all in the first half of the results and discussion; and is only touched in the Taylor diagram part

**Response:**

Thank you for your comment. The initial part of our analysis involves evaluating the models' ability to simulate the climatology of biogeochemical parameters in the study area, i.e., southern South China Sea (SSCS). This region is significantly influenced by two monsoon regimes: the boreal summer (JJA) and boreal winter monsoon (DJF). Therefore, it is important for the models to have a high skill in capturing the seasonal climatology, even though our primary focus is on how biogeochemical parameters will be affected by climate change on an annual time scale. Evaluating model performance solely on an annual basis could lead to selecting models that may not perform well seasonally, yet at annual scale they might look fine. Thus, for the first part of the evaluation, we assessed the models on a seasonal scale. This approach ensures confidence in our later future projection assessments, which focus on annual time scale changes of biogeochemical parameters in our study area.
* * *
**Comment 18:**

Line 620 – is this only at the surface? Perhaps also consider the deeper depths as well because this is the conclusion section.

**Response:**

Thank you for your suggestion. Accordingly, we have presented the statistical results for nitrate and oxygen at deep layer of 1000m in **L581-L585** as *"Similarly, at the depth of 1000m, GFDL-ESM4 and MRI-ESM2-0 models alone shows positive correlation of 0.02 and 0.46, respectively and the remaining*

*models showed negative correlation ranging -0.77 to -0.08. At the depth of 1000m for oxygen, ACCESS-ESM1-5, GFDL-ESM4 and UKESM1-0-LL alone showed negative correlation of -0.2, -0.26 and -0.06, respectively and the remaining models showed positive correlation ranging 0.05 to 0.6.''*

\*\*\*\*\*\*\*\*\*\*\*\*\*\*\*\*\*\*\*\*\*\*\*\*\*\*\*\*\*\*\*\*\*\*\*\*

**Comment 19:**

Line 634 – do not use etc.

**Response:**

Thank you for suggestion. We have rectified this.

\*\*\*\*\*\*\*\*\*\*\*\*\*\*\*\*\*\*\*\*\*\*\*\*\*\*\*\*\*\*\*\*\*\*\*\*

**FIGURE & TABLES:**

**Table S1** Validation results of CMEMS (1993 - 2014) climatology against the Observation climatology of WOA18 data (1960 – 2017) and Satellite data (GlobColour: 1997 - 2014) across the study domain (southern South China Sea). Satellite provides chlorophyll and phytoplankton data and WOA18 provides nitrate and oxygen data. Correlation Coefficient (CC), Root Mean Square Difference (RMSD), Mean Bias Error (MBE) and Mean Percentage Bias (MPB).

| Variables | CC | | RMSD | | MBE | | MPB (%) | |
|---|---|---|---|---|---|---|---|---|
| | DJF | JJA | DJF | JJA | DJF | JJA | DJF | JJA |
| chlorophyll (mg m$^{-3}$) | 0.69 | 0.68 | 0.19 | 0.2 | -0.18 | -0.21 | -3.5 | -4.8 |
| phytoplankton (mmol m$^{-3}$) | 0.68 | 0.7 | 1.19 | 1.27 | 0.55 | 0.54 | 3.25 | 3.17 |
| nitrate (mmol m$^{-3}$) | 0.33 | 0.31 | 0.33 | 0.36 | -0.1 | -0.1 | -0.96 | -0.97 |
| oxygen (mmol m$^{-3}$) | 0.47 | 0.68 | 8.4 | 6.7 | 1.9 | 0.4 | 2.79 | 1.2 |

[Figure]

**Figure S1** Seasonal Percentage Bias of chlorophyll (a-b) and phytoplankton (c-d) from CMEMS against Satellite data (GlobColour).

[Figure]

**Figure S2** Seasonal Percentage Bias of nitrate (a-b) and oxygen (c-d) from CMEMS against observation data (WOA18).

**Table S2** Spatial statistics of nitrate and oxygen at depth of 1000m for the selected 13 CMIP6 ESMs.

| CMIP6 ESMs | nitrate 1000m | | | oxygen 1000m | | |
|---|---|---|---|---|---|---|
| | CC | NSD | NRMSD | CC | NSD | NRMSD |
| ACCESS-ESM1-5 | -0.20 | 7.01 | 7.27 | -0.21 | 5.70 | 5.99 |
| CanESM5 | -0.08 | 1.77 | 2.10 | 0.44 | 1.27 | 1.22 |
| CanESM5-CanOE | -0.20 | 2.45 | 2.82 | 0.22 | 2.12 | 2.14 |
| GFDL-ESM4 | 0.02 | 3.86 | 3.96 | -0.26 | 3.04 | 3.43 |
| MIROC-ES2H | -0.77 | 0.18 | 1.14 | 0.30 | 0.69 | 1.03 |
| MIROC-ES2L | -0.27 | 0.12 | 1.04 | 0.34 | 0.31 | 0.94 |
| MPI-ESM1-2-HR | -0.79 | 1.62 | 2.49 | 0.61 | 2.91 | 2.43 |
| MPI-ESM1-2-LR | -0.77 | 0.89 | 1.78 | 0.05 | 1.89 | 2.10 |
| MPI-ESM-1-2-HAM | -0.74 | 0.60 | 1.49 | 0.19 | 1.73 | 1.82 |
| MRI-ESM2-0 | 0.46 | 2.10 | 1.86 | 0.25 | 1.66 | 1.71 |
| NorESM2-LM | -0.14 | 0.34 | 1.10 | 0.49 | 1.11 | 1.07 |
| NorESM2-MM | -0.21 | 0.43 | 1.17 | 0.53 | 1.38 | 1.20 |
| UKESM1-0-LL | -0.43 | 1.93 | 2.53 | -0.06 | 2.65 | 2.89 |

[Figure]

**Figure S3** Depth profile of nitrate up to 1000 meters for reference (CMEMS) and observation (WOA18) data with 13 selected CMIP6 ESMs

[Figure]

**Figure S4** same as Figure S3 but for depth from 1000 to 6000 meters.

[Figure]

**Figure S5** Depth profile of oxygen up to 1000 meters for reference (CMEMS) and observation (WOA18) data with 13 selected CMIP6 ESMs

[Figure]

**Figure S6** same as Figure S5 but for depth from 1000 to 6000 meters.